# A previously uncharacterized Factor Associated with Metabolism and Energy (FAME/C14orf105/CCDC198/1700011H14Rik) is related to evolutionary adaptation, energy balance, and kidney physiology

In this study we use comparative genomics to uncover a gene with uncharacterized function (*1700011H14Rik/C14orf105/CCDC198*), which we hereby name *FAME* (Factor Associated with Metabolism and Energy). We observe that *FAME* shows an unusually high evolutionary divergence in birds and mammals. Through the comparison of single nucleotide polymorphisms, we identify gene flow of *FAME* from Neandertals into modern humans. We conduct knockout experiments on animals and observe altered body weight and decreased energy expenditure in *Fame* knockout animals, corresponding to genome-wide association studies linking *FAME* with higher body mass index in humans. Gene expression and subcellular localization analyses reveal that FAME is a membrane-bound protein enriched in the kidneys. Although the gene knockout results in structurally normal kidneys, we detect higher albumin in urine and lowered ferritin in the blood. Through experimental validation, we confirm interactions between FAME and ferritin and show co-localization in vesicular and plasma membranes.

Through natural selection, major animal groups have developed unique mechanisms for adaptations to the environment. The genetic landscape corresponds to developmental, morphological, and physiological adaptations[1]. Gene regulatory regions undergo rapid evolutionary change to tune the production of mRNAs encoding the actual effectors of adaptation, the proteins. The gene products also undergo natural selection, which shapes them according to various benefits derived from their functions[2]. Importantly, not all such products, including mainly proteins, are essential for basic embryonic development and the mere survival of animals. Numerous gene knockout experiments in mice have highlighted a cohort of proteins that are functionally important to some extent, and yet the animals can live perfectly without them under beneficial circumstances[3]. These nonessential proteins convey an adaptive advantage to their hosts when the animals are exposed to the diversity of challenging natural environments[4].

Furthermore, these seemingly non-essential genes might become a basis for diverse pathologies or loss of fitness[5,6]. Finally, similar to essential genes, non-essential genes might be a perfect substrate for evolution, especially for tuning metabolic, stress- and energy-related features. Being non-essential, such genes can evolve much faster to provide the necessary evolvability under intense selective pressure. In extreme cases, the evolution of proteins (especially when it comes to a non-essential group) might result in a complete change of function[7,8] or a pseudogenization, which occurs in genes involved in dental genesis in birds, turtles, and toothless mammals[9].

The comparative genomics approach[10] is perfectly designed to elucidate genomic protein-coding and non-coding regions responsible

✉e-mail: julian.petersen@medizin.uni-leipzig.de; igor.adameyko@meduniwien.ac.at

for different animal groups' divergence and adaptive radiation[11,12]. For instance, such analyses brought forward the genomic changes associated with "birdness" or "mammalness". Many identified regions appeared to be involved in the evolution of egg production, placental development, or genital shaping[11,12]. Indeed, the divergence of major vertebrate groups such as birds, reptiles, and mammals resulted in heavy modifications of metabolism[13,14], reproduction[15], and excretion[16], together with associated genomic changes and adapting protein structures. For example, the processes of water and nutrient re-absorption in birds and mammals differ dramatically at structural, cellular, and molecular levels, including genetics. Birds and reptiles predominantly excrete uric acid instead of the urea used by mammals and, thus, rely on negligible amounts of water for nitrogen excretion[17].

Although many of the comparative studies pinpointed well-characterized genes that can be analyzed in the context of functional networks involved in the diverging organ systems and physiological functions, the uncharacterized genes remained enigmatic in this evolutionary paradigm. Although the human and mouse genomes contain around 20.000 protein-coding genes, not all of these are identified, annotated, and characterized in terms of their expression and biological function[18-20]. Characterizing such genes functionally and investigating their evolutionary roles are essential to complete the holistic picture of genome transformations through time.

Here we uncovered an uncharacterized protein-coding *1700011H14Rik/C14orf105/CCDC198* gene hereby named *FAME* (*Factor Associated with Metabolism and Energy*), that evolves at an exceptional rate in birds and mammals. Specific alleles of *FAME* flow from Neandertals into modern humans, highlighting its involvement in our fitness. We addressed the expression, subcellular localization, molecular structure, functional roles, and potential disease association of *FAME*. Our results establish *FAME* as a fast-evolving gene modulating iron exchange, excretion, energy expenditure, and processes potentially associated with cancer progression.

## Results

### FAME sequence evolves at an extra-high rate during the divergence of reptiles, birds, and mammals

To identify previously identified genes with uncharacterized function that could ensure diverging adaptations in major amniote groups, we took advantage of the comparative genomics approach. Evolutionary pressure on proteins are often quantified by the ratio of substitution rates at non-synonymous and synonymous sites. To elucidate proteins co-evolving with major vertebrate groups, we compared the ratio of the number of non-synonymous substitutions per non-synonymous site (dN) to the number of synonymous substitutions per synonymous site (dS) (dN/dS signature) for 27, 16, and 28 pairs of genomes of birds, reptiles and mammals, respectively (Supplementary Data 1 and 2). We started with 20 pairs of mammals and birds and added 8 more pairs of mammals and 7 more pairs of birds to increase diversity. We had difficulty finding pairs for reptiles because of fewer available genomes and fewer branches on their evolutionary tree, but included several selected pairs from different clades of reptiles.

We identified 312 proteins, which showed significantly different dN/dS ratios between reptiles and birds (FDR < 0.001) (Supplementary Data 3). Among them, 129 genes had significantly higher dN/dS in reptiles, and 183 proteins had significantly higher dN/dS in birds. Interestingly, when performing bidirectional comparisons to identify proteins with the most flexible sequence in reptiles, only three proteins (STOX1, CEP126, and CCDC198) had a mean of identity of less than 60% (Fig. 1a, b and Supplementary Data 4). Two of them have been previously described. STOX1 is a protein involved in free radical equilibrium and mitochondrial function[21], whereas CEP126 is a centrosomal protein involved in primary cilium formation[22]. The third protein, CCDC198/C14orf105 in humans or 1700011H14Rik in mice (hereafter called FAME, Factor Associated with Metabolism and

Energy), however, has not been characterized yet. *FAME* demonstrated a higher average of dN/dS in mammals and birds than in reptiles: 0.3912, 0.4235, and 0.2779, respectively, which indicates a high evolutionary rate in mammals and birds (Fig. 1b and Supplementary Data 4).

Despite the high rate of evolutionary changes, this protein did not undergo pseudogenization in any of the studied clades (Supplementary Fig. 1a). Next, we tested how this gene's evolution coincided with the animals' lifestyles. For this, we created the matrix of lifestyles based on the Thera-base database (https://esapubs.org/archive/ecol/E090/184/metadata.htm ("PanTHERIA_1-0_WR93_Aug2008.txt")) with 42 recorded parameters for species with known and published genomes. Then we tested correlations between these parameters and protein alignments, but also with the selection of specific regions in the alignments (Fig. 1c, Supplementary Fig. 2a). The result suggested that specific portions of FAME co-evolved with metabolic and excretion traits (Fig. 1d, Supplementary Fig. 2b). This indicated that FAME might be involved in the control of water and nutrient exchange. To test this, we created our own matrix based on the animals' habitats, including desert-living species and water-dwelling mammals such as whales (Supplementary Data 5).

Interestingly, scanning for the highest correlation of intracellular location suggested the importance of the N-terminal region within the protein. A more in-depth sequence analysis revealed a possible N-myristoylation site and several phosphorylation sites (Supplementary Fig. 3a–d, Supplementary Data 6). This analysis also showed correlations of the FAME structure with the habitat (Supplementary Fig. 3e). Furthermore, no homologs in any of the explored genomes were evident, whereas a domain with unknown function DUF4619 was present.

To test if evolutionary changes of *FAME* also occurred in humans, we next analyzed the evolutionary modifications in *FAME* since the split between modern humans and Neanderthals. Although some contact may have occurred[23], Neanderthals and modern humans evolved independently before the major out-of-Africa dispersal ~70,000 years ago[24,25]. During ~0.5 million years, these two lineages accumulated mutations that reached fixation, or near fixation in their genome, in both groups. To explore the recent evolution in *FAME* we investigated all single-nucleotide polymorphisms (SNPs) for which 108 Nigerian Yorubans (representing the modern human)[26] carry one allele in a homozygous form and three high-coverage Neanderthals[27-29] homozygously carry a different allele. In this region encompassing *FAME* and 50 kb upstream and downstream (chr14:57886018-58010575, hg19), which we compared with the ancestral allele, we found 23 such SNPs (Fig. 1e, f, Supplementary Fig. 1b, c). For 20 of these SNPs, modern humans carry the ancestral allele, and for three SNPs Neanderthals carry the ancestral allele. That more alleles are derived from the Neanderthal lineage is compatible with the lower effective population size of Neanderthals[30], and the fact that we only have three high-coverage Neanderthals and 108 Yorubans bias us to detect alleles that are derived on the Neanderthal lineage.

Overall, these results indicate evidence for gene flow from Neanderthals at the locus encompassing *FAME*. However, there are two alternative scenarios as to why the Neanderthal-like alleles in Fig. 1f could be found among present-day people. These alleles were present in the shared ancestral population between Neanderthals and modern humans, or these alleles were introduced by gene flow from Neanderthals when these two groups met[31]. In the geneflow scenario, we expect to find these alleles on long Neanderthal-like haplotypes because meiotic recombination has not had time to break these DNA segments down to shorter pieces during the time since the gene flow took place (~50,000 years). To explore these two scenarios, we calculated the linkage disequilibrium between the 23 SNPs in Fig. 1f. We found that 14 of these alleles are inherited together (r2 > 0.8) among the individuals (n = 2504) in the 1000 Genomes Project

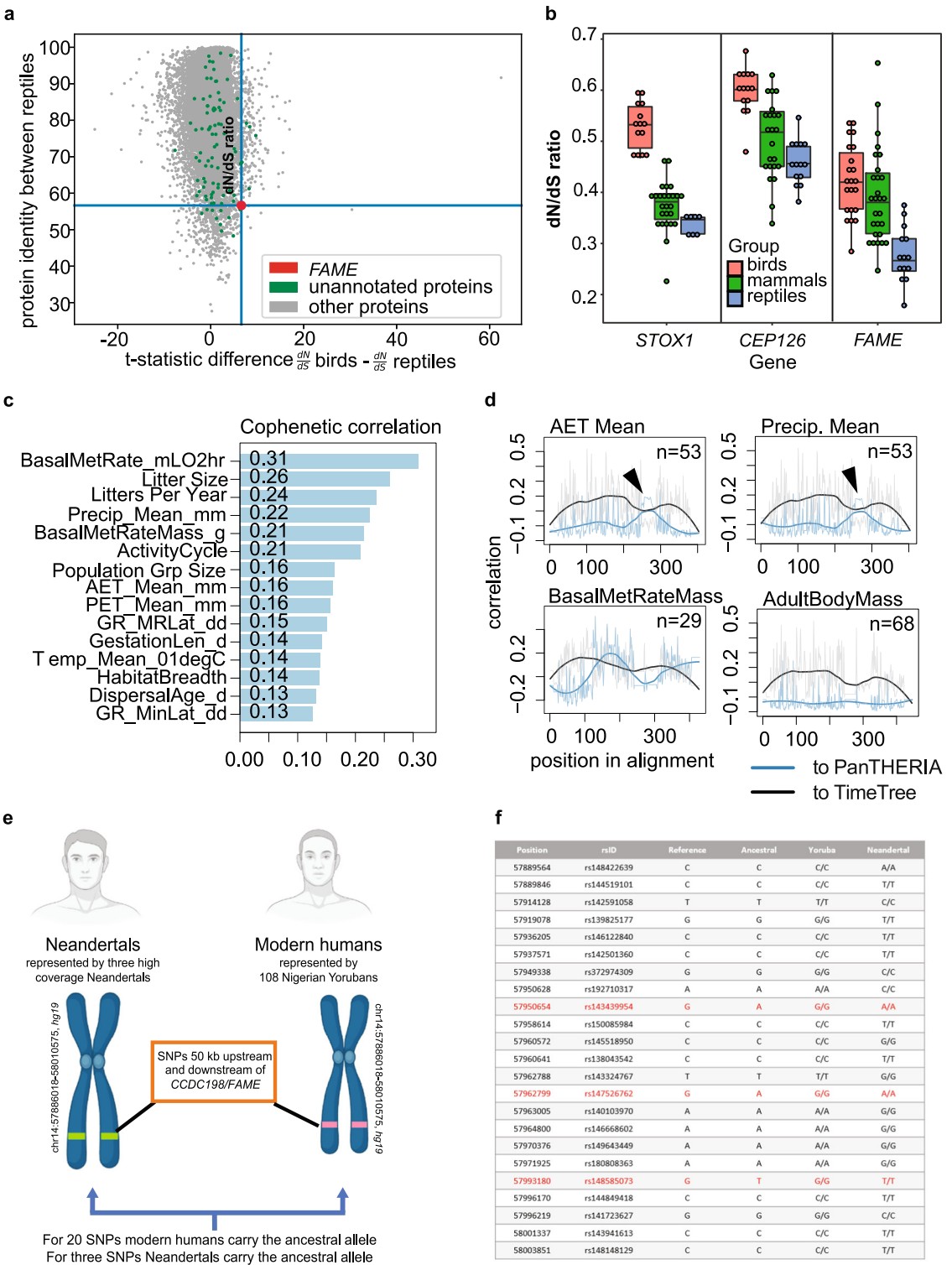

**Fig. 1 | *FAME* sequence changes during the evolutionary divergence of reptiles, birds, and mammals. a** Identification of FAME in a screen for evolutionary diverging proteins corresponding to the split of major vertebrate groups. **b** Comparison of dN/dS ratios of unannotated proteins of low protein identity between reptiles. Mean ± SEM and *n* (genome pairs): *STOX1* birds 0.5297 ± 0.0120 *n* = 14 mammals 0.3733 ± 0.0095 *n* = 26 reptiles 0.3369 ± 0.0071 *n* = 7. *CEP126* birds 0.5982 ± 0.0126 *n* = 14 mammals 0.5026 ± 0.0159 *n* = 23 reptiles 0.4581 ± 0.0116 *n* = 14. *FAME* birds 0.4235 ± 0.0150 *n* = 22 mammals 0.3912 ± 0.0176 *n* = 28 reptiles 0.2779 ± 0.0144 *n* = 14. Descriptive statistics can be accessed in Supplementary Data 13. **c** Cophenetic correlation between dendrograms obtained from amino acid sequence alignments and dendrograms based on PanTHERIA scores of different ecological

factors. The longest protein sequences were obtained for 68 mammalian species using biomaRt (Ensembl). **d** correlation between 2-mer alignment regions to ecologic (PanTHERIA) and phylogenetic (TimeTree) dendrograms. The species count with both ecologic data and orthologue sequence available is indicated in the top right corner. Regions with trends ties are marked with arrows. These regions are likely connected with ecological factors. Other examples are shown in Supplementary Fig. 2. Notably, there are different shapes of trends and positions of the ties. **e**, **f** Single nucleotide polymorphisms (SNPs) for which Neanderthals (*n* = 3) and Yorubans (*n* = 108) homozygously carry different alleles. Genomic coordinates are in hg19. The ancestral alleles were taken from Ensembl[90] and the Neanderthal alleles from previously published genomes[27,28].

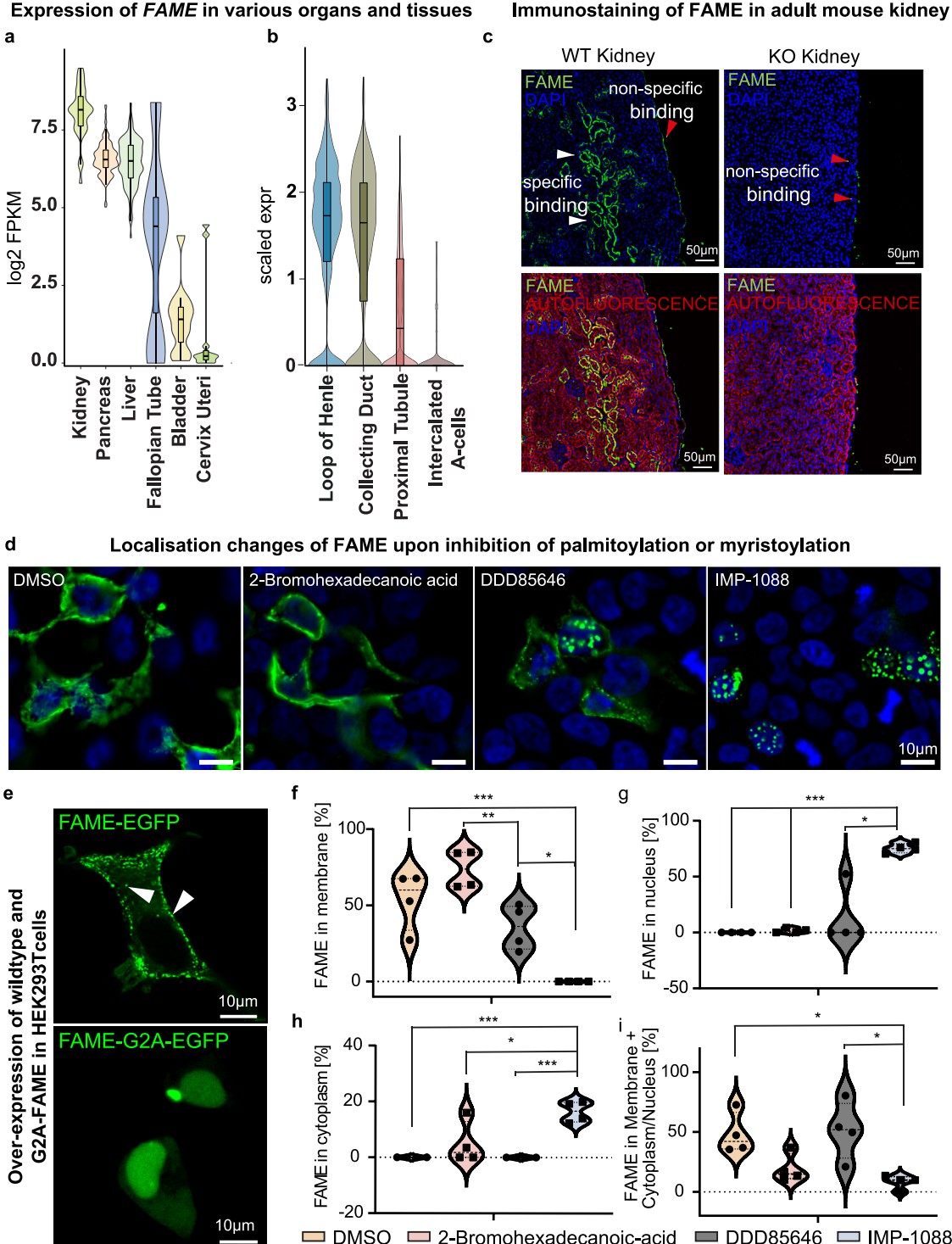

**a** Expression of *FAME* in various organs and tissues

**c** Immunostaining of FAME in adult mouse kidney

**d** Localisation changes of FAME upon inhibition of palmitoylation or myristoylation

**e** Over-expression of wildtype and G2A-FAME in HEK293Tcells

(Supplementary Fig. 1b). This haplotype, tagged by rs149643449-G, has a length of ~87 kb (chr14:57958614-58046101) and includes the promoter region and the first two exons of FAME (Supplementary Fig. 1c), as well as the first four exons of the neighboring gene SLC35F4. We investigated if a haplotype this length could have survived since the time of the common ancestor as previously described[32], i.e., using the equation 1-GammaCDF (m, shape = 2, rate = 1/L), where m is the measured haplotype length and L the expected length given by the equation L = 1/(r × t). Here r is the recombination rate per generation per bp and t is the length of the human and Neanderthal branches since divergence. Furthermore, we used the local recombination rate (1.47 cM per Mb)[33], and previously published estimates of branch

lengths and generation time[32]. Under this assumption, the probability of a haplotype of this length surviving since the common ancestor of modern humans and Neanderthals is low ($p = 3.7e{-}06$). We thus conclude that this haplotype has been introduced in the gene pool of present-day people by gene flow from Neanderthals. In the 1000 Genomes data set[26], these haplotypes are found at low frequencies in Asia, reaching a maximum allele frequency of 1.0% among Han Chinese ($n = 208$) and in admixed Americans, where it reaches an allele frequency of 0.8% in people of Mexican ancestry ($n = 64$).

**FAME is a membrane-associated protein enriched in the kidney, pancreas, liver, and fallopian tube.** Using Genotype-Tissue

**Fig. 2 | FAME is highly expressed in kidneys and localises to the cell membranes and vesicles. a** Analysis of *FAME* expression in various human tissues based on Genotype-Tissue Expression data. *FAME* expression is particularly high in the kidney, pancreas, liver, and fallopian tube. Mean ± SEM and *n* (GTEx samples): kidney (8.071 ± 0.1468 *n* = 32) pancreas (6.575 ± 0.0392 *n* = 171) liver (6.447 ± 0.06853 *n* = 119) fallopian tube (4.702 ± 1.230 *n* = 5) bladder (1.414 ± 0.3387 *n* = 11) cervix uteri (0.9330 ± 0.5028 *n* = 11.) **b** Detailed expression analysis of *Fame* using Tabula Muris, a single-cell atlas of the mouse. The data demonstrate high expression in kidney epithelial cell types, the loop of Henle and collecting duct cells, but also in the proximal tubules. Mean ± SEM and *n* (single mouse cells): loop of Henle (1.546 ± 0.03942 *n* = 471) collecting duct (1.403 ± 0.04278 *n* = 443) proximal tubule (0.6419 ± 0.02091 *n* = 1198) intercalated A-cells (0.07076 ± 0.03823 *n* = 45). **c** Immunofluorescence of the adult wild type and knockout mouse kidney stained for FAME and imaged together with auto fluorescence. Representative image from 3 different animals is shown. See Supplementary Fig. 5 for additional stainings. **d** Validation of the FAME N-myristoylation site. Visualization of overexpressed fluorescently tagged FAME-EGFP in HEK293T cells upon treatment with the N-myristoylation inhibitors DDD85646 and IMP-1088 and the palmitoylation inhibitor 2-bromohexadecanoic acid (2-BP) as control. Data from 4 independent experiments is shown. **e** Overexpression of the fusion protein in HEK293T cells shows the localisation of FAME in the plasma membrane and intracellular vesicles (white arrows). **f–i** Quantification of localisation changes of overexpressed FAME-GFP upon N-myristoylation inhibition in cellular compartments. **f** Violin plots of the percentage of FAME-EGFP localised in the plasma membrane upon treatment with the indicated inhibitors. Mean ± SEM and *n*: DMSO (53.78 ± 9.491 *n* = 4) 2-BP (73.71 ± 6.262 *n* = 4) DDD85646 (35.62 ± 7.446 *n* = 4) IMP-1088 (0.00 ± 0.00 *n* = 4), *p*-value DMSO vs IMP-1088 = 0.0013, *p*-value 2-BP vs DDD85646 = 0.0078, *p*-value DDD85646 vs IMP-1088 = 0.0030. **g** Violin plots for nuclear FAME localization upon treatment. Mean ± SEM and *n*: DMSO (0.00 ± 0.00 *n* = 4) 2-BP (1.622 ± 1.029 *n* = 4) DDD85646 (13.10 ± 13.10 *n* = 4) IMP-1088 (75.36 ± 2.151 *n* = 4), *p*-value DMSO vs IMP-1088 = 0.0001, *p*-value 2-BP vs IMP-1088 = 0.0001. **h** Violin plots for cytoplasmic Fame localization upon treatment. Mean ± SEM and *n*: DMSO (0.00 ± 0.00 *n* = 4) 2-BP (4.882 ± 3.806 *n* = 4) DDD85646 (0.00 ± 0.00 *n* = 4) IMP-1088 (16.38 ± 1.882 *n* = 4), *p*-value DMSO vs IMP-1088 = 0.0001, *p*-value 2-BP vs IMP-1088 = 0.0351, *p*-value DDD85646 vs IMP-1088 = 0.0001. **i** Violin plots for membranous and cytoplasmic/nuclear Fame localisation upon treatment. Mean ± SEM and *n*: DMSO (48.09 ± 8.612 *n* = 4) 2-BP (19.05 ± 5.971 *n* = 4) DDD85646 (51.34 ± 12.16 *n* = 4) IMP-1088 (8.255 ± 2.894 *n* = 4), *p*-value DMSO vs IMP-1088 = 0.0046, *p*-value DDD85646 vs IMP-1088 = 0.0137. Source data are provided as a Source Data file.

Expression (GTEx) data, we discovered that the expression of *FAME* was particularly high in the kidney and to a smaller extent in the pancreas, liver, and fallopian tube (Fig. 2a and Supplementary Fig. 4a). The analysis of publicly available single-cell transcriptomics data of the mouse kidney[34,35] further confirmed the specific expression in kidney epithelial cell types. This includes the loop of Henle and collecting duct cells, proximal tubules, and minor expression in intercalated A cells (Fig. 2b and Supplementary Fig. 4b).

By utilizing publicly available mass spectrometry data, we found evidence for the presence of FAME at the protein level in different tissues and species. For instance, FAME protein is detected in ProteomicsDB [https://www.proteomicsdb.org], Phosphomouse [https://phosphomouse.hms.harvard.edu] and PeptideAtlas [https://db.systemsbiology.net/sbeams/cgi/PeptideAtlas/Search] public mass spectrometry databases[36–39]. Strong experimental evidence for FAME protein production exists in both the human[40] and mouse kidney[37]. Furthermore, FAME protein was detected in cultured murine collecting duct cells[41], validating the presence of FAME protein in a cell type shown to produce its mRNA in vivo (Fig. 2b).

Therefore, we next focused on the kidney and validated the presence of FAME protein in the proximal tubules by immunohistochemistry. This is supported by the fact that we did not detect FAME in samples from knockout animals (Fig. 2c and Supplementary Fig. 5). Importantly, we ensured that our antibody is functional and specific via detecting FAME as a part of FAME-EGFP fusion in cultured cells that do not produce FAME endogenously (Supplementary Figs. 5 and 6). However, although we validated the functionality of the antibody, we must also acknowledge its limitations connected to potential low sensitivity, which results in inability to detect FAME in western blot without overexpressing FAME, which we discussed in detail in the method section.

With the help of molecular cloning and overexpression in HEK293T cells, we found that FAME localizes to plasma membranes as well as to small cytoplasmic vesicles (Fig. 2d, e, and Supplementary Fig. 4c). To further validate the predicted myristoylation site (Supplementary Fig. 3c), we treated HEK293T cells overexpressing FAME-EGFP with IMP-1088 (an inhibitor of the human N-myristoyltransferases NMT1 and NMT2) and DDD85646 (an inhibitor of *T. brucei* N-myristoyltransferase (TbNMT)), as well as, 2-Bromohexadecanoic acid as a negative control (a non-selective inhibitor of lipid metabolism) (Fig. 2d, f–i). These results show that once HEK293T cells are treated with myristoylation-specific inhibitors, the localization of FAME shifts from the plasma membrane toward the nucleus (Fig. 2f–i). DMSO treatment or the non-selective inhibition of lipid metabolism did not alter the localisation of FAME. These results support the existence of a myristoylation site in FAME. In addition, mutating the amino-terminal glycine residue (site of myristoylation) to alanine also resulted in the nuclear localisation of the protein (Fig. 2e). Live-cell imaging experiments revealed fast trafficking of FAME in the membranes and exocytosis-related or exosome transport from transfected to non-transfected cells, as well as membrane sharing (Supplementary Fig. 4c–e and Supplementary Movie 1).

Overall, these data show a high expression and vesicular nature of FAME in the mouse kidney, suggesting a link to cellular/membrane transport.

**Binding partners of FAME suggest a role in iron metabolism and cell cycle-association.** To elucidate the molecular binding partners of FAME, we performed a yeast two-hybrid screen. For this, we used two different libraries (mouse kidney embryo from E18.5 and mouse brain embryo mix of E10.5 and E12.5) to cover more possible interaction partners (Fig. 3a). Both screens identified Ferritin Heavy Chain 1 (FTH1), Formin Binding Protein 1 Like (FNBP1L), as well as Tousled Like Kinase 1 (TLK1) as top hits (Fig. 3b). *Fth1* encodes the heavy subunit of ferritin, the major intracellular iron storage protein in prokaryotes and eukaryotes. A main function of ferritin is the storage of iron in a soluble nontoxic state and further iron uptake in capsule cells of the developing kidney[42]. FNBP1L, on the other hand, is required to coordinate membrane tubulation with the reorganization of the actin cytoskeleton during endocytosis[43,44]. TLK1 has several diverse substrates and is active when phosphorylated. Activation by phosphorylation is cell cycle-dependent with its peak activity in S-phase. Overexpression of TLK1 protein causes severe growth defects and cell cycle arrest in the G2/M phase with apoptosis[45]. Our yeast-two-hybrid screen also detected FAME interactions with transcription factors and proteins with histone H4-specific acetyltransferase activity, including CREB/ATF BZIP Transcription Factor (CREBZF) and Lysine Acetyltransferase 7 (MYST2). Both CREBZF and MYST2 are associated with cell cycle progression and replication. CREBZF arrests the growth of osteosarcoma cells by displacing MDM2 and stabilizing p53[46], whereas MYST2 has crucial functions in transcription, replication, and DNA repair[47].

However, since FAME was not located in the nucleus (Fig. 2d, e), the putative Y2H interaction partners in the nucleus (Fig. 3b) must be considered with caution. Nevertheless, we confirmed the interaction with FTH1 using mCherry-labelled FTH1 expressed in HEK293T cells cotransfected with GFP-tagged FAME (Fig. 3c).

To obtain a better picture of the interactome of FAME we performed a proximity-dependent biotin identification (Bio-ID) experiment together with classical immunoprecipitation followed by mass

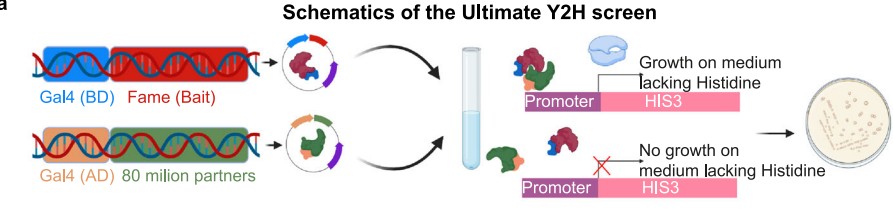

**a** Schematics of the Ultimate Y2H screen

**b**

| Global Predicted Biological Score from 118 million analysed interactions | | Kidney | Brain |
|---|---|---|---|
| A | Very high confidence in the interaction | FTH1 and FNBP1L | TLK1 |
| B | High confidence in the interaction | CREBZF | FTH1, GM3411, ZfP821 |
| C | Good confidence in the interaction | MYST2 and TLK1 | FNBP1L |

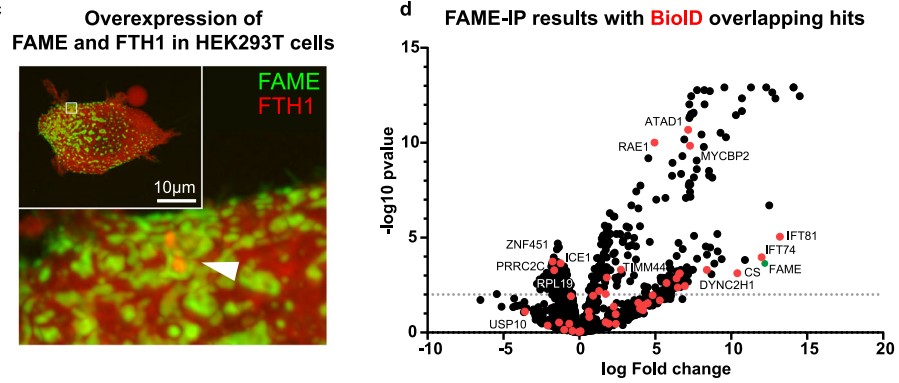

**c** Overexpression of FAME and FTH1 in HEK293T cells

**d** FAME-IP results with **BioID** overlapping hits

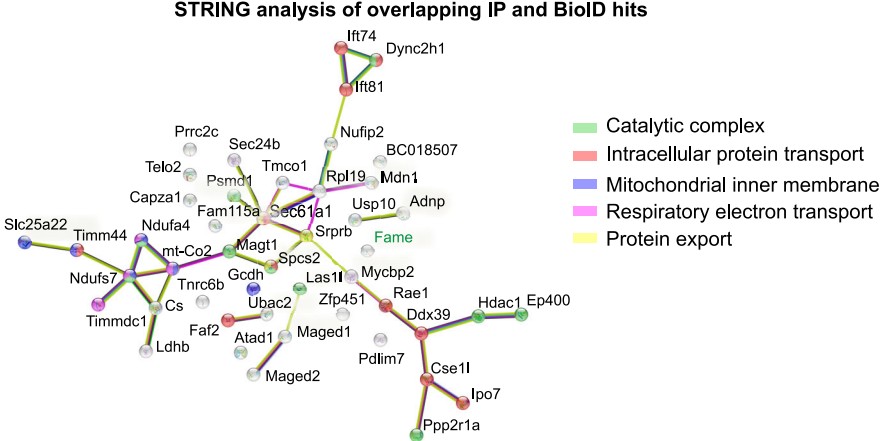

**e** STRING analysis of overlapping IP and BioID hits

Catalytic complex
Intracellular protein transport
Mitochondrial inner membrane
Respiratory electron transport
Protein export

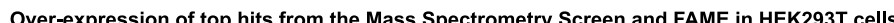

Over-expression of top hits from the Mass Spectrometry Screen and FAME in HEK293T cells

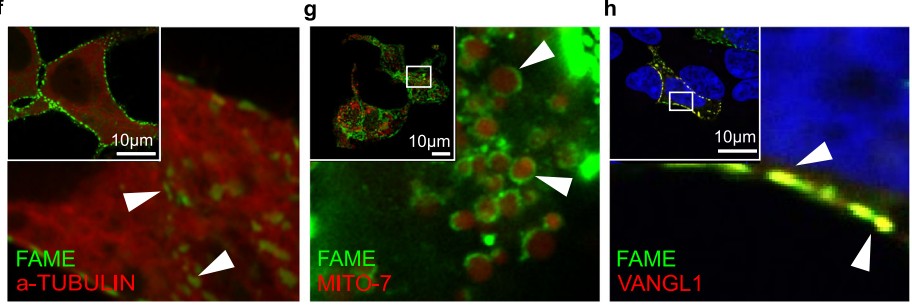

**f** FAME / α-TUBULIN

**g** FAME / MITO-7

**h** FAME / VANGL1

spectrometry (Fig. 3d and Supplementary Data 7, 8). The FAME-IP results are visualized in Fig. 3d together with overlapping hits from the BioID experiment. STRING analysis of these overlapping hits revealed a strong association of FAME with the catalytic complex, intracellular protein transport, mitochondrial inner membrane, respiratory electron transport, and protein export (Fig. 3e). These data are supported by gene correlation data from publicly available single-cell transcriptomics data of various tissues (Supplementary Fig. 7). From these positive correlations, we could show the co-localization of genes specific for the microtubule, mitochondria, and PCP-pathway association of the FAME protein (Fig. 3f–h).

**FAME controls the excretion of nutrients and iron.** To understand the functional role of FAME in general development, the morphology

**Fig. 3 | Binding partners of FAME suggest a role in iron metabolism. a** Graphical representation of the ULTImate Y2H™ screen performed by Hybrigenics. Mouse FAME bait was cloned into the pB27 (N-LexA-AKR2-C fusion) vector, and used for screening using mouse kidney embryo_RP1 and mouse embryo Brain_RP2 fragment libraries as prey. The interaction of two proteins reconstitutes an active transcription factor and enables yeast growth. **b** Top scoring interaction partners for kidney embryo and embryo brain libraries are indicated. In total, 118 million interactions were tested. **c** Validation of FTH1 as top FAME interaction partner from the ULTImate Y2H™ screen. FAME-EGFP was overexpressed together with FTH1-mCherry to visualize the heavy subunit of ferritin in HEK293T cells. The white arrow points towards encapsulated FTH1 by FAME. Representative image of 3 independent experiments. **d** Visualization of FAME interaction partners identified by both immunoprecipitation (IP)/mass spectrometry and proximity-dependent biotin identification analysis (BioID). **e** STRING analysis from all overlapping IP and BioID hits. **f–h** Validation of top FAME interaction partners from the mass spectrometry analysis. Representative image of 3 independent experiments. **f** Overexpression of FAME-EGFP together with α-Tubulin-mCherry in HEK293 cells for the visualization of microtubules. White arrows point at FAME co-localising with α-Tubulin. **g** Overexpression of FAME-EGFP together with MITO-7-mCherry to identify mitochondria. White arrows highlight the membranous localisation of FAME in mitochondria. **h** Overexpression of FAME-EGFP and VANGL1-myc visualized with an anti-VANGL1 antibody. White arrows show co-localization of both proteins within the plasma membrane.

of organs, and its physiology, we generated *Fame* knockout mice using an *FVB/Ant* genetic background. CRISPR/Cas9 was used to create these knockout mice by inserting a double STOP codon downstream of the initiator ATG (see Methods section). These mice appeared viable without major developmental, morphological, or behavioural defects. We could effectively propagate the colony in the homozygous knockout state.

Interestingly, body weight and mean lean body mass were significantly altered in *FVB/Ant Fame* knockout mice (Fig. 4a(i), ii). When analysing serum ferritin levels, we revealed a significant decrease in ferritin in knockout animals compared to the control (Fig. 4a(iii)). Furthermore, we found an unusually high amount of albumin in the urine of knockout animals (Fig. 4a(iv)). To examine this phenotype in more detail, we carefully investigated the kidneys of control and knockout animals. Analysis of the kidney volumes of adult, 12-week old mice and the amount and size of the filtering glomeruli did not show any differences (Supplementary Fig. 8a). Histological analyses of Periodic acid–Schiff (PAS) sections showed normal histomorphology in both wild type and *Fame* knockout mice, without any signs of pathological alterations of glomeruli, vessels, or the tubulointerstitium (Supplementary Fig. 8b). As consequences of kidney malfunction, particularly proteinuria, can be affected by changes only visible at the ultrastructural level, we also analysed the kidneys using transmission electron microscopy. The analysis showed an intact and normally developed filtration barrier of the glomeruli (Supplementary Fig. 8b), with regularly shaped podocyte foot processes, regular glomerular basal membrane, and thin fenestrated endothelium of glomerular capillaries. Also, the tubular cells showed a normal ultrastructural appearance with a prominent brush border in proximal tubules and high amounts of mitochondria. No signs of metabolic stress were observed, including no signs of intracellular accumulations of lipids or glycogen, increased vesicles, or high lysosomal activity.

Because adult homozygous knockout animals did not show any structural phenotype at the level of the kidney, we hypothesized that FAME might convey specific adaptive highly tunable functions and is important for the competitiveness of animals in different ecological niches. To address the role of FAME in adverse environmental conditions and to challenge the developing systems of organs, we tested the effect of a low iron diet during embryonic development of wild type and knockout mice (*FVB/Ant* background). For this, we kept females on a low-iron or control diet for four weeks before mating. New-born pups were then analysed using microCT coupled with 3D segmentation. All *Fame (FVB/Ant)* knockout mice on the control diet showed a size reduction of the kidneys, adrenal glands, interscapular brown adipose tissue, and liver compared to the wild type mice on a control diet. This phenotype was comparable with control pups being on a low-iron diet. *Fame (FVB/Ant)* knockout mice on a low-iron diet showed no further changes in organ size (Fig. 4b–f). These findings suggest that FAME is important for scaling the inner organs in response to adverse conditions, including the energy storages observed by the reduced interscapular brown adipose tissue.

To dig deeper into the function of FAME, particularly in the kidney, we performed single-cell RNA sequencing of wild type and knockout kidney samples (*FVB/Ant* background) (Fig. 5a–c and Supplementary Fig. 9). The data confirmed the presence of two consecutive stop codons in the protein-coding region of *Fame* mRNA in individual cells from the knockout condition (Supplementary Fig. 10a). As a main result, the single-cell transcriptomics analysis confirmed the presence of all cell populations in knockout kidneys, with only minor compositional changes (Fig. 5a, b). By analyzing the gene expression within clusters in more detail, we identified only a few significant differentially expressed genes (Fig. 5c and Supplementary Fig. 10b–e), including the previously identified FAME-binding, iron-transporting *Fth1* and *Slc25a39*–a mitochondrial membrane transporter involved in biosynthesis and potentially iron homeostasis[48]. This suggests that the knockout phenotype has a molecular nature without noticeable defects at tissue organization levels.

Next, to investigate if such metabolism tuning roles of FAME depend on the genetic background[49,50] and additional modifications of experimental conditions, we created and tested a second mouse knockout model using a *C57BL/6NCrl* background. Although the excretion of albumin in the urine and serum ferritin levels were not altered in these mice with knocked out *Fame*, the body weight and lean body mass were significantly changed (Supplementary Fig. 11). These two knockout models on different genetic backgrounds revealed that depending on the exact genetic background, the body weight and lean body mass show different significant alterations compared to the controls of the same background. The reason for these differences can be multifaceted and might include the complex and divergent context of differently expressed interacting molecules. For instance, the controls of different backgrounds showed differences in the excretion of albumin and serum levels at steady state (compare Fig. 4a and Supplementary Fig. 11). Furthermore, the differences in fine movement and the changed metabolic status in both animal groups can cause differences in body weight and lean body mass. Interestingly, despite the differences in weight changes in the two knockout models, the level of the energy balance influencing hormones ghrelin and leptin were not significantly changed in either mouse model (Supplementary Fig. 12). Furthermore, analysis of multiple additional blood parameters did not show significant differences, except a decreased platelet count in knockout females with an *FVB/Ant* background (Supplementary Fig. 13) and lower eosinophil number in females with a *C57BL/6NCrl* background (Supplementary Fig. 14).

**Knockout of *Fame* influences metabolic parameters and activity of the animals.** To elucidate the metabolic phenotype of the knockout animals, we performed metabolic phenotyping. Measurements were performed on 75-day old male mice with an *FVB/Ant* background, four wild type and four *Fame* knockout mice, after a three-day training phase in specific metabolic cages. For the *C57BL/6NCrl* experimental group of similar age, we had eight wild type and ten knockout males and seven wild type and five knockout females, that underwent metabolic phenotyping. Food and water intake, locomotor activity, O₂

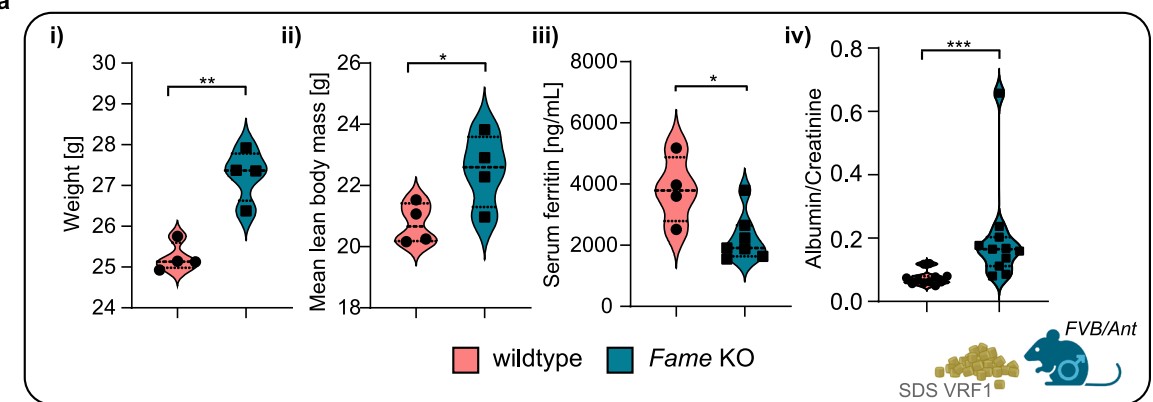

Analysis of nutrient excretion in urine and serum ferritin levels in *FVB/Ant* mice

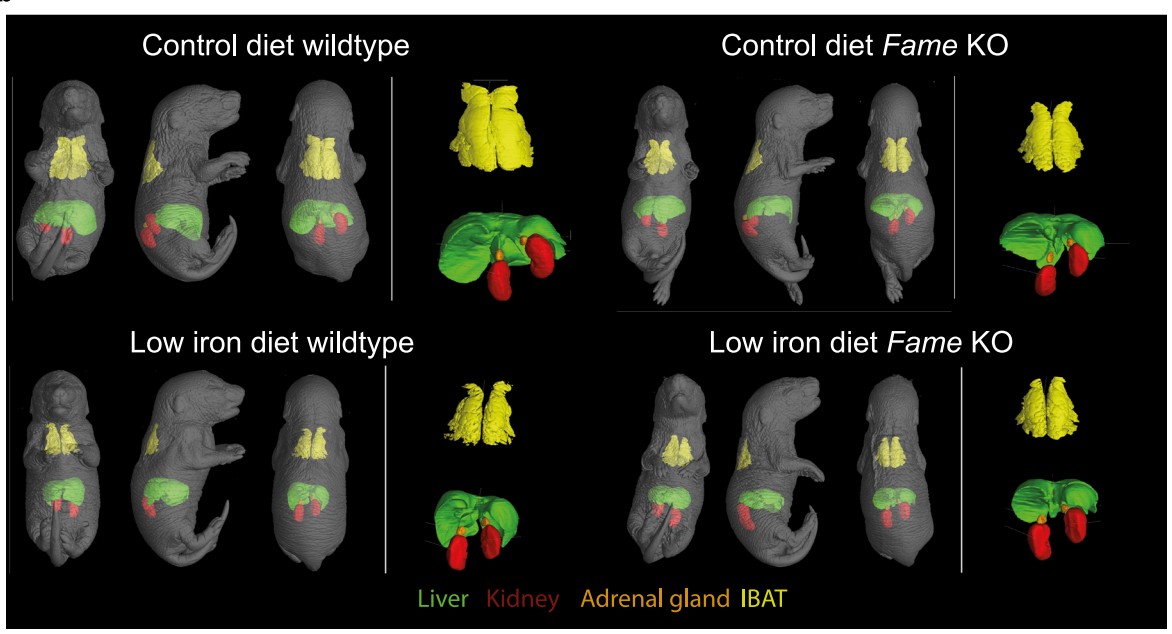

Micro-CT images and organ segmentaion of P0 pups (*FVB/Ant*) after control and low iron diet

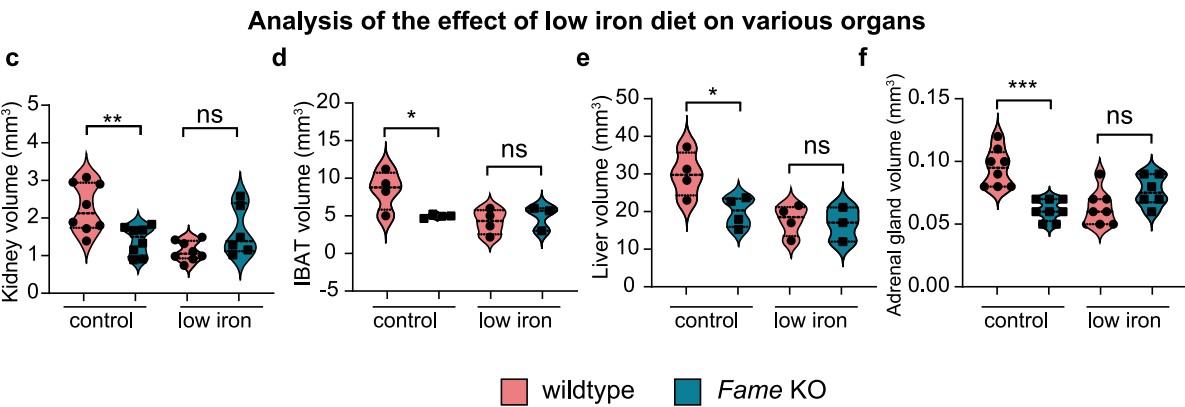

Analysis of the effect of low iron diet on various organs

consumption, and $CO_2$ production of the mice were monitored for 48 h. Before and after the measurement, body composition analysis of the mice was carried out by EchoMRI (Fig. 6a). As stated above, knockout animals with *FVB/Ant* genetic backgrounds showed a significantly higher average body weight as compared to the wild types (Fig. 4a). Interestingly, the effect was reversed in the *C57BL/6NCrl* background group (Supplementary Fig. 11). These differences in body

weight suggest that *Fame* deletion alters energy homeostasis in a way depending on different genetic backgrounds.

When we investigated food intake and normalized it to total body weight, only the food intake of the knockout mice with *FVB/Ant* background was significantly lower during the daytime (Fig. 6b).

In addition, the energy expenditure of the *FVB/Ant Fame* knockout animals normalized for total body weight (Fig. 6b), as well as Z-activity,

**Fig. 4 | *Fame* is involved in the excretion of proteins and participates in the scaling of inner organs under adverse conditions such as nutrient deficiency. a** (i) Violin plots showing the differences in weight between male wildtype and *Fame* knockout (KO) animals on an *FVB/Ant* background. Mean ± SEM and *n* for each group: WT (27.26 ± 0.3204 *n* = 4), KO (25.24 ± 0.1775 *n* = 4), *p*-value WT vs KO = 0.0015. ii Comparison of mean lean body mass. Mean ± SEM and *n*: WT (20.75 ± 0.3303 *n* = 4) KO (22.50 ± 0.5986 *n* = 4), *p*-value WT vs KO = 0.0436. (iii) Comparison of serum ferritin levels. Mean ± SEM and *n*: WT (3817 ± 548.6 *n* = 4) KO (2238 ± 295.5 *n* = 7), *p*-value WT vs KO = 0.0206. (iv) Comparison of urine albumin to creatinine ratio. Mean ± SEM and *n*: WT (0.07496 ± 0.007053 *n* = 11) KO (0.1983 ± 0.04809 *n* = 11), *p*-value Mann-Whitney test WT vs KO < 0.0001. **b** Micro-CT images of P0 pups with control and low iron diet, containing 178.58 mg iron/kg or 5.16 mg iron/kg, respectively. The kidney (red), interscapular brown adipose tissue (IBAT) (yellow), liver (green) and adrenal glands (orange) are segmented using 3D Visualization software and superimposed onto the pups. **c**–**f** Violin plots comparing inner organ scaling amongst P0 wildtype and knockout pups on different diets. **c** Kidney volume. Mean ± SEM and *n*: WT control (2.261 ± 0.2301, *n* = 8), KO control (1.401 ± 0.1354, *n* = 8), *p*-value WT control vs KO control = 0.0062, WT low iron (1.120 ± 0.0973, *n* = 8), KO low iron (1.642 ± 0.2669, *n* = 6). **d** IBAT volume. Mean ± SEM and *n*: WT control (8.44 ± 1.299, *n* = 4), KO control (4.955 ± 0.1169, *n* = 4), *p*-value WT control vs KO control = 0.037, WT low iron (4.210 ± 0.8292, *n* = 4), KO low iron (4.897 ± 0.9432, *n* = 3). **e** Liver volume. Mean ± SEM and *n*: WT control (29.94 ± 2.959, *n* = 4), KO control (19.90 ± 1.988, *n* = 4), *p*-value WT control vs KO control = 0.0304, WT low iron (17.72 ± 2.049, *n* = 4), KO low iron (16.72 ± 2.632, *n* = 3). **f** Adrenal gland volume. Mean ± SEM and *n*: WT control (0.095 ± 0.005345, *n* = 8), KO control (0.06125 ± 0.002950, *n* = 8), *p*-value WT control vs KO control < 0.0001, WT low iron (0.06250 ± 0.00491, *n* = 8), KO low iron (0.07667 ± 0.004944, *n* = 6). Source data are provided as a Source Data file.

a measure for exploratory behaviour, of *C57BL/6NCrl* knockout animals differed significantly compared to wild type mice of their corresponding genetic background (Fig. 6c and Supplementary Fig. 15).

Of note, we detected a noticeable tendency for changes in fine activity in the *FVB/Ant* background mice: the knockout animals showed fewer fine movements at night, and their average fine movements were fewer as compared to the wild types (Fig. 6b).

Since both mouse models exhibited slightly different metabolic profiles, we challenged them using different approaches to further validate the previously observed phenotypes. To exclude the effect of food on energy expenditure and the potential impact of the differing daytime food intake between the genotypes on a *FVB/Ant* background, we repeated the experiment upon fasting. For this, we removed the food from the cage, and started the measurement after an 8-hour-long fasting period without the further addition of food. During this setup, the same parameters as in the experiments before were monitored for 24 h.

Similar to the experiment before, we observed differences between the two groups of *FVB/Ant* mice in activity at the beginning of the night-time. Both fine and Z-activity differed significantly early during the 12-hour dark period (Supplementary Fig. 16a–d). In line with this, the energy expenditure of the *FVB/Ant* knockout mice was significantly less between 7 and 9 PM as compared to controls of the same background (Supplementary Fig. 16e, f). These data indicate that the food-seeking activity of the *FVB/Ant* knockout animals was less pronounced compared to wild types. These time points early during the dark period overlap with the intensive food-seeking activity of *FVB/Ant* wild type mice, indicating that the significantly higher energy expenditure of the *FVB/Ant* wild type animals is due to the higher locomotor activity. On the other hand, it is worth mentioning that the energy expenditure of the *FVB/Ant* knockout mice stayed below the energy expenditure of *FVB/Ant* wild type mice during the whole measurement. Overall, *FVB/Ant* knockout mice had higher body weight with higher lean body mass. Furthermore, *FVB/Ant* knockout mice seemed to be less hungry after 8-hour fasting periods resulting in less intense food-seeking behavior. The energy expenditure of the *FVB/Ant* knockout mice was significantly lower due to their decreased locomotor activity.

Next, we challenged *C57BL/6NCrl* mice by exposing them to a first warm and then cold environment to study effects on energy expenditure of knockout animals. While effects on energy expenditure were small, we did find that depending on the sex of the animals, different measured parameters, such as Z-activity, food intake and to a low degree energy expenditure, were altered differently between females (Supplementary Fig. 17) and males (Supplementary Fig. 18). This suggests that FAME has a sex-specific role.

These results support the evolutionary diverging and plastic role of FAME in tuning the energy expenditure balance in various animal groups and different genetic backgrounds.

In humans, the analysis of genome-wide association studies (GWAS) showed a correlation of mutations in *FAME* with higher body mass index and diabetes-related pathologies as well as macular degeneration (Supplementary Fig. 19). These results are in line with our previous findings, fitting that iron homeostasis and age are factors related to macular degeneration[51], together with the role of iron in diabetes and body mass indexes[52].

To investigate this further, we examined the phenotypic effects of recent evolutionary changes in *CCDC198 (FAME)* on the modern human lineage. Of the three mutations derived from the modern human lineage (rs143439954, rs147526762, rs148585073) (Fig. 1f), only rs147526762 has an ancestral allele still present among Europeans (allele frequency ~0.1%). Given that the vast majority of genetic association studies have been carried out on Europeans, we explored whether rs147526762 had any phenotypic effects relating to the phenotypes discussed above (metabolic syndromes, kidney-related disorders, and macular degeneration). We examined possible associations using PhenoScanner[53], and although no association passed correction for multiple comparisons, the hit with the lowest *p*-value was an association with high-density lipoprotein (*p* = 0.02, beta = 0.74, inverse normally transformed units) from the UK Household Longitudinal Study[54]. For this association, the ancestral allele increased the HDL levels. We also investigated any association between rs147526762 and 1,400 broad phenotypes among 400,000 Britons (using UK BioBank data and PheWeb (PheWeb, n.d.). The association with the lowest *p*-value was against hypertensive chronic kidney disease, for which the ancestral Neanderthal-like allele increases the risk (*p* = 0.04, beta = 6.6, log odds units). Although these associations did not pass correction for multiple comparisons, we note that the tentative associations match the here suggested role of *CCDC198 (FAME)*. The low allele frequency of the ancestral allele makes phenotypic analyses challenging. Future, more extensive studies, particularly among Southern Han Chinese people, where 3% carry the ancestral allele[26], are needed to corroborate these putative associations.

**Correlation of FAME with cancer.** Due to the metabolic phenotypes observed in *FAME* knockout mice, we became interested in the role of FAME in tumors, where energy expenditure and metabolism are altered. Indeed, the expression of *FAME* appeared stably maintained in all tumor types derived from healthy human *FAME*[+] tissues and cell types according to our analysis of public TCGA and GTEx datasets (Fig. 7a, b and Supplementary Fig. 20a). The survival probability of different types of cancer can be stratified by low or high *FAME* expression (Supplementary Fig. 20b). To test this in vitro, we overexpressed *FAME* in HEK293T and A549 cells (human adenocarcinoma from the lung), which led to a decrease in proliferation (Fig. 7c, d). The knockout of *FAME* in HEK293T cells by CRISPR-Cas9 genome editing did not lead to a change in proliferation (data not shown). This is likely because *Fame* is not endogenously expressed in HEK293T cells, as confirmed by qPCR (Supplementary Fig. 20c) and according to human protein atlas data. Conversely, the knockout of

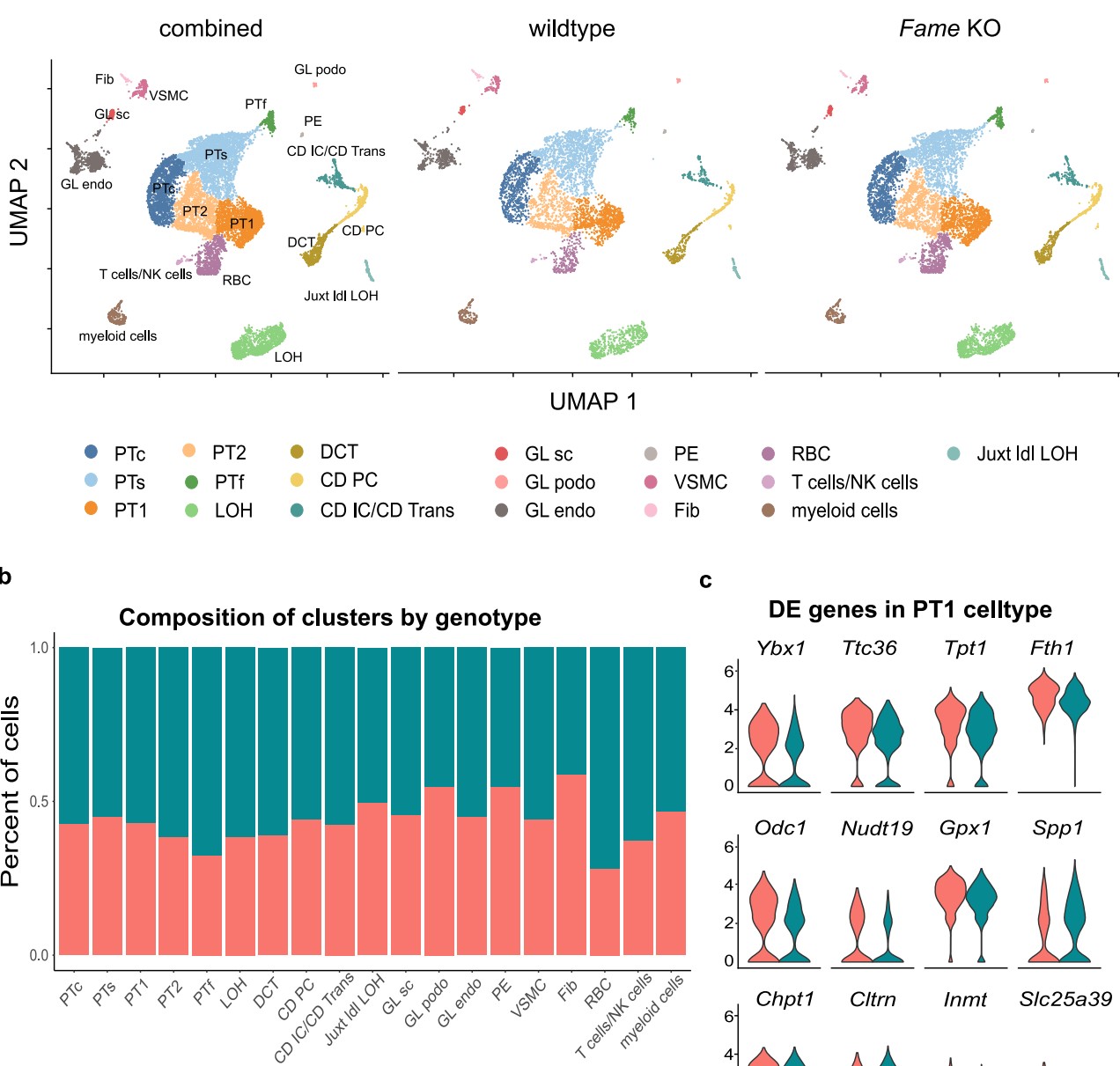

**Fig. 5 | Single-cell sequencing reveals minor molecular changes in *Fame* knockout kidneys. a** UMAP embedding of the adult mouse kidney scRNA-seq combined dataset, with the distribution of wildtype and *Fame* KO cells. Cell types are indicated. PTc proximal convoluted tubules, PTs proximal straight tubule, PTf female-specific cells, PT1 unidentified subcluster of PT, PT2 unidentified subcluster of PT, GL endo endothelial cells in glomeruli, GL podo podocytes in glomeruli, GL sc putative stem cells, LOH loop of Henle, DCT distal convoluted tubule, CD PC collecting duct principal cells, CD IC/CD Trans collecting duct intercalated cells and transitional cells, VSMC vascular smooth muscle cells, Juxt ldl LOH long descending limb of the loop of Henle in juxtamedullary nephrons, PE parietal epithelium in glomeruli, Fib fibroblasts, RBC red blood cells. **b** Stacked bar plots of combined dataset illustrating the composition of cell types by genotype. **c** Violin plots of significantly differentially expressed genes in the PT1 cluster.

*FAME* in A549 cells that express it at high endogenous levels resulted in a proliferation increase, as shown in two independent knockout cell lines (Fig. 7e, f).

To get a better picture of FAME in relation to cancer, we aimed to check the localization of the protein in different tumor cell types. For this, we used two tumor cell lines, A549 and 786-O (human renal cell adenocarcinoma) that also expresses *FAME* at high endogenous levels. FAME was highly present in these cells in membranous protrusions and appeared specifically enriched during cell division (Supplementary Fig. 21a, b). Next, the immunohistochemistry analysis on cryo-slices of human renal healthy and tumor tissues showed that FAME localized predominantly to the plasma membrane of healthy cells. In contrast, it appeared heavily internalized in malignant tissue (Supplementary Fig. 21c). This might indicate the role and trafficking activity of this protein during an altered metabolic state of tumor cells.

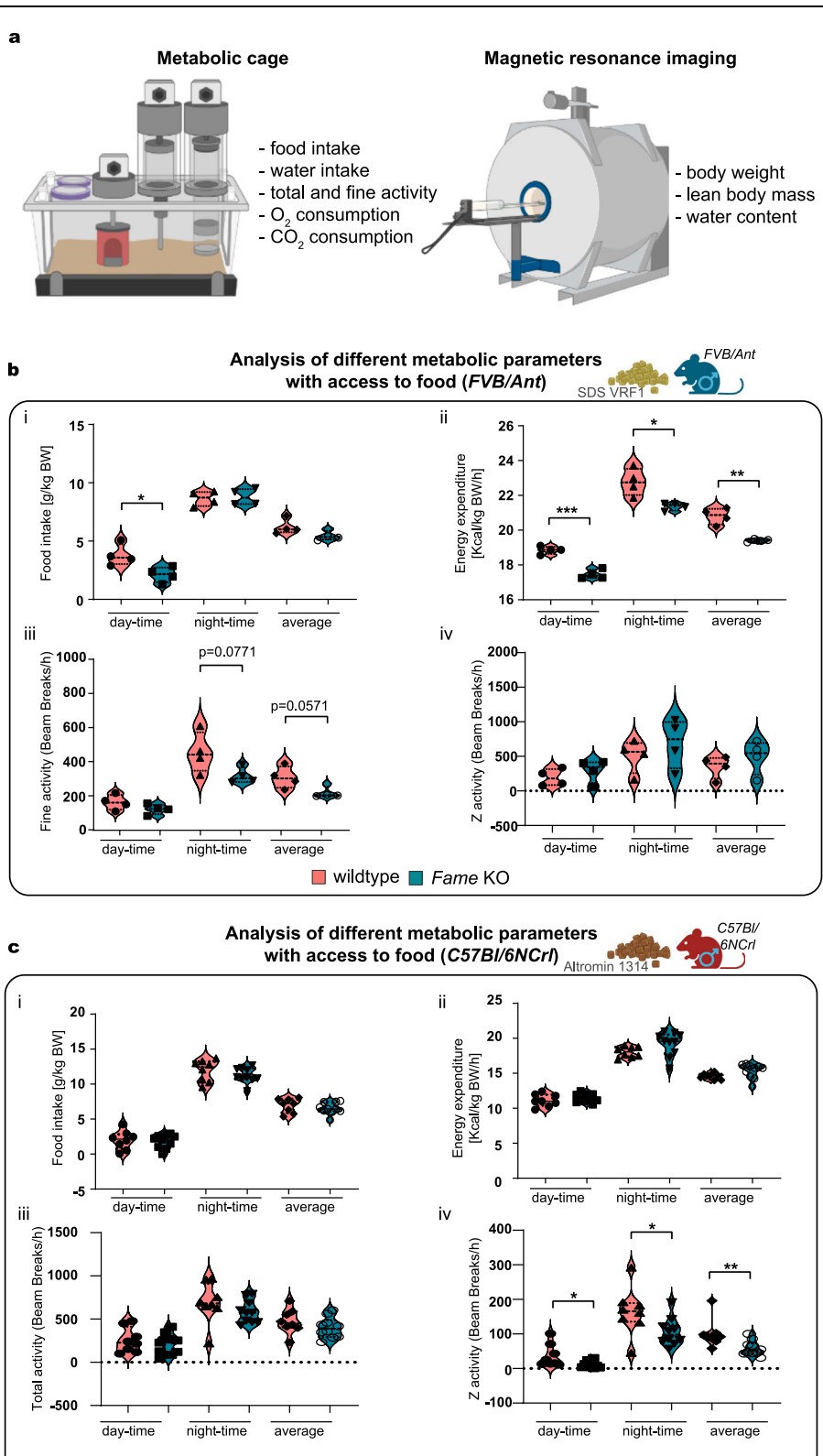

**a**

Metabolic cage

- food intake
- water intake
- total and fine activity
- $O_2$ consumption
- $CO_2$ consumption

Magnetic resonance imaging

- body weight
- lean body mass
- water content

**b** Analysis of different metabolic parameters with access to food (*FVB/Ant*)

**c** Analysis of different metabolic parameters with access to food (*C57Bl/6NCrl*)

wildtype  *Fame* KO

Comparing the survival probability of patients with kidney renal papillary cell carcinoma (KIRP) and kidney renal clear cell carcinoma (KIRC) indicated a tendency towards a higher survival of patients with higher expression levels of *FAME* with a *p*-value of 0.08 and 0.071, respectively (Supplementary Fig. 20b). Along this line we tested whether cells with lowered FAME protein levels—given its interaction with iron-storing FTH1—could be specifically targeted using Ferroptosis inducing drugs. However, this does not seem to be the case in human A549 KO cells (Supplementary Fig. 21d). Yet, our single-cell transcriptomics experiment revealed that *FAME* knockout kidneys show significant downregulation of genes associated with epithelial to mesenchymal transition (EMT) (Fig. 5c and Supplementary Data 9). Among those were: *Ybx1*[55], *Tpt1*[56], *Odc1*[57], *Gpx1*[58], and *Spp1*[59]. Correspondingly, the analysis of a public single-cell transcriptomics atlas of

**Fig. 6 | Knock out of *Fame* influences metabolic parameters and activity.**
**a** Graphical representation of the performed metabolic cage and magnetic
resonance imaging experiments. Measured parameters are listed. **b** (i-iv) Meta-
bolic cage experiments using 75-day old male mice. Wildtype and *Fame* KO ani-
mals on *FVB/Ant* background are compared. 12-hour light (day) and dark (night)
periods and 24-hour averages are shown. (i) Food intake normalized to body
weight. Mean ± SEM and *n* for each group: WT day-time (3.780 ± 0.4648 *n* = 4), KO
day-time (2.120 ± 0.3283 *n* = 4), *p*-value (day-time WT vs KO) = 0.0267, WT night-
time (8.648 ± 0.3195 *n* = 4), KO night-time (8.785 ± 0.3498 *n* = 4), WT average
(6.215 ± 0.3284 *n* = 4), KO average (5.453 ± 0.2056 *n* = 4). (ii) Energy expenditure,
normalized by body weight. Mean ± SEM and *n* for each group: WT day-time
(18.85 ± 0.1104 *n* = 4), KO day-time (17.46 ± 0.1311 *n* = 4), *p*-value (day-time WT vs
KO) = 0.0002, WT night-time (22.77 ± 0.3909 *n* = 4), KO night-time
(21.36 ± 0.1067 *n* = 4), *p*-value (night-time WT vs KO) = 0.0130, WT average
(20.81 ± 0.2344 *n* = 4), KO average (19.41 ± 0.04249 *n* = 4), *p*-value (average WT vs
KO) = 0.0011. (iii) Fine activity. Mean ± SEM and *n* for each group: WT day-time
(152.0 ± 22.28 *n* = 4), KO day-time (111.6 ± 15.09 *n* = 4), WT night-time (444.2 ± 59.6
*n* = 4), KO night-time (307.4 ± 23.88 *n* = 4), *p*-value (night-time WT vs KO) =
0.0771, WT average (298.1 ± 32.48 *n* = 4), KO average (209.5 ± 17.0 *n* = 4), *p*-value
Mann-Whitney test (average WT vs KO) = 0.0571. (iv) Z-activity. Mean ± SEM and *n*
for each group: WT day-time (193.9 ± 61.34 *n* = 4), KO day-time (291.7 ± 83.16
*n* = 4), WT night-time (504.4 ± 120.1 *n* = 4), KO night-time (691.0 ± 176.1 *n* = 4), WT

average (349.2 ± 81.28 *n* = 4), KO average (491.4 ± 122.9 *n* = 4). **c** (i-iv) Metabolic
cage experiments using 70-day old male mice. Wildtype and *Fame* KO animals on
*C57Bl/6NCrl* background are compared. 12-hour light (day) and dark (night)
periods and 24-hour averages are shown. (i) Food intake normalized to body
weight. Mean ± SEM and *n* for each group: WT day-time (1.996 ± 0.4721 *n* = 8), KO
day-time (1.834 ± 0.3098 *n* = 10), WT night-time (11.92 ± 0.5476 *n* = 8), KO night-
time (11.24 ± 0.3413 *n* = 10), WT average (6.958 ± 0.3622 *n* = 8), KO average
(6.535 ± 0.2606 *n* = 10). (ii) Energy expenditure, normalized to body weight.
Mean ± SEM and *n* for each group: WT day-time (11.13 ± 0.3114 *n* = 8), KO day-time
(11.44 ± 0.1955 *n* = 10), WT night-time (18.04 ± 0.2678 *n* = 8), KO night-time
(19.19 ± 0.5414 *n* = 10), WT average (14.59 ± 0.1513 *n* = 8), KO average
(15.31 ± 0.3281 *n* = 10). (iii) Total activity (fine movements and beam breaks).
Mean ± SEM and *n* for each group: WT day-time (256.6 ± 50.68 *n* = 8), KO day-time
(214.6 ± 37.37 *n* = 10), WT night-time (691.9 ± 82.49 *n* = 8), KO night-time
(584.6 ± 41.67 *n* = 10), WT average (474.4 ± 50.82 *n* = 8), KO average (399.5 ± 37.80
*n* = 10). (iv) Z-activity. Mean ± SEM and *n* for each group: WT day-time
(36.63 ± 11.72 *n* = 8), KO day-time (11.90 ± 3.136 *n* = 10), *p*-value (day-time WT vs
KO) = 0.0388, WT night-time (164.9 ± 24.29 *n* = 8), KO night-time (105.8 ± 12.59
*n* = 10), *p*-value (night-time WT vs KO) = 0.0359, WT average (100.8 ± 14.47 *n* = 8),
KO average (58.80 ± 6.772 *n* = 10), *p*-value Mann-Whitney test (average WT vs
KO) = 0.0059. Source data are provided as a Source Data file.

mouse neural crest development[60] showed that *FAME* is explicitly
expressed during epithelial-to-mesenchymal transition (EMT), which
we validated with immunohistochemistry (Supplementary Fig. 5e, f).
Thus, a potential role of FAME in the process of EMT may be linked to
local invasion and metastasis during cancer progression, which war-
rants further investigations.

## Discussion

The elucidations of the role of uncharacterized protein-coding genes
in the mammalian genome have been fairly uncommon in recent years.
Despite the early genome sequencing efforts that have fuelled waves of
massive characterizations[61], some proteins have avoided close atten-
tion, and 1700011H14Rik/C14orf105/CCDC198 has remained such a
"lost in a genome" exception.

During this study, we initially aimed to find proteins conveying
specific molecular adaptations behind the grand evolutionary splits
between reptiles, birds, and mammals in different environments. Using
a sophisticated comparative genomics screen, we identified a range of
such proteins, among them, we spotted 1700011H14Rik/C14orf105/
CCDC198, which we named FAME (Factor Associated with Metabolism
and Energy). This protein appeared to be one of a kind, without
additional paralogs in the genome. The phylogenetic analysis revealed
a fast evolutionary pace with specific and distinct divergence in the
corresponding gene structure in birds and mammals identified by
comparing dN/dS ratios. Furthermore, we wanted to establish if the
evolutionary divergence in specific parts within the protein structure
can be linked to the lifestyles of various mammals. This could give a
hint of the role of the protein itself. By performing such correlation
analyses, we found specific regions in this protein where evolutionary
changes coincided with land-based, water-based, desert-dwelling, or
other specific modalities of life. Altogether, these correlations sug-
gested the possible role of FAME in balancing energy expenditure and
excretion.

Furthermore, the involvement of FAME in fitness control is sup-
ported by allele flow from Neandertals into modern humans. Com-
pelled by the fact that the function of this protein has remained
unknown, we decided to proceed with molecular and functional
characterization of the protein in vivo and in vitro systems.

The analysis of expression patterns suggested a role of FAME in
inner organs, including the kidney, pancreas, fallopian tube, and liver.
The prominent expression in kidneys led us to investigate these in
more detail. Here, the knockout of *Fame* in mice with an *FVB/Ant*
background confirmed that the excretory function of kidneys was

failing, leading to a decline of ferritin levels in the blood and excessive
excretion of protein in the urine. These data related to the effects of
FAME on iron homeostasis might be important in the context of reports
that show how an iron deficiency in humans influences a variety of body
functions, such as brain activity and energy expenditure[62]. This also
connects to the fact that the regulation of iron metabolism by ferritin
sustains organismal redox homeostasis. In this way, ferritin is essential
to support organismal energy expenditure and thermogenesis[63].

Interestingly, knockout mice with a *C57BL/6NCrl* background did
not exhibit a kidney excretion phenotype. We can only speculate about
the reasons for these differences. One such reason could be the default
differences in ferritin and albumin measurements in wild type *FVB/Ant*
and *C57BL/6NCrl* mice. For instance, *FVB/Ant* wild type mice show ten
times more ferritin in serum as compared to *C57BL/6NCrl* wild type
mice. Additionally, *C57BL/6NCrl* mice excrete albumin two times as
much as compared to *FVB/Ant* mice.

On the other hand, knockout models with two different genetic
backgrounds (*FVB/Ant* and *C57BL/6NCrl*) revealed significant effects as
compared to the same background wild type controls in metabolic
cage experiments, including a different BMI of knockout animals,
variation in day/night activity, and general differences in food intake
and energy expenditure. At the same time, the adult knockout animals
stayed morphologically normal and fertile, with kidneys histologically
and anatomically indistinguishable from wild type controls. This pic-
ture fits our predictions about the possible evolutionary fine-tuning
role of FAME in the diversification and adaptation of amniotes.

To determine if adverse environmental conditions can be toler-
ated differently with and without *FAME*, we subjected pregnant *FVB/
Ant FAME* knockout females to iron deficit. This resulted in different
developing embryos compared to the wild type genotype, with altered
sizes of inner organs and interscapular brown adipose tissue. Overall,
the tuning role of FAME appears to be pleiotropic and might be
important during both embryonic development and adulthood. This
variety of molecular roles is supported by our results showing that
FAME can interact not only with proteins involved in iron homeostasis
(or regulate their expression), such as FTH1[64] and SLC25A39[48], but also
with those involved in metabolic/mitochondrial processes, signaling
molecules and transcriptional factors, as supported by IP and Bio-ID
analysis of potential interactors. Finally, since FAME is expressed in
other inner organs, such as the pancreas and liver, which play rather
dominant roles in metabolism[65], it is critical to focus on these organs
and corresponding interacting molecules in future studies in more
detail.

**FAME expression in healthy tissue and various tumor**

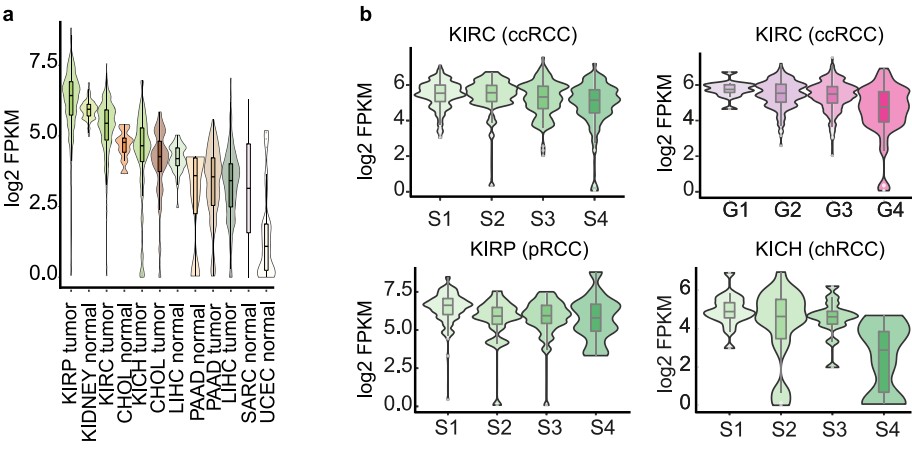

**Proliferation analyis of HEK293T and A549 cells with FAME-EGFP overexpression**

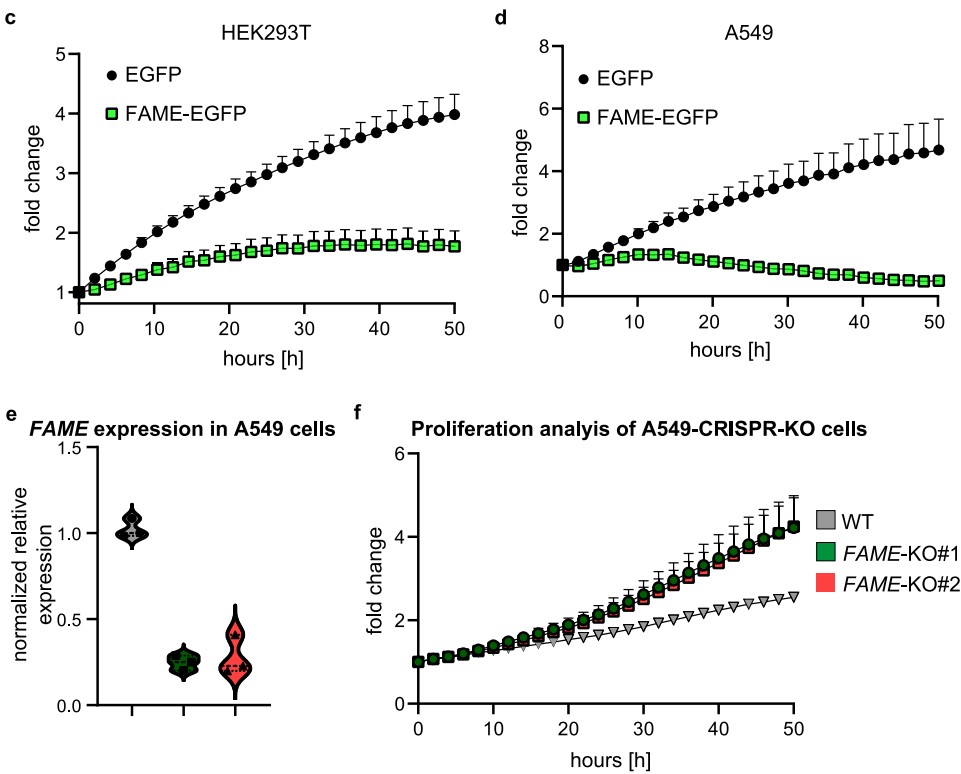

**Fig. 7 | Association of FAME with cancer. a** Analysis of *FAME* expression in various tumors and healthy tissues using RNAseq data obtained from The Cancer Genome Atlas (TGCA) Program. Mean ± SEM and *n* (tumor or healthy tissue samples): KIRP tumor (6.251 ± 0.0671 *n* = 289) kidney normal (5.916 ± 0.0301 *n* = 126) KIRC tumor (5.309 ± 0.0418 *n* = 530) CHOL normal (4.663 ± 0.1816 *n* = 9) KICH tumor (4.331 ± 0.1988 *n* = 65) CHOL tumor (4.056 ± 0.2020 *n* = 36) LIHC normal (4.199 ± 0.0688 *n* = 50) PAAD normal (2.872 ± 0.9882 *n* = 4) PAAD tumor (3.319 ± 0.0996 *n* = 178) LIHC tumor (3.255 ± 0.0623 *n* = 373) SARC normal (3.145 ± 3.145 *n* = 2) UCEC normal (1.295 ± 0.2736 *n* = 24). **b** *FAME* expression in kidney tumors (TCGA data) stratified by tumor grade (G)/stage (S). The expression is maintained in all tumor types derived from *FAME* expressing tissue. Mean ± SEM and *n* (tumor samples): KIRC S1 (5.437 ± 0.0472 *n* = 266) KIRC S2 (5.451 ± 0.1307 *n* = 57) KIRC S3 (5.236 ± 0.0894 *n* = 123) KIRC S4 (4.905 ± 0.1459 *n* = 81) KIRP S1 (6.455 ± 0.0737 *n* = 172) KIRP S2 (5.816 ± 0.3306 *n* = 22) KIRP S3 (5.811 ± 0.1692

*n* = 51) KIRP S4 (5.743 ± 0.3829 *n* = 15) KICH S1 (4.917 ± 0.1813 *n* = 20) KICH S2 (4.200 ± 0.3868 *n* = 25) KICH S3 (4.541 ± 0.2704 *n* = 14) KICH S4 (2.438 ± 0.8041 *n* = 6). **c** Proliferation analysis of HEK293T cells overexpressing FAME-EGFP or an EGFP control plasmid. EGFP⁺ cells were included in the analysis. Mean values of 5 independent cell clones per condition with positive error bars (SEM) are shown. Figure **d** as in **c** using human A549 cells. **e** Normalized RNA expression levels of *FAME* in A549 cells measured by qPCR. Technical replicates (*n* = 3) of two independent KO clones and one wildtype clone are shown. **f** Proliferation analysis of A549 clones in which *FAME* was knocked out using the CRISPR-Cas9 system. As control (WT), a FAME-sgRNA-transduced, but unedited clone in the locus of interest, was included. Mean values from 3 independent experiments with 2 replicates each per KO or WT clone (*n* = 6), with positive error bars (SEM), are shown. Source data are provided as a Source Data file. **c**, **d**, **f** Error bars are shown as positive SEM from the mean. Some bars are smaller than data symbols.

The fine-tuning and pleiotropic functions of FAME discovered in animal models suggested that FAME might be associated with human tumor development. The analysis of expression levels of *FAME* in human kidney tumors differentially correlated with patient survival trends with a *p*-value close to significance in some cases. Although this analysis of survival curves of cancer patients did not result in predictions of strong effects, our experimental analysis of FAME localization in healthy versus tumor tissues suggested the potential importance of FAME in kidney tumors. Furthermore, our experiments with knockout and overexpression of *FAME* in human cell lines (HEK293T and A549), with and without endogenous expression of *FAME,* showed a role of FAME in controlling cell proliferation. Further, they supported the potential role of FAME in cancer. However, to answer the direct question if FAME represents a cancer progression suppressor or driver (and not a passenger molecule, being only a part of the prognostic signature) in specific tumors or heterogeneous tumor cell populations, further experiments in rodents involving patient-derived xenograft models are required. Furthermore, we identified FAME as a gene associated with neural crest epithelial mesenchymal transition (EMT) and interacting with several proteins involved in the EMT process. These results are consistent with a previous report showing that metabolic activity affects the EMT of neural crest cells[66]. Additionally, according to the Yeast two-hybrid data, ferritin, the interacting partner of FAME is involved in controlling cancer cell growth[67]. Furthermore, it is interesting to note that the additional putative interaction partners revealed by our screening (TLK1, CREBZF, and MYST2) are mostly involved in cell cycle regulation. Since our functional experiments, including overexpression and knockout in two cell lines, strongly indicate that FAME plays a role in cell proliferation, we do not want to neglect the possibility of these interactions being important. However, this is subject to further experiments that are not part of this manuscript.

At last, the genetic and pharmacological experiments revealed that the membranous localization of FAME is maintained due to N-myristoylation[68]. Interestingly, the localization of FAME changed from membranous to vesicle-forming structures in the malignant tissue. This pattern of FAME redistribution during tumor formation follows a well-known phenomenon of increased membrane-associated protein trafficking in tumor growth[69].

In conclusion, FAME appears to be a non-essential gene involved in tuning the metabolism, energy expenditure, and excretion processes. Being a fine-tuning and fast-evolving factor, it influences the evolutionary diversification of amniote animal groups. It shows diverse effects depending on the animals' genetic background, metabolic needs, and sexes. As a result, the high evolutionary speed of changes observed in FAME structure and its correlation with bird speciation might be defined by specifics of the bird's metabolism, excretion, and anatomy. For instance, birds excrete uric acid (unlike mammals excreting water-soluble urea), show the presence of a renal portal vein, and do not have a bladder. Last but not least, FAME might be a potential confounder of tumor progression and human metabolic disorders, according to our experimental data and published GWAS analysis.

## Methods

### Statement on ethical considerations
All animal work was approved and permitted by the Local Ethical Committee on Animal Experiments and conducted according to the Guidelines for Animal Experimentation recommendations (ARRIVE guidelines). In particular, mouse work related to *C57BL/6NCrl* mice was approved and permitted by the Institute of Molecular Genetics of the Czech Academy of Sciences (licence: 45/2017 AVCR). Mouse work related to *FVB/Ant* by the BMBWF-V/3b (Animal experimentation and genetic engineering, Austria) (licence: BMWFW-66.009/0018-WF/V/3b/2017).

Mice were sacrificed by isoflurane overdose. In all instances, apart from blood drawing, cervical dislocation was carried out additionally.

### Antibodies
The following antibodies were used in this study:

Mouse monoclonal anti-1700011H14Rik/FAME (B-1) antibody (Santa Cruz, sc-398907, 1:50) (the limitations of this antibody are discussed below, validation is shown in Supplementary Figs. 5 and 6)

Fluorescein-labeled Lotus Tetragonolobus Lectin (Vector Laboratories, FL13212, 1:200), Mouse monoclonal VANGL1 (E-3) antibody (Santa Cruz, sc-166844, 1:200), Chicken polyclonal anti-GFP antibody (Abcam, ab13970, 1:250), Normal mouse IgG (1 µg) (12-371, Merck), Alexa Fluor 555, Donkey anti-Mouse IgG secondary antibody (Invitrogen, A-31570, 1:1000), Alexa Fluor 647, Donkey anti-Chicken IgY secondary antibody (Invitrogen, A-78952, 1:1000).

**Discussion of limitation of the anti-FAME antibody.** Given that FAME was an uncharacterized protein, we were limited in the number of commercially available antibodies. We tested several antibodies for immunohistochemistry and western blot applications. Of the 4 tested antibodies (Santa Cruz mouse monoclonal SC-398907, Santa Cruz goat polyclonal SC-245243, Invitrogen rabbit polyclonal PA5-70922 and St. John's Lab rabbit polyclonal STJ195864), only the Santa Cruz mouse monoclonal SC-398907 antibody provided consistent and specific results in immunohistochemistry on mouse wild type tissues, and did not show a similar signal in FAME knockout tissues. Notably, FAME antibody did not stain all proximal tubules in the mouse kidney sections in Fig. 2c. We were unable to determine the reason for this, but it might be due to differences in *Fame* expression levels within the tubules depending on the region of the kidney and section plane. Furthermore, while being able to detect FAME in an overexpression setting, we were not able to detect endogenous FAME protein by western blot. It is possible that the used antibody can only detect a certain threshold of FAME preferably in non-denaturing conditions, and the concentration of FAME in total kidney extract is low because the protein is produced only in few cell types associated with tubules. This suggestion corresponds well to our immunohistochemistry results on kidney sections. NOTE: This antibody has not been validated to detect the human variant of FAME.

**Comparative genomics approach.** Protein and CDS sequences from different Amniota organisms were obtained from the NCBI database. For dN/dS analysis, organisms were paired up in their classes, and in each pair bidirectional Blastp was performed. We chose pairs of mammalian and bird genomes based on the criteria that the median dS between a pair of organisms would be 0.27–0.47 to obtain a similar evolutionary distance between species and avoid redundancy and to not repeat the species. Then for each pair, we aligned proteins by using MUSCLE[70], then we aligned the CDS to the protein sequences by tranalign, and finally obtained the dN/dS ratio by using PAML. To group orthologous pairs, we also aligned each of them to Homo sapiens or Gallus gallus by bidirectional blast.

To compensate for non-ideal genomes, we used many pairs for bird, reptile, and mammal genomes. We used the Ensembl database to access gene sequence data and only kept 1-to-1 orthologues for our analysis. Of the 94 amino acid sequences we used, only 4 were fewer than 200 amino acids long and were labelled as partial. These partial sequences likely resulted from genome sequence incompleteness but should not affect our results overall.

**Expression analysis (Genotype-Tissue Expression (GTEx)/ Tabula muris)**
Gene expression data from 8555 samples from 30 different human tissues were obtained from the Genotype Tissue Expression Project (GTEx) repository (release V6p) [https://www.gtexportal.org/home/][71].

Gene transcript level expression data was calculated as reads per kilobase million (RPKM).

Gene expression data from single mouse cells were obtained from the Tabula Muris consortium[34] and [https://github.com/czbiohub/tabula-muris]. Mouse tissue scaled gene expression values, as processed by the Seurat software[72], were obtained from 55,656 cells from 12 different tissues classified into 55 different cell types.

## Microscopy

**Staining of tissue on paraffin.** Tissue samples were collected and fixed in 4% paraformaldehyde for 24 h at 4 degrees under rotation. The fixed tissue samples were then dehydrated with graded series of ethanol, cleared with xylene, and finally embedded in paraffin. The embedded tissue blocks were then sectioned into 5 μm sections using a microtome. The sections were mounted on positively charged glass slides and allowed to dry at room temperature. Antigen retrieval was performed using Dako Target Retrieval Solution, pH 9, 1x concentration. Briefly, the slides were immersed in DAKO solution and heated in a pressure cooker for 60 min. After cooling to room temperature, the slides were washed with phosphate-buffered saline (PBS) to remove any residual solution. The slides were then incubated with the primary antibody diluted in 1:50 concentration for 1 h at room temperature. After washing with PBS, the slides were incubated with the appropriate secondary antibody conjugated with a fluorochrome for 1 h at room temperature. The slides were then counterstained with DAPI to visualize the cell nuclei.

**Immunocytochemistry (ICC).** Coverslips in 24-well plates were treated with Collagen I (rat tail, 50 μg/ml, Gibco) for 1 h at room temperature. After 3 PBS washes, HEK293T cells were cultured overnight, followed by transfection of 250 ng of FAME-pEGFP-N1 using Lipofectamine LTX® and Plus™ Reagent (Invitrogen). After overnight incubation, the cells were fixed using 4% paraformaldehyde in PBS for 20 min at room temperature. After 3 PBS washes, cells were either directly stained with primary anti-1700011H14Rik/FAME antibody (Santa Cruz) and subsequent donkey anti-mouse 555 secondary antibody (Invitrogen), or treated with 1x Dako Target Retrieval Solution, pH 9 (Agient) for 40 min in a steamer at 97 °C. Next, cells were washed 3 times with PBS-T and stained sequentially with primary anti-FAME and anti-GFP (Abcam, ab13970) diluted in a mixture of 5% donkey serum, 20% dimethyl sulfoxide (DMSO) (Sigma) and 75% PBS for one hour, followed by 3 PBS-T washes. Secondary antibody staining with Donkey anti-mouse 555 (Alexa Fluor™) and donkey anti-chicken 647 (Invitrogen) was done for 1 h. Following 3 PBS-T washes, cells were stained with 4′,6-Diamidino-2-phenylindole (DAPI) at a 1:10.000 dilution and mounted with Mowiol mounting medium (Sigma). Fluorescence images were obtained by Leica DMi8 automated fluorescence microscopy.

**Staining of embryonic/tissue cryo-sections.** Embryos/tissue were harvested and fixed in 4% PFA for 4 h at 4 °C while rotating and subsequently put into a 30% sucrose (VWR, C27480) solution for 24 h. The embryos/tissue were then embedded in OCT (Sakura, 4566) and sectioned at 25 μm on a Cryostar NX70 cryostat (Thermo Fisher). For staining, the tissue was encircled with a hydrophobic pen (Agilent, S2002). Slides were washed with PBS-T and incubated with primary antibody diluted in Dako Antibody Diluent (Agilent, S3022) for 1 h at room temperature or at 4 °C overnight. Slides were washed in PBS-T and incubated with secondary antibody diluted in PBS-T for 1 h at room temperature. The tissue was washed with PBS-T, counterstained with DAPI, washed and mounted using Mowiol mounting medium (Sigma).

**Staining of tissue on paraffin.** Tissue samples were collected and fixed in 4% paraformaldehyde for 24 h at 4 degrees under rotation. The fixed tissue samples were then dehydrated with graded series of ethanol,

cleared with xylene, and finally embedded in paraffin. The embedded tissue blocks were then sectioned into 5 μm sections using a microtome. The sections were mounted on positively charged glass slides and allowed to dry at room temperature. Antigen retrieval was performed using Dako Target Retrieval Solution, pH 9, 1x concentration. Briefly, the slides were immersed in DAKO solution and heated in a pressure cooker for 60 min. After cooling to room temperature, the slides were washed with phosphate-buffered saline (PBS) to remove any residual solution. The slides were then incubated with the primary antibody diluted in 1:50 concentration for 1 h at room temperature. After washing with PBS, the slides were incubated with the appropriate secondary antibody conjugated with a fluorochrome for 1 h at room temperature. The slides were then counterstained with DAPI to visualize the cell nuclei. The stained slides were observed and images were acquired using a fluorescence microscope equipped with appropriate filters. The images were captured using a camera and processed using image analysis software.

**Confocal microscopy.** Images were acquired using a Zeiss LSM880 Airyscan confocal microscope. The 488, 555, and 647 nm Alexa fluor conjugated secondary antibodies were excited using 488, 561, and 633 nm VIS lasers, respectively. For the visualization of DAPI in cell culture experiments, a UV laser exciting at 405 nm was utilized. The corresponding filters channeling the emitted light were 493–556 nm (488 excitations), 562–624 nm (555), 638–755 nm (633), and 412–474 nm (405). The C-Apochromat 40x/1.2 W Korr FCS M27 objective was used for live-cell imaging and fixed ICC. Imaging of sections was performed at the highest available magnification with the Plan-Apochromat 63x/1.4 Oil objective.

Imaris 8.3 by Bitplane was used for image analysis.

**Live-cell imaging.** pEGFP-N1-FAME was transfected into HEK293T cells as described above and imaged using a confocal microscope. The localization and possible movement of the protein of interest was to be determined. The resolution was kept low, allowing images to be taken within milliseconds. Every second, two images were captured. Cells were tracked for 7 min, on average, until the sample was bleached.

## Cell culture

**Cell lines.** HEK293T and A549 were cultured in DMEM (Sigma, D5796), supplemented with 10% fetal bovine serum (Sigma, F7524), 1% Penicillin-Streptomycin (Sigma, P4333), and 1% L-Glutamate (Sigma, G7513). Cells were cultured in humified incubators at 37 °C and 5% atmospheric CO2. Both cell lines are classified as commonly misidentified by the ICLAC. HEK293T were purchased from ATCC (CRL-3216). A549 cells were authenticated by autosomal STR profiling by Microsynth Austria, matching 100% the DNA profile of ATCC CRM-CCL-185.

**Plasmids for overexpression experiments.** For overexpression studies 1700011H14Rik/Fame (Dharmacon, MMM1013-202768062), Vangl1-myc (Addgene), Fth1 (Biocat, BC012314-TCM1004-GVO-TRI), pPAmCherry-α-tubulin (Addgene, 31930), mCherry-Mito-7 (Addgene, 55148), were either cloned into pEGFP-N1 (Clontech) or mCherry-expressing vectors (mCherry2-C1, Addgene, 54563). Cloning was done using restriction enzymes. Restriction sites were introduced by PCR primers. For creation of a glycine to alanine (G2A) *Fame* mutant, respective changes were introduced by primers. Primers were ordered from Sigma Aldrich. Restriction enzymes and ligase were purchased from ThermoFisher. Plasmids were validated by sequencing.

**Myristylation inhibition.** Coverslips in a 24-well plate were treated with Collagen I (50 μg/ml, Gibco) for 1 hour at room temperature. After washing 3 times with PBS, HEK293T-cells were cultured overnight, followed by treatment of DMSO, 2-BP bromohexadecanoic acid

(100 μM), IMP-1088 (100 nM) or DDD85646 (1 μM) 30 min before transfection. The transfection was performed with 250 ng of RIK-pEGFP-N1 using Lipofectamine LTX® and Plus™ Reagent (Invitrogen). After overnight incubation, the cells were fixed using 4% PFA for 20 min at room temperature, washed 3 times with PBS and stained with DAPI. Imaging and counting was performed using a Thunder System (Leica). Violin charts were generated using Prism Software v 9.0 (Graphpad).

## Interaction partners

**Yeast Two-Hybrid Y2H Screening.** The ULTImate Y2H screening was performed by Hybrigenics Services (Paris, France; www.hybrigenics-services.com) following previously described methods[73,74]. The mouse 1700011H14Rik (aa 1-294) bait was PCR-amplified, sequenced, cloned in the pB27 (N-LexA-AKR2-C fusion) vector, and used for screening using mouse kidney embryo_RP1 and mouse embryo Brain_RP2 fragment libraries as prey. A total of 174 and 94 prey fragments, respectively, of the positive clones, were amplified by PCR and sequenced at their 50 and 30 junctions. The resulting sequences were used to identify the corresponding interacting proteins in the GenBank database (NCBI) using a fully automated procedure.

**Bio-ID.** The Proximity-dependent Biotin Identification (BioID) experiments were performed using the PROFACGEN Service (www.profacgen.com). The Detailed report, as well as RAW data, are available under [https://datadryad.org/stash/share/ojrXiYXvS3yzg5SZwdrXHoggtgeQBDaSgPvhRBdU8Yw].

**Immunoprecipitation (IP) and MS/MS analysis of peptides.** HEK293T cells were transfected with a plasmid encoding untagged mouse 1700011H14Rik/FAME. The day after transfection, cells were washed with PBS, scraped in ice-cold PBS, and pelleted by 200 × g, 4 °C centrifugation. Cells were lysed in 1 ml of cold lysis buffer (0.5 % NP40, 50 mM Tris buffer, pH 7.4; 300 mM NaCl; 1 mM EDTA) supplemented with protease inhibitors (Roche, 11836145001) and 0.1 mM dithiothreitol (Sigma, E3876). After 15 min, the lysate was cleared by centrifugation at 16000 × g for 15 min. 1 μg of mouse monoclonal 1700011H14Rik antibody or normal mouse IgG as negative control (Merck, 12-371) was used per sample and incubated overnight at 4 °C on a rotating wheel. 40 μl of protein G-Sepharose beads slurry (GE Healthcare, 17-0618-05) was washed in 1 ml of lysis buffer and 200 × g centrifugation. Equilibrated beads were added to the lysates with antibodies, and incubated at 4 °C on a rotating wheel for 4 h. The beads were washed 6 times in lysis buffer by centrifugation. The last two washes were done in buffer without detergent.

Following IP washes, bead bound protein complexes were digested directly on the beads by adding of 1 μg trypsin (Promega, sequencing grade) in 50 mM NaHCO₃ buffer. Beads were mixed and incubated at 37 °C with mild agitation for two hours. Beads were vortexed and removed, while the released protein complexes were further incubated at 37 °C overnight (16 h) without agitation. The resulting peptides were extracted in LC-MS vials by 2.5% formic acid (FA) in 50% acetonitrile (ACN) and 100% ACN with the addition of polyethylene glycol (PEG-20.000; final concentration 0.001%) and concentrated in a SpeedVac concentrator (ThermoFisher) to a final volume of 15 μl.

Six independent replicates were analyzed by mass spectrometry (Supplementary Figs. 23 and 24).

LC-MS/MS analyses of peptide mixtures were done using an Ultimate 3000 RSLCnano system connected to an Orbitrap Elite hybrid spectrometer (Thermo Fisher Scientific). Prior to LC separation, tryptic digests were online concentrated and desalted using a trapping column (100 μm × 30 mm) filled with 3.5-μm X-Bridge BEH 130 C18 sorbent (Waters). After washing of the trapping column with 0.1% FA, the peptides were eluted (flow rate 300 nl/min) from the trapping

column onto an analytical column (Acclaim Pepmap100 C18, 3 μm particles, 75 μm × 500 mm; Thermo Fisher Scientific) by a 100 min nonlinear gradient program (1–56% of mobile phase B; mobile phase A: 0.1% FA in water; mobile phase B: 0.1% FA in 80% ACN). Equilibration of the trapping column and analytical column was done prior to sample injection to sample loop. The analytical column outlet was directly connected to the Digital PicoView 550 (New Objective) ion source with a PicoTip emitter SilicaTip (New Objective, FS360-20-15-N-20-C12). ABIRD (Active Background Ion Reduction Device) was installed.

MS data were acquired in a data-dependent strategy selecting up to top 10 precursors based on precursor abundance in the survey scan (*m/z* 350-2000). The resolution of the survey scan was 60 000 (*m/z* 400) with a target value of $1 \times 10^6$ ions, one microscan and maximum injection time of 200 ms. HCD MS/MS spectra were acquired with a target value of 50 000 and resolution of 15 000 (*m/z* 400). The maximum injection time for MS/MS was 500 ms. Dynamic exclusion was enabled for 45 s after one MS/MS spectra acquisition and early expiration was disabled. The isolation window for MS/MS fragmentation was set to 2 *m/z*.

For data evaluation, the MaxQuant software (2.0.1.0)[75] with inbuild Andromeda search engine was used using default settings unless otherwise noted. Search was done against UniProtKB proteome database for *Homo sapiens* (downloaded from [https://ftp.uniprot.org/pub/databases/uniprot/current_release/knowledgebase/reference_proteomes/Eukaryota/UP000005640/UP000005640_9606.fasta.gz], version from 2021-06-16, 20,600 protein sequences), a separate fasta file containing the mouse C14orf105/FAME Q9CPZ1 (CN105_MOUSE) sequence and the cRAP contaminants database (112 sequences, version from 2018-11-22, downloaded from [http://www.thegpm.org/crap]. Modifications were set as follows for the database search: oxidation (M), deamidation (N, Q), and acetylation (Protein N-term) as variable modifications, with carbamidomethylation (C) as a fixed modification. Enzyme specificity was tryptic/P with two permissible miscleavages. Second peptides and match between runs (MBR) features were enabled. Only peptides and proteins with false discovery rate threshold under 0.01 were considered. Data are publicly available in the PRIDE database with the identifier PXD039259.

The proteinGroups.txt file, the resulting output from MaxQuant, was further processed in R, v. 4.1.1. using the Differential Enrichment of Proteomics Data (DEP) R package[76]. In the workflow, firstly contaminant hits were filtered out and protein intensities were log2 transformed. Only proteins with intensity > 0 in more than 4/6 samples of at least one condition were retained. Intensities were normalized using LoessF normalization, and missing values were imputed using minimal value. Finally, limma test with Benjamini-Hochberg adjustment for multiple comparison was used to test for the differentially expressed proteins. Proteins were denoted as upregulated when passing the threshold of log2 fold change > 1 and adjusted *p*-value < 0.05. Corresponding cellular localizations of upregulated proteins were visualized using the Human Cell Map resource[77] (Supplementary Fig. 23c).

## Mouse models

The original *FVB/Ant* colony was a kind gift from the University of Antwerp, Belgium. The colony was refreshed from the Jackson Laboratory [https://www.jax.org/strain/004828] several times. All experiments were performed in accordance with the Institutional Ethical Codex, Hungarian Act of Animal Care and Experimentation (2013, 40/2013) and the European Union guidelines (Directive 2010/63/EU), and with the approval of the Institutional Animal Care and Use Committee of the Institute of Experimental Medicine. Mice were maintained on a 12-hour light/dark cycle, and food and water were provided *ad libitum* behind a SPF barrier according to FELASA recommendation. All mice were healthy with no obvious behavioral phenotypes. Low iron (C1038) and control diet (C1000) were obtained from Altromin.

All *C57Bl/6NCrl* animals were bred and housed at the Czech Center for Phenogenomics, Vestec, which is accredited by the Ministry of Agriculture of the Czech Republic. Mice were housed under standard conditions (12:12 - light:dark cycle) in the individually ventilated cages (Tecniplast green line) and had free access to standard chow (Altromin 1310) and purified chlorinated water. Laboratory animal facility ventilation is set for optimal air quality (HEPA fitration) and quantity for animal and working staff comfort and stable temperature (in range 19-21 °C) and humidity (45–65%). All animal experiments were approved by the Animal Care and Use Committee of the Institute of Molecular Genetics of the Czech Academy of Sciences, Prague, in accordance with guidelines and practices established by the Directive 2010/63/EU of the European Parliament on the Protection of Animals Used for Scientific Purposes.

**Generation of *Fame* KO mice.** For *FVB/Ant* transgenic mice generation, we co-injected Cas9 protein (30 ng/μl), the gRNA (GGAACACA GGGCCAGTTGA(GGG)) (50 ng/μl), and ssODN donor (CAGTCTCGTGA ATGAGCTTTCTTCTTCCAGGTTCCGATTCAATGCAAAGAACACAGGG CCTCACTAGGGTGTCTCTGTTTCTTGGCTTTGTAAAGGTG) (15 ng/μl) into *FVB/Ant* fertilized eggs. Injected embryos were transplanted into the oviduct of B6CBAF1 pseudopregnant females. The genotyping of the new-born mice was carried out by a PCR-based strategy, where we could detect the correct modification by using specific primers for the modified sequence.

**Generation of the *Ccdc198/Fame* knockout model.** The *Ccdc198/ Fame* KO mouse (*C57BL/6NCrl*- Ccdc198 em1(IMPC)Ccpcz) was generated on a *C57BL/6NCRL* background (Charles River Laboratories) by deletion of a critical exon, specifically exon 3 of the *Ccdc198/Fame* gene (ENSMUSG00000021850) by using the CRISPR/Cas9 system. The guide RNAs (gRNAs) of highest score and specificity were designed using CRISPOR. The following guide RNAs were selected to generate the exon deletion: gRNAs, 5′- AAGGACCTGAATCTAGCACT-3′ and 5′-CATTTCCAGTACAGACTAGT-3′. The gRNAs were assembled into a ribonucleoprotein (RNP) complex with Cas9 protein (Integrated DNA Technologies, 1081058, 1072532), electroporated into 1-cell zygotes, and transferred into pseudopregnant foster females (Crl:CD1(ICR)). Putative founders were analyzed by PCR and sequencing. A founder harboring a 661 bp deletion overlapping the entire exon 3 of the *Ccdc198/Fame* gene was chosen for subsequent breeding. Genotyping was performed by PCR with the forward primer 5′- GCTGAACTGT GGAGCAGGTA-3′ and reverse primer 5′-CAATCCACCCCCAATACC CC-3′.

**Tissue contrasting and X-ray computed microtomography measurements.** To increase the contrast of soft tissues for X-ray computed microtomography (microCT), the samples were stained with 1% iodine. After embryo dissection in ice-cold PBS, the samples were fixed in 4% formaldehyde in PBS solution for 24 h at 4 °C with slow rotation. Subsequently, samples were dehydrated in incrementally increasing ethanol concentrations (30%, 50%, 70%), 1 day in each concentration. Samples were transferred into 1% iodine in 90% methanol for tissue contrasting. The iodine-methanol solution was changed after 3 days. P0 pups were stained for 7 days. The contrasting procedure was followed by rehydration of the samples by incubation in ethanol series (90%, 70%, 50%, and 30%). Then, the rehydrated embryos were embedded in 1% agarose gel (Sigma-Aldrich, A5304) and placed in polypropylene conical tubes to avoid motion artifacts during microCT scanning. The microCT measurements were conducted using the system GE phoenix v|tome|x L 240 (GE Sensing and Inspection Technologies) with a 180 kV/15 W maximum power nano focus X-ray tube and flat panel dynamic 41|100: 4000 × 4000 px, with pixel size 100 × 100 μm. The exposure time was 900 ms and 2000 projections were taken over 360°. Three projections in every position were averaged for reduction of the noise in the data. The utilized acceleration voltage and current were 60 kV and 200 μA. X-ray spectrum was filtered by 0.2 mm of aluminum plate. The voxel size of the reconstructed data was 12 μm for P0 pups, 5.8 μm for the whole kidneys, and 1.3 μm for sections of the kidneys. The tomographic reconstructions were performed using GE phoenix datos|x 2.0 3D computed tomography software (GE Sensing and Inspection Technologies). For 3D visualization, the segmentation was done by an operator using a combination of software Avizo (ThermoFisher Scientific) and VG Studio MAX (Volume Graphics).

**Urine and blood analyses.** Urine was collected for each mouse at the same time in the morning. To collect urine, the mouse was held over a Petri dish. The urine was transferred into a collection tube and stored at 4 °C or directly used for further analysis.

Blood collection was performed from the tail vein of the mice. Subsequently, the serum was isolated by centrifugation for 20 min at 2500 rpm. The serum was then transferred into a clean tube and used for immediate analysis. The following kits were used according to the manufacturer's instruction. Mouse Creatinine Assay Kit (Crystalchem, 80350), Mouse Albumin ELISA Kit (Crystalchem, 80630), Mouse Ferritin ELISA Kit (Crystalchem, 80636), Sodium Assay Kit (Crystalchem, 80179), Potassium Assay Kit (Crystalchem, 80169).

**Transmission electron microscopy.** Kidney tissue was cut in small pieces and fixed in 3% glutaraldehyde in PBS. Samples were washed in 0.1 M Soerensen's phosphate buffer (Merck), post-fixed in 1% $OsO_4$ (Roth, Karlsruhe, Germany) in 25 mM sucrose buffer (Merck) and dehydrated by ascending ethanol series (30, 50, 70, 90 and 100%) for 10 min each. The last step was repeated 3 times. Dehydrated specimens were incubated in propylene oxide (Serva) for 30 min, in a mixture of Epon resin (Serva) and propylene oxide (1:1) for 1 h, and finally in pure Epon for 1 h. Samples were embedded in pure Epon. Epon polymerization was performed at 90 °C for 2 h. Ultrathin sections (70–100 nm) were cut by ultramicrotome (Reichert Ultracut S, Leica) with a diamond knife (Leica) and picked up on Cu/Rh grids (HR23 Maxtaform, Plano). The contrast was enhanced by staining with 0.5% uranyl acetate and 1% lead citrate (both EMS). Samples were examined using a Zeiss Leo 906 transmission electron microscope (Carl Zeiss) operating at an acceleration voltage of 60 kV.

**Single-cell preparation from adult kidneys.** Control and mutant littermate mice (male and female) were used for single-cell transcriptomic and morphological analysis of the adult kidney. Mice were anesthetized with isoflurane overdose and transcardially perfused with PBS. Kidneys were dissected. One kidney per mouse was fixed in 4% PFA at 4 °C overnight to be used for morphological analysis. The second kidney was chopped in small pieces and digested with 2 mg/ml Collagenase P for 10 min at 37 °C, an equal volume of 0.10% trypsin/ EDTA was added, and tissue was digested for an additional 10 min at 37 °C. Following enzymatic digestion, tissue was triturated using a wide-bore pipette tip, and clumps were mechanically dissociated using a 100 μm mesh and a syringe plunger.

Cell suspensions were washed twice with PBS/10% FBS and resuspended at a concentration of 1,000 cells/μl for processing with a 10x controller (10x Genomics).

**Library preparation and sequencing.** Library preparation was performed using a 10x controller (10x Genomics) with the Single Cell 3′ v3 chemistry. Sequencing was performed using a HiSeq 3000 (Illumina) at the Biomedical Scientific Facility (BSF), Vienna, Austria.

Single-cell RNA sequencing of the five samples (KO female, KO male, KO mix, WT female, WT mix) resulted in 103,554,203 reads (86.30% and 67.90% of them were confidently mapped to the genome

and the transcriptome, respectively) for KO female; 58,088,304 reads (87.30% and 69.10%) for KO male; 73,512,818 (90.30% and 75.00%); 75,266,219 reads (86.40% and 65.40%) for WT female; and 58,636,386 reads (92.50% and 77.90%) for WT mix. The insert size for each sequencing was 350 bp.

To check whether DE genes between wildtype and knockout are not sex-specific, we compared them with the list of the corresponding DE genes between female and male proximal tubule (PT) samples from[78]. The genes whose adjusted *p*-values are less than 0.01 were excluded from the comparative analysis. Both lists have a little intersection; moreover, in the case of these few intersected genes, the most significant DE genes between wildtype and KO were not even in the top 40 sex-specific genes. The exceptions are Inmt (male-specific) and Spp1 (female-specific) genes. The most intriguing findings are that the FAME-binding iron-transporting *Fth1* and potentially iron homeostasis associated *Slc25a39* gene seem to be gender neutral. These Data are now outlined in Supplementary Data 10.

**Single-cell RNAseq analysis.** Raw files were processed, mapped, and counted to the Cell Ranger mm10-2020-A genome and its corresponding annotation by Cell Ranger version 4.0.0. The output count matrices for each sample were further processed with the Seurat package pipeline (v.3.2.2.9001)[79]. Each dataset was filtered to remove genes expressed in less than three cells and to remove cells that had fewer than 1000 mRNA counts per cell and more than 50 % of mitochondrial counts (accounting for the high metabolic rate observed in the kidney)[35]. Putative doublet cells were predicted by Scrublet before the filtration and log-normalization steps[80]. For further comparative analysis, five datasets were integrated into one joint dataset. To improve downstream dimensionality reduction and clustering, the mitochondrial gene content was regressed out by using the ScaleData function from the Seurat package. The first 40 principal components and 30 nearest neighbors were used for graph-based clustering and further visualization by UMAP. Each cluster was assigned a cell type based on the marker genes reported in previous studies[35,78,81–83] and Supplementary Fig. 9. To estimate the difference in genotype compositional content, the exact Fisher test was applied. Identification of differentially expressed genes in wild type and mutant genotypes in each cluster was done by the FindMarkers function that implemented Wilcoxon rank-sum test and used a cutoff for minimum log FC difference (0.25). More details can be found online https://github.com/ipoverennaya/RIK_paper.

**Metabolic cage experiments (*FVB/Ant*).** To examine the effect of the absence of *1700011H14Rik/Fame* gene on the metabolism of adult mice, metabolic phenotyping was carried out by using the PhenoMaster System (TSE Systems). Mice were placed individually into metabolic training cages to habituate to the new environment for 3 days. After that, the body composition, including total body weight, total and free water content, fat and lean body mass of the animals were determined by an EchoMRI whole-body magnetic resonance analyzer (Zinsser Analytic). During phenotyping, the water and food intake (g), the locomotor activity (counts/hour), and the calorimetric parameters, like $O_2$ consumption ($VO_2$), $CO_2$ production ($VCO_2$) of the animals were continuously monitored for 24 or 48 h. The energy expenditure (kcal/h) was calculated according to the Weir equation[84]. The resting energy expenditure was estimated from the energy expenditure data in specific time points when a mouse individually moved less than 1% of its maximum ambulatory value for the last 30 min and ate less than 0.1 g for the last 1 h[85]. The respiratory exchange ratio is the ratio of $CO_2$ produced and $O_2$ used ($VCO_2/VO_2$). At the end of the metabolic measurement, the body composition analysis was repeated. All RAW Data from these measurements can be found in Supplementary Data 11.

**Metabolic cage experiments (*C57BL/6NCrl*).** A PhenoMaster (TSE Systems) system was used for the indirect calorimetry. The software used in the PhenoMaster PC was TSE PhenoMaster v.7.1.2. Before the start of the indirect calorimetry measurements, a complete calibration protocol for the gas analyzers was performed according to the manufacturer's recommendations using normal air-compressed, $CO_2$ 1% and N2 100%. The mice were individually housed in a multiplex system with 8 cages plus a reference cage. The sampling frequency to measure the $CO_2$ and $O_2$ gas measurements were every 15 min. All measurements were initiated in the morning between 9:00 and 11:00.

We provided every cage ad libitum access to water and food, a standard chow diet (Altromin, 1314). Wooden chips bedding volume was limited to approximately 125 ml during indirect calorimetry measurements to properly detect the locomotor activity of the mice by an infrared beam break frame surrounding the cage in the horizontal plane.

The environmental conditions inside the climatic chamber were 55% relative humidity and a light cycle of 12 h of light (6:00 to 18:00) and 12 h of darkness (18: to 6:00) synchronized with the animal facility where the mice were previously housed. For the indirect calorimetry measurement, the temperature was set up as follows: at 23 °C for 48 h, and after that, we kept the mice in thermoneutrality (30 °C) for approximately 24 h, in total 36 h.

For indirect calorimetry using a cold challenge, the temperature was set up as follows: We started the indirect calorimetry measurement using 23 °C for 7 h. At 17:00 the climate chamber was warming up the environment to 30 °C for 7 h. At 00:00 the temperature started to decrease gradually to reach 4 °C. At approx. 8:00, we kept 4 °C for 4 h, and at 12:00 the temperature was increased back to 23 °C and remained at 23 °C for approx. 22 h when the calorimetry finished, after a total of 48 h. This allowed us to evaluate whether the cold challenge produced a metabolic carry over effect.

The indirect calorimetry recorded the $CO_2$ production and $O_2$ consumption. From these values, the Energy Expenditure (EE) and Respiratory exchange ratio (RER) was calculated. Moreover, measurements for locomotor activity and food and water intake were recorded every 15 min. When the indirect calorimetry protocol ended, the experiment was stopped, and the mice were weighed and placed into their original cages. All RAW Data from these measurements can be found in Supplementary Data 12.

### Expression analysis−tumor data
The Cancer Genome Atlas (TCGA) gene expression data were downloaded from the Genomic Data Commons (GDC) portal in December 2018 [https://portal.gdc.cancer.gov/]. Replicates and samples flagged by the Pan-Cancer Atlas initiative (PanCanAtlas; gdc.cancer.gov/about-data/publications/pancanatlas) were removed, yielding 9,510 tumors and 713 matched normal tissue samples from 31 different tissue sites. Gene expression values represent upper quantile normalized HTSeq-acquired Fragments Per Kilobase per Million reads mapped (FPKM) processed by the TCGA IlluminaHiSeq_RNASeqV2 platform. Pathologic tumor stage data for kidney tumors were obtained from the PanCanAtlas project site.

### Generation of *FAME* KO cell lines
CRISPR-Cas9 based gene editing was used to create *FAME* KO lines in A549 cells. Single Guide RNAs (sgRNAs) were designed using CRISPOR[86] and cloned into lentiCRISPR v2 (Addgene #52961) flowing the Zheng lab protocol[87] (sgRNA: AAGTCCACACGGCCAGCCGA). Plasmids were validated by sequencing. Lentiviral particles were produced in HEK293T cells by co-transfection of 2.4 µg of psPAX2 (Addgene #12260), 1.8 µg MD2.G (Addgene #12259) and 3.6 µg lenti-CRISPR v2-sgRNA construct using polyethyleneimine (PEI, 25 K, Polysciences) at a 1:3 DNA:PEI (1 µg/µl) ratio. Supernatants were harvested 48 h post-infection. For knocking out *FAME*, $1 \times 10^5$ cells were seeded

into 6-well plates, virus-particle containing supernatant was added, and protamine sulfate (Merck) at 8 μg/mL was added. Cells were transduced by spinfection at 800 x g for 60 min. Selection of infected cells was started 24 h post-transduction using 2 μg/mL puromycin (Cayman Chemical). Single-cell clones were obtained by limiting dilution. Genomic DNA was extracted using the Monarch gDNA Purification Kit (New England Biolabs) according to manufacturer instructions. *FAME* KO was validated by sequencing of the *FAME* locus with the sequencing primers (Microsynth): sgRNA KO locus, fwd 5′ GCCAT-GAAGGAAATGACTGCT 3′, rvs 5′ TCAAAACCACAAAGTCTGGTGC 3′. The validation of these knockouts can be found in Supplementary Fig. 22.

**Proliferation analysis of *FAME* KO cell lines.** For proliferation assays, $5 \times 10^4$ A549 or HEK293T cells were seeded into 12-well plates. The next day, cells were transfected with 1 μg of pEGFP-N1 as control or pEGFP-N1-FAME using Lipofectamine 3000 (Invitrogen) and placed into an Incucyte S3 Live-Cell Analysis Instrument (Sartorius). Images of transmitted light and green fluorescence were taken every 2 h. After ~95 h, the image acquisition was stopped. Green area confluence was analyzed using the Incucyte Base Analysis Software (Sartorius) and Prism 9 (Graphpad).

**KO validation.** RNA isolation was performed using the Nucleospin RNA II kit (Macherey-Nagel). The concentration of RNA was measured using a NanoDrop ND-1000 system (Thermo Fisher Scientific). RNA quality and integrity were assessed by the TapeStation 2200 system (Agilent Technologies). For qRT–PCR expression analysis, the RNA was reverse transcribed using the Verso cDNA Synthesis Kit (Thermo Fisher Scientific) according to the manufacturer's instructions. Quantitative expression analysis was performed using the QuantStudio 5 Real-Time PCR instrument and predesigned TaqMan gene expression assays (Thermo Fisher Scientific): C14ORF105 (Taqman probe Hs00216847_m1). GAPDH expression was used as an internal control. Relative quantification was performed according to the ΔΔCT method[88].

### Statistics and reproducibility
Statistical analysis was performed using appropriate software (Graphpad Prism 9). Descriptive statistics are displayed as mean ± standard error of the mean (SEM). All data following Gaussian distribution were analyzed by unpaired *t*-tests (two-tailed). *p*-values smaller than 0.05 were considered statistically significant (\*$p < 0.05$, \*\*$p < 0.01$, \*\*\*$p < 0.001$). All *p*-values without additional indication stem from *t*-tests. Nonparametric data were anlyzed by Mann-Whitney test and specifically indicated in the figure legends.

### Reporting summary
Further information on research design is available in the Nature Portfolio Reporting Summary linked to this article.

### Data availability
Two knockout mice strains were generated for this manuscript. Mice on a *C57BL/6NCrl* background will be available through the International Mouse Phenotyping Consortium [https://www.mousephenotype.org/data/search?term=CCDC198&type=gene] whereas *FVB/Ant* are deposited to the Jackson Laboratory (JAX Stock No. 038293). All the information necessary to reproduce the single cell analysis is presented in the Methods Section. All other relevant data supporting the key findings of this study are available within the article and its Supplementary Information files or from the corresponding author upon request. SNP data for ancestral alleles (108 Nigerian Yorubans, YRI) was taken from the 1000 Genomes Project deposited in Ensembl [https://www.ensembl.org/info/genome/variation/species/populations.html]. The Neanderthal genomes were published previously[27]: [http://ftp.eva.mpg.

de/neandertal/Chagyrskaya/VCF/][28], European Nucleotide Archive (ENA) PRJEB21157 [https://bioinf.eva.mpg.de/jbrowse][29], 2014 ENA ERP002097 [http://cdna.eva.mpg.de/neandertal/altai/]. The RAW mass spectrometry can be accessed through PRIDE with the identifier PXD039259. RAW Single cell sequencing files can be downloaded from GEO with the accession number GSE206860. RAW Yeat-Two-Hybrid data can be accessed through [https://datadryad.org/stash/share/ojrXiYXvS3yzg5SZwdrXHoggtgeQBDaSgPvhRBdU8Yw]. Protein and CDS sequences from different amniota organisms were obtained from NCBI database (Supplementary Fig. 3). The Cancer Genome Atlas Program (TCGA, December 2018, [https://portal.gdc.cancer.gov/]) and Genotype-Tissue Expression (GTEx) project datasets (release V6p) [https://www.gtexportal.org/home/] were used to collect data for Fig. 7a, b and Supplementary Fig. 4a. Tabula muris [https://tabula-muris.ds.czbiohub.org/] was used to obtain data for Fig. 2a and Supplementary Fig. 4b and downloaded from [https://figshare.com/projects/Tabula_Muris_Transcriptomic_characterization_of_20_organs_and_tissues_from_Mus_musculus_at_single_cell_resolution/27733]. GWAS[89] was used to generate data from Supplementary Fig. 19. Source data are provided with this paper.

### Code availability
All custom-made scripts used in the analysis are available at: [https://github.com/ipoverennaya/RIK_paper].

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

## Acknowledgements

We would like to thank Vendula Novosadova (Institute of Molecular Genetics of the Czech Academy of Sciences) and Roldan Medina De Guia (Institute of Molecular Genetics of the Czech Academy of Sciences) for feedback on the statistics as well as clinical chemistry. M.T., M.K., T.Z., and J.K. acknowledge CzechNanoLab Research Infrastructure supported by MEYS CR (LM2018110). M.T. acknowledges the Brno City Municipality as a Brno Ph.D. Talent Scholarship Holder and Martina Roeselova Memorial Fellowship. R.S. was supported by the Czech Academy of Sciences RVO 68378050 and by LM2018126, Czech Centre for Phenogenomics, provided by the Ministry of Education, Youth and Sports of the Czech Republic. CIISB, Instruct-CZ Centre of Instruct-ERIC EU consortium, funded by MEYS CR infrastructure project LM2023042 and European Regional Development Fund-Project „UP CIISB" (No. CZ.02.1.01/0.0/0.0/18_046/0015974), is gratefully acknowledged for the financial support of the measurements at the CEITEC Proteomics Core Facility. Computational resources for the IP LC-MS/MS data processing were provided by the e-INFRA CZ project (ID:90140), supported by the Ministry of Education, Youth and Sports of the Czech Republic. I.A. was supported by Bertil Hallsten Research Foundation, Medical University of Vienna and Gustafsson's Foundation Prize 2021, T.B. was supported by a grant from the Austrian Science Fund M2688-B28, J.K. was supported by the Grant Agency of Masaryk University (MUNI/H/1615/2018). I.P. was supported by the European Union's Horizon 2020 Research and Innovation Program under Marie Sklodowska-Curie (grant agreement No. 860635, ITN NEUcrest), P.B. was supported by the German Research Foundation (DFG, Project IDs 322900939, 454024652, 432698239 & 445703531), European Research Council (ERC Consolidator Grant No 101001791), and the Federal Ministry of Education and Research (BMBF, STOP-FSGS-01GM2202C). The work of J.K. was supported by Czech Science Foundation (22-02794 S). Parts of Figs. 1, 3, 6 and Supplementary Figs 6, 11 – 18 were created with BioRender.com. OG and RD were supported by Ministry of Science and Higher Education of the Russian Federation grant 075-15-2021-1344 and Japan Society for the Promotion of Science (JSPS) KAKENHI JP23H02226.

## Author contributions

J.Pe., L.E., A.A., I.P., R.M., T.B., M.T., R.D., A.S.S., E.E.A., D.P.R., H.Z., M.K., M.E.K., J.Kr., T.R., K.G., S.K., D.P., Z.Z., R.S.G., A.G., M.E.B., M.iK., H.A., and D.L. acquired all biological data and performed the relevant analysis. R.K. and C.K. provided human samples and gave feedback on experimental aspects. J.Pe., I.A., T.Z., F.E., Z.M., G.S., T.K., V.B., T.H., K.F., J.Ka., P.B., C.F., J.R., P.K., J.P.R., R.S. and O.G. gave feedback on experimental aspects, supervised experimental approaches, and implemented the data interpretation. J.Pe., L.E. and I.A. made all figures containing data and resulting analysis. J.Pe. and I.A. designed the study, organized the experimental work, and wrote the manuscript. All authors provided feedback on figures, manuscript composition, and structure.

## Funding

## Competing interests

The authors declare no competing interests.

## Additional information

Julian Petersen [1,25] ✉, Lukas Englmaier [2,3,25], Artem V. Artemov[4], Irina Poverennaya[4], Ruba Mahmoud[1], Thibault Bouderlique[4], Marketa Tesarova [5], Ruslan Deviatiiarov [6,7], Anett Szilvásy-Szabó[8], Evgeny E. Akkuratov[9,10], David Pajuelo Reguera [11], Hugo Zeberg [12,13], Marketa Kaucka [14], Maria Eleni Kastriti[4,13], Jan Krivanek [15], Tomasz Radaszkiewicz [16], Kristína Gömöryová [16], Sarah Knauth[1], David Potesil [17], Zbynek Zdrahal[17], Ranjani Sri Ganji[17], Anna Grabowski [4], Miriam E. Buhl [18], Tomas Zikmund [5], Michaela Kavkova [5,15], Håkan Axelson[19], David Lindgren[19], Rafael Kramann[20], Christoph Kuppe [20], Ferenc Erdélyi[21], Zoltán Máté[21], Gábor Szabó[21], Till Koehne[1], Tibor Harkany [22], Kaj Fried [13], Jozef Kaiser [5], Peter Boor [18], Csaba Fekete [8], Jan Rozman [11,23], Petr Kasparek[11], Jan Prochazka [11], Radislav Sedlacek [11], Vitezslav Bryja [16], Oleg Gusev[6,24] & Igor Adameyko [4,13] ✉

[1]Department of Orthodontics, University Leipzig Medical Center, Leipzig, Germany. [2]CeMM Research Center for Molecular Medicine of the Austrian Academy of Sciences, 1090 Vienna, Austria. [3]Ludwig Boltzmann Institute for Rare and Undiagnosed Diseases, 1090 Vienna, Austria. [4]Department of Neuroimmunology, Center for Brain Research, Medical University Vienna, Vienna, Austria. [5]Central European Institute of Technology, Brno University of Technology, Brno, Czech Republic. [6]Regulatory Genomics Research Center, Institute of Fundamental Medicine and Biology, Kazan Federal University, Kazan, Russia. [7]Endocrinology Research Center, Moscow, Russia. [8]Laboratory of Integrative Neuroendocrinology, Institute of Experimental Medicine, 1083 Budapest, Hungary. [9]Department of Applied Physics, Royal Institute of Technology, Science for Life Laboratory, 171 65, Stockholm, Sweden. [10]University of Oxford, MRC Weatherall Institute of Molecular Medicine, Radcliffe Department of Medicine, Oxford OX3 9DS, UK. [11]Institute of Molecular Genetics of the Czech Academy of Science, Czech Centre for Phenogenomics, Vestec, Czech Republic. [12]Department of Neuroscience, Karolinska Institutet, Stockholm, Sweden. [13]Department of Physiology and Pharmacology, Karolinska Institutet, Stockholm, Sweden. [14]Max Planck Institute for Evolutionary Biology, Plön 24306, Germany. [15]Department of Histology and Embryology, Faculty of Medicine, Masaryk University, Brno, Czech Republic. [16]Institute of Experimental Biology, Faculty of Science, Masaryk University, Brno, Czech Republic. [17]Central European Institute of Technology, Masaryk University, Brno, Czech Republic. [18]Institute of Pathology & Electron Microscopy Facility, RWTH Aachen University Hospital, Aachen, Germany. [19]Translational Cancer Research, Department of Laboratory Medicine, Lund University, Medicon Village, Scheelevägen 2, Lund, Sweden. [20]Institute of Experimental Medicine and Systems Biology, RWTH Aachen University, Aachen, Germany. [21]Medical Gene Technology Unit, Institute of Experimental Medicine, Budapest, Hungary. [22]Department of Molecular Neurosciences, Center for Brain Research, Medical University Vienna, Vienna, Austria. [23]Luxembourg Centre for Systems Biomedicine, University of Luxembourg, 6, avenue du Swing, 4367 Belvaux, Luxembourg. [24]Intractable Disease Research Center, Graduate School of Medicine, Juntendo University, Tokyo, Japan. [25]These authors contributed equally: Julian Petersen, Lukas Englmaier. ✉e-mail: julian.petersen@medizin.uni-leipzig.de; igor.adameyko@meduniwien.ac.at

