## [Peer Review File · Nature Communications]

A previously uncharacterized Factor Associated with Metabolism and Energy (FAME/C14ORF105/CCDC198/1700011H14Rik) is related to evolutionary adaptation, energy balance, and kidney physiologyREVIEWER COMMENTS

Reviewer #1 (Remarks to the Author):

Summary

In this manuscript the authors identify and describe an evolutionarily fast evolving protein coding gene in birds and mammals. Gene expression data suggest that it is highly expressed in kidneys, and antibody-based approaches suggest that it is localized to the plasma membrane and RAB5-positive vesicles. Using both Y2H and IP-MS, a number of putative interactors are identified. KO mice are viable, but display alterations in kidney function.

General comments

While the manuscript is overall well-written, the text would benefit by some editing by a native English speaker. The protein functional data is particularly weak.

Specific comments

1. The primary data used to argue for tissue specificity is based on mRNA (not protein) data. Of course, it's well documented at this point that mRNA levels are not actually a great predictor of steady-state protein abundance. This point would be much more convincing if the authors were to conduct some simple Western blotting and IF/IHC of multiple mouse tissues using both their WT and KO mice.
2. Reagent specificity. More importantly, much of the protein data in this manuscript appears to be based on a single commercial (i.e. probably unvalidated) antibody. The authors must therefore first convincingly demonstrate (in their hands) that this critical reagent actually detects and is (at least relatively) specific to their protein of interest. Fortunately, they have generated a great experimental control for this purpose - KO mice. Western blots and IF of multiple tissues derived from WT and KO mice would go a long way in convincing this reviewer that the data generated with this antibody is credible.
3. Co-localization data with "interactors" is not convincing. Panel g of Fig 2 displays a single, rather "sick" looking 293 cell, with a single vesicular structure showing possible co-localization. The TLK1 panel in Fig 3b shows a single green/yellow dot swimming in a "sea" of nuclear red signal. Neither of these panels is at all convincing. This type of work is typically conducted by observing hundreds of cells over multiple experiments, and reporting both the number of cells in which co-localization is observed and calculating the average degree of co-localization per cell. This is simply not credible data at present.
4. The authors have not credibly "identified" a myristoylation site here. Altering a single amino acid residue and reporting protein mislocalization is a good start, but does not convince. In order to suggest that this change is due to a loss of myristoylation, a number of additional experiments should be conducted, e.g. the use of myristoylation inhibitors (a number of these have been reported in the literature) - but not treatment with inhibitors of other lipid modifications - should lead to the same phenotype. And changing other residues in the protein of interest should not.
5. The protein-protein interaction data are highly problematic. (a) The yeast two hybrid screen was repeated multiple times, which is reassuring. However, a major caveat to this approach is that Y2H "hits" are detected in a way that completely ignores in vivo localization context: i.e. proteins that would never actually be localized to the same intracellular compartment can often be identified as "interactors" by this method. The authors must critically consider this point when they discuss the putative interactions discovered using this technique. For example, the fact that several nuclear proteins are reported as "hits" by Y2H, when the authors' own IF data convincingly suggests that this protein is almost completely localized to the plasma membrane and vesicles (and in fact appears to be excluded from the nucleus) highlights this issue. (b) The IP-MS data are also completely reliant on the specificity of the KAMP antibody, and whether it actually works for IP. In addition to first addressing the specificity of this critical reagent (above), the authors must also include at least e.g. a

Western blot demonstrating that the antibody can IP e.g. their GFP-tagged KAMP protein. In addition, as above, rather than comparing the IP-MS data to another antibody, a much better approach would be to conduct IP-MS on both WT and KO mouse tissues/cells. Anything observed in the KO tissues can be considered background. (c) Where is the IP-MS data? I can't find the actual MS data, and it is not referred to in the text. There does appear to be a related list of GO categories for this dataset, but that's it? The authors must obviously include all of this primary MS data with the manuscript. (d) Any type of IP-based approach applied to a membrane protein is fraught with difficulty. This is because the IP must be conducted using buffer/detergent conditions in which protein-protein interactions are maintained, yet at the same time the membrane-associated fraction of the protein of interest must be released into the lysate for successful immunoprecipitation. This can be quite challenging to accomplish, and is thus normally conducted using a variety of different detergents at varying concentrations, to first establish successful IP conditions. The data reported here are (first assuming that the Ab is relatively specific and can successfully IP the protein of interest) likely to be highly enriched for the "non-membrane localized" subpopulation of the protein of interest. A much simpler (and better) approach for identifying membrane protein interacting partners would be to use a proximity-dependent labeling technique (e.g. BioID / APEX). (e) The authors don't appear to compare the Y2H and IP-MS datasets - what is the degree of overlap (if any)? (f) And a final small, but important point : the MS data reported here are apparently based on a single unique tryptic peptide, according to the Methods. Proteomics journals generally require at least two unique peptides of a given protein to be observed to consider it a high confidence identification, and all raw data are generally deposited in a public database.

Reviewer #2 (Remarks to the Author):

This study by Peterson et al is a tour de force achievement, going from an evolutionary-driven analysis to discovery of specialized function of a previously uncharacterized gene, KAMP, in birds and mammals. It is one of the few studies that I have seen that has taken such a story from the beginning to the end. On top of that, the work is high quality, the scientific findings are interesting, the conceptual overview has broad impact. It is one of the few studies where I have no major concerns. I think this study has the potential to become a classic. Only minor concerns are listed below.

Line 56. Abstract, change party to partly. Change high speed to high rate.

Line 115. Need to mention the overall criteria for selecting the specific bird, reptile, and mammals genomes. Genome assembly quality?

On a related note, where there any issues in genome assembly qualities that impacted finding differences in genes between species? Many of these genomes are not complete. See Rhie et al 2021 Nature on the Vertebrate Genomes Project.

Line 131. Give the URL for the Thera-base database.

Line 174. This is odd to reference a paper for an unpublished equation. In order to repeat this analysis, readers need to know the equation used.

Line 262-253. The authors state that there are no general morphology differences in the KO mice, but then later show data on weight and other features to indicate some general differences. It comes off as contradictory, without being more specific.

Line 350. I assume this is human tissue? Need to specific the species.

Line 385. I think the authors mean paralogs within the same genome, not homologs.

Line 711. More details need to be stated about the RNA sequencing output. How many reads

sequenced? What insert sizes?

Sorry for my late review

Reviewer #3 (Remarks to the Author):

In this study, Petersen et al comprehensively characterized an unannotated gene 1700011H14RIK/C14ORF105/CCDC198 (named Kamp) in various species using multiple approaches including comparative genomics, molecular cloning, animal behavior, metabolic analysis, gene knockout, and single cell RNA-seq. Although the phenotypic change associated with Kamp is not strikingly drastic, the authors revealed some interesting observations such as the association of Kamp with body weight, energy expenditure, and iron metabolism. Since my field of study is in kidney single cell biology, my comments below will mostly be limited to the single cell analysis.

1. I can't find any data to support that Kamp has been successfully knockout. Can the authors include some genotyping data? It is also strongly advised the authors to check Kamp expression from their scRNA-seq data and compare the expression between wild type and KO. This will reassure that Kamp has been knockout'ed in multiple kidney cell types.
2. For the scRNA-seq data in figure 5, is the PT1 the only cell type that expressed Kamp? The authors should plot Kamp expression in the whole scRNA-seq data and see what cell types expressed Kamp.
3. The authors should generate a plot to highlight the number of DE genes identified from each cell type by comparing wild type versus KO. This figure can support the conclusion that "This suggests that the knockout phenotype has a molecular nature without noticeable defects at tissue organization levels"
4. In Figure 5a, the authors named a cluster GL endo. That might be not accurate because from supplementary figure 7, this cluster only expresses pan-EC markers such as Pecam1 and Kdr. The authors need to show that this cluster also expressed the glomerular EC specific marker Ehd3 in order to claim that this is a GL endo. Otherwise, just simply annotate it as EC.-
5. From Figure 5c, it is very interesting to note that Tpt1 is down regulated after Kamp is KO. Tpt1 is a tumor gene that regulates cell growth and proliferation. Does this data link to one of the conclusions the authors made about the correlation of Kamp with cancer? When the authors overexpressed Kamp in HEK293 cells (Figure2d-g), any proliferation phenotype was observed?
6. The data will be more convincing if the authors can choose some of the genes from scRNA-seq, and perform immunohistochemistry to validate. I would suggest the authors to choose some DE genes related to proliferation, metal metabolism, or fatty acid metabolism for validation since those genes, if validated, can add more evidence to support the role of Kamp in cancer, energy consumption, and metal homeostasis.
7. The authors included both male and female mice in the single cell experiment, can the authors specify how many male and female mice are used in the wild type and KO group respectively? I ask this because a previous study has shown that there is strong sex dimorphism observed in the proximal tubule (PMID: 31689386). That may potentially affect the data interpretation in the PT data. For example, the differential genes shown in Figure 5c can also be sex specific genes.
8. Supplemental Figure 8a-e, are the data about comparing the low iron diet in KO versus wildtype significant? The authors should show the p-value in the figure legends.

Reviewer #4 (Remarks to the Author):

In the manuscript entitled "A previously uncharacterized Kidney-Associated Membrane Protein (KAMP) is associated with evolutionary adaptation, energy balance and kidney physiology", Petersen et al. have shown the importance of Kamp (Kidney-associated membrane protein) in evolutionary adaptation and kidney homeostasis. They have indicated that Kamp is involved in energy expenditure and scaling of organ with the experiments using Kamp knockout mice, especially in iron-deficit condition. Besides, they have suggested the possibility that Kamp plays an important role in cancer proliferation and invasion.

Overall, the manuscript contains some novel information. The idea that Kamp is a crucial role in kidney physiology is attractive. However, there are several important concerns that should be addressed adequately.

Major concerns:

1. Albuminuria is significantly augmented in KAMP KO mice, KAMP is highly expressed at tubules in the kidney. Do the authors think that the main reason for albuminuria is an insufficient reabsorption of albumin in tubules? It should be very important to confirm the origin of albuminuria. In this regard, histological analysis (especially electronic microscope analysis) of the glomerulus is mandatory to exclude the possibility that glomerulus is the cause. Then, functional roles of the specific molecules that reabsorb proteinuria at proximal tubules (megalin, etc.) needs to be investigated.
2. In related to the point above, main site of KAMP expression is tubules. However, I cannot understand why the authors evaluated the glomerulus size but not morphological or functional changes in tubules. In addition, I believe that the authors cannot conclude that "these results suggest insufficient kidney function and re-absorption defects" from the data in this manuscript.
3. As authors noted, ferritin is the major intracellular iron storage protein, and ferritin heavy chain 1 and KAMP have intracellular interaction. Besides, the authors showed that genes associated with iron transport such as Fth1 and Slc25a39 are not much altered in KO mice. I cannot understand why this leads to lower serum ferritin observed in KAMP KO mice. Serum Fe, hemoglobin and hematocrit level should be evaluated.
4. It is not clear why the mice born from the KAMP KO female mice showed organomegaly. Since it is not clear whether the changes in iron metabolisms or anemia itself affect organ sizes, it should be investigated whether effect of anemia other than iron-deficit could cause similar changes.
5. They showed that overexpression of Kamp in HEK293 cells decreased cell proliferation in vitro. It should be determined whether Kamp deletion actually induces cell proliferation. In addition, cell cycle analysis should be evaluated. It should also be important to determine whether KAMP is associated with ferroptosis.
6. It is not clear whether expression of Kamp is associated with survival of cancer patients. If they wish to show the association between Kamp and cancer prognosis, it is desirable to show whether Kamp deletion or overexpression affects cancer prognosis in rodents.

Minor concerns:

1. They showed that KAMP is highly expressed in loop of Henle and collecting duct cell, but also in the proximal tubule. Do they express in lumen side or basolateral side? Please also discuss why KAMP KO does not result in much difference in electrolyte excretion.
2. It seems that there is heterogeneity in KAMP staining in kidney in that KAMP is not stained in some tubules. The authors should show better pictures and the reason for this should be discussed.
3. The meaning of Figure 4e is not clear. Please be clear about what you want to show with these data.
4. Please discuss the possibility that increases in body weight and lean body mass might be attributable to less fine movements or the changes in metabolic status.
5. Please discuss why water content is higher in KO group?

6. They indicated that food seeking activity of KO animals was less pronounced. I cannot understand why the metabolic changes caused by KAMP deletion can lead to reduced food seeking activity. Is there any possibility that appetite itself is altered due to the changes in brain or appetite-related hormones such as ghrelin or leptin? If you have plasma or serum samples, please measure these parameters.

7. The authors have indicated that birds showed higher average of Kamp dN/dS than in reptiles. It seems that birds exhibit rather higher number of dN/dS than mammals. It should be discussed whether Kamp plays an important role also in birds, and the difference in significance of Kamp between mammals and birds. Indeed, the kidney of mammals and birds is different in that birds have renal portal vein and do not have bladder.

8. The authors have indicated that pancreas or liver are also the organs highly express KAMP. It is rather understandable that the changes in these organs induce phenotype changes in KAMP KO animals since pancreas and liver play rather dominant roles in metabolism. Please discuss these possibilities.

REVIEWER COMMENTS

- We thank all four reviewers for their thorough review of our manuscript. We appreciated their comments and suggestions and have substantially improved the manuscript accordingly.
- **DISCLAIMER: We renamed the KAMP protein/gene in the resubmitted version to FAME (Factor Associated with Metabolism and Energy)**, as it became evident that the investigated gene is expressed beyond the kidney and plays a role in a broad range of energy processing and metabolic processes.
- Among the major improvements during this revision, we have validated our mouse-monoclonal antibody and confirmed that it detects the mouse variant of FAME (former KAMP). Next, we have performed new experiments that strongly support our initial findings that FAME (former KAMP) possesses a myristoylation site. Furthermore, combining IP and new Bio-ID experiments, we found more support for the involvement of FAME (former KAMP) in metabolic and protein export pathways, as our knockout mouse model suggests. Finally, we cross-validated and extended the FAME phenotype in our second new KO model on a different *C57Bl/6NCrl* background for this revision.

Below we are responding to each comment one by one:

Reviewer #1 (Remarks to the Author):

Summary

In this manuscript the authors identify and describe an evolutionarily fast evolving protein coding gene in birds and mammals. Gene expression data suggest that it is highly expressed in kidneys, and antibody-based approaches suggest that it is localized to the plasma membrane and RAB5-positive vesicles. Using both Y2H and IP-MS, a number of putative interactors are identified. KO mice are viable, but display alterations in kidney function.

General comments

- **DISCLAIMER: We renamed the KAMP protein/gene in the resubmitted version to FAME (Factor Associated with Metabolism and Energy)**

While the manuscript is overall well-written, the text would benefit by some editing by a native English speaker. The protein functional data is particularly weak.

- We thank the Reviewer for all the important comments. A native English speaker has now checked the manuscript. In addition, we performed additional functional experiments, including knockout in two cell lines and another transgenic animal model, to improve our understanding of the protein function.

As stated

Specific comments

1. The primary data used to argue for tissue specificity is based on mRNA (not protein) data. Of course, it's well documented at this point that mRNA levels are not actually a great predictor of steady-state protein abundance. This point would be much more convincing if the authors were to conduct some simple Western blotting and IF/IHC of multiple mouse tissues using both their WT and KO mice.

- We thank the Reviewer for this critical comment and have performed several experiments accordingly to provide evidence that our monoclonal antibody is working. We want to point out that we have tried four commercially available antibodies (STJ195864_Stjohnslabs, PA5-70921_ThermoFisher, SantaCruz-sc514267, and SantaCruz-sc-398907) and can reliably say that SantaCruz-sc-398907 was the only one detecting FAME (former KAMP). The results confirming that our antibody detects FAME (former KAMP) are presented in **Supplementary Figures 21 and Supp. Figure 23**. However, despite a great effort, we could not detect endogenous protein on Western Blot (we tried multiple protocols for extraction, buffers, etc). We assume this is due to the protein's low abundance and strong membrane-associated nature. Even in the kidney, the mRNA is reported to be expressed at reasonably low level at 60.1 transcripts per million (TPM), according to the Human Protein Atlas, and 97.3 TPM, according to the Genotype-Tissue Expression Project. Nevertheless, IHC supports the fact that FAME (former KAMP) is indeed expressed in kidneys and liver at the protein level, as the KO does not show any specific staining when compared to wildtype sections with specific membranal structures stained (**new Supplementary Figure 21**).

2. Reagent specificity. More importantly, much of the protein data in this manuscript appears to be based on a single commercial (i.e. probably unvalidated) antibody. The authors must therefore first convincingly demonstrate (in their hands) that this critical reagent actually detects and is (at least relatively) specific to their protein of interest. Fortunately, they have generated a great experimental control for this purpose - KO mice. Western blots and IF of multiple tissues derived from WT and KO mice would go a long way in convincing this Reviewer that the data generated with this antibody is credible.

- We fully agree with the Reviewer and thank for this valuable comment. For this purpose, we confirmed the antibody specificity by various approaches. At first, we validated the antibody using tagged and untagged FAME (former KAMP) in HEK293 cells using western blot; then, we performed staining on the kidney tissue section on WT and KO mice (**new supplementary Figure 21**). In addition, we have overexpressed GFP-tagged FAME (former KAMP) in HEK293 cells and stained the cells with the antibody (**Supplementary Figure 21**). Finally, we stained KO and WT kidneys and found a specific signal in WT membranal structures in kidney cells versus no specific signal in the KO condition – please see new **Supplementary Figure 21**.

3. Co-localization data with "interactors" is not convincing. Panel g of Fig 2 displays a single, rather "sick" looking 293 cell, with a single vesicular structure showing possible co-localization. The TLK1 panel in Fig 3b shows a single green/yellow dot swimming in a "sea" of nuclear red

signal. Neither of these panels is at all convincing. This type of work is typically conducted by observing hundreds of cells over multiple experiments, and reporting both the number of cells in which co-localization is observed and calculating the average degree of co-localization per cell. This is simply not credible data at present.

- The Reviewer has a valid point (thanks for this comment!), and we decided to remove this data from the present manuscript. Even though co-localization experiments are important and can further shed light on the protein's function, we feel that this data needs a separate manuscript as the current version focuses more on the general role of this protein.

4. The authors have not credibly "identified" a myristoylation site here. Altering a single amino acid residue and reporting protein mislocalization is a good start, but does not convince. In order to suggest that this change is due to a loss of myristoylation, a number of additional experiments should be conducted, e.g. the use of myristoylation inhibitors (a number of these have been reported in the literature) - but not treatment with inhibitors of other lipid modifications - should lead to the same phenotype. And changing other residues in the protein of interest should not.

- We thank the Reviewer for this comment and have performed the following experiment to address this concern. We have inhibited the myristoylation using IMP-1088 (enzyme inhibitor of the human N-myristoyltransferases NMT1 and NMT2), DDD85646 (inhibitor of *T. brucei* N-myristoyltransferase (TbNMT)) and 2-Bromohexadecanoic acid as a negative control (non-selective inhibitor of lipid metabolism). The results are present in **Figure 2** and support that FAME (former KAMP) possesses a myristoylation site.

5. The protein-protein interaction data are highly problematic. (a) The yeast two hybrid screen was repeated multiple times, which is reassuring. However, a major caveat to this approach is that Y2H "hits" are detected in a way that completely ignores in vivo localization context: i.e. proteins that would never actually be localized to the same intracellular compartment can often be identified as "interactors" by this method. The authors must critically consider this point when they discuss the putative interactions discovered using this technique. For example, the fact that several nuclear proteins are reported as "hits" by Y2H, when the authors' own IF data convincingly suggests that this protein is almost completely localized to the plasma membrane and vesicles (and in fact appears to be excluded from the nucleus) highlights this issue. (b) The IP-MS data are also completely reliant on the specificity of the KAMP antibody, and whether it actually works for IP. In addition to first addressing the specificity of this critical reagent (above), the authors must also include at least e.g. a Western blot demonstrating that the antibody can IP e.g. their GFP-tagged KAMP protein.

- We are very thankful for this critical comment and agree with the Reviewer regarding the putative interaction we discovered using the Yeast Two Hybrid system. Therefore, we have critically discussed this situation in the revised manuscript, significantly

improved our IP-MS Data, and provided evidence that the antibody is working (**Supplementary Figures 21 and 23**).

- *"However, since FAME was not located in the nucleus (Figures 2d and e), we must consider the putative Y2H interaction partners in the nucleus (Figure 3b) with caution."*
- *" Furthermore, it is interesting to note that the additional putative interaction partners revealed by our Yeast two-hybrid screening (TLK1, CREBZF, and MYST2) are mostly involved in cell cycle regulation. Since our functional experiments, including overexpression and knockout in two cell lines, strongly indicate that FAME plays a role in cell proliferation, we do not want to neglect the possibility of these interactions. However, this is subject to further experiments that are not part of this manuscript. "*

In addition, as above, rather than comparing the IP-MS data to another antibody, a much better approach would be to conduct IP-MS on both WT and KO mouse tissues/cells.

Anything observed in the KO tissues can be considered background.

- This approach would indeed be a great way to conduct the IP-MS experiment. However, since we could not detect the endogenous protein using a Western Blot approach, we could not perform this type of experiment. Nevertheless, we have now significantly improved our IP-MS Data. In total (after revision), we have completed six replicates for control and six replicates for FAME (former KAMP). We have processed the data using differential enrichment analysis of proteomics data (Zhang, Smits et al. 2018): used a contaminants filter, LoessF normalization, imputation by global minimum, and limma test with Benjamini-Hochberg correction for multiple testing. We, therefore, hope that these data now have the sufficient quality and significance to be considered by readers.

(c) Where is the IP-MS data? I can't find the actual MS data, and it is not referred to in the text. There does appear to be a related list of GO categories for this dataset, but that's it? The authors must obviously include all of this primary MS data with the manuscript.

- We apologize for not including this in our previous submission, which occurred simply because of the rushed submission. We have now deposited all RAW data, including IP-MS, Bio-ID, Metabolic cage, as well as single-cell data either as tables or into a publicly available database:

(<https://datadryad.org/stash/share/ojrXiYXvS3yZg5SZwdrXHoggtgeQBdASgPvhRBdU8Yw> and in PRIDE with the identifier PXD039259).

For the GEO with the accession number GSE206860, the secure token allowing access for the Reviewer is as follows "ajcjiwkorxittqh"

(d) Any type of IP-based approach applied to a membrane protein is fraught with difficulty. This is because the IP must be conducted using buffer/detergent conditions in which protein-protein interactions are maintained, yet at the same time the membrane-associated fraction of the protein of interest must be released into the lysate for successful immunoprecipitation.

This can be quite challenging to accomplish, and is thus normally conducted using a variety of different detergents at varying concentrations, to first establish successful IP conditions. The data reported here are (first assuming that the Ab is relatively specific and can successfully IP the protein of interest) likely to be highly enriched for the "non-membrane localized" subpopulation of the protein of interest. A much simpler (and better) approach for identifying membrane protein interacting partners would be to use a proximity-dependent labelling technique (e.g. BioID / APEX).

- We followed the Reviewer's advice and performed a Bio-ID experiment, although it was really challenging. Then, we compared the results from multiple approaches. These new data are combined in Figure 3 and support that FAME (former KAMP) has a role in tuning the metabolic processes.

(e) The authors don't appear to compare the Y2H and IP-MS datasets - what is the degree of overlap (if any)?

- We have now compared the results of our IP-MS and Bio-ID experiments in the revised manuscript (updated Figure 3). We kept the Y2H as a stand-alone, as the approach resulted in a drastically different list of interacting partners, and it ignores *in-vivo* localization context, as the Reviewer pointed out, which we carefully discuss now in the text of Results:
- *"However, since FAME was not located in the nucleus (Figures 2d and e), we must consider the putative Y2H interaction partners in the nucleus (Figure 3b) with caution."*
- However, we feel that these data still provide sufficiently important information regarding this protein's possible interactions and function. In the revised manuscript, we critically discussed the putative interactions discovered using this technique in comparison with other methods:
- *"Furthermore, it is interesting to note that the additional putative interaction partners revealed by our Yeast two-hybrid screening (TLK1, CREBZF, and MYST2) are mostly involved in cell cycle regulation. Since our functional experiments, including overexpression and knockout in two cell lines, strongly indicate that FAME plays a role in cell proliferation, we do not want to neglect the possibility of these interactions. However, this is subject to further experiments that are not part of this manuscript."*

(f) And a final small, but important point : the MS data reported here are apparently based on a single unique tryptic peptide, according to the Methods. Proteomics journals generally require at least two unique peptides of a given protein to be observed to consider it a high confidence identification, and all raw data are generally deposited in a public database.

- We updated the Methods: raw data were re-searched using MaxQuant instead of ProteomeDiscoverer, which controls the FDR both at peptide and protein level. Additionally, we employed a strict filtering of the proteinGroups.txt file, the output of

MaxQuant. The filtering was done in a two step manner: firstly, contaminants (including cRAP proteins, proteins labelled as Reverse or Only Identified By Site by MaxQuant) were filtered out; and additionally, only proteins present in $\geq 4 / 6$ replicates in at least one condition were retained, resulting in high-stringency of the dataset.

- Also, the script is available as .Rmd/.html file -- is there any github repository with the analysis of RNAseq data where it could be added?
- We also tried further filtering on ≥ 2 unique peptides resulting in 266 upregulated proteins (293 if accounting for the batch effect). If filtering on 2 unique peptides is not performed, we get 291 proteins (or 319 when accounting for the batch effect)
- We uploaded all the raw data to the following public database (PRIDE, with the dataset identifier PXD039259).

At last, we want to note that we understand that different methods resulted in different lists of potentially interacting proteins. This can be due to a multitude of legitimate and biological reasons. As we do not conclude much in terms of interpreting specific molecular interactions, which we cannot even follow within this manuscript (which is very large at this point), we would like to report these results as is, because we trust that we performed these experiments as good as we could (including new Bio-ID approach requested by reviewer), and these data might be important for future research in this field. At this point, our conclusions are rather general and based on the analysis of the roles of many interacting partners coming up in these lists (metabolism, energetics etc.). We believe this is the most accurate way of dealing with this.

We hope the reviewer will appreciate our effort to improve the manuscript during this revision by running another year of additional experiments.

Reviewer #2 (Remarks to the Author):

This study by Peterson et al, is a tour de force achievement, going from an evolutionary-driven analysis to discovery of specialized function of a previously uncharacterized gene, KAMP, in birds and mammals. It is one of the few studies that I have seen that has taken such a story from the beginning to the end. On top of that, the work is high quality, the scientific findings are interesting, the conceptual overview has broad impact. It is one of the few studies where I have no major concerns. I think this study has the potential to become a classic. Only minor concerns are listed below.

- We thank the Reviewer for this well-appreciated comment and are happy to see that the expert appreciates our work.

DISCLAIMER: We renamed the KAMP protein/gene in the resubmitted version to FAME (Factor Associated with Metabolism and Energy)

Line 56. Abstract, change party to partly. Change high speed to high rate.

- This is changed now. Thanks a lot!

Line 115. Need to mention the overall criteria for selecting the specific bird, reptile, and mammals genomes. Genome assembly quality?

- *“We chose pairs of mammalian and bird genomes based on the criteria that the median dS between a pair of organisms would be 0.27-0.47 to get obtain the a relatively same similar evolutionary distance between species and avoid redundancy and not repeat the species”*
- *“We started with 20 pairs of mammals and birds and added 8 more pairs of mammals and 7 more pairs of birds to increase diversity. We had difficulty finding pairs for reptiles because they have fewer genomes and fewer branches on their evolutionary tree, so we selected a few pairs from different clades of reptiles.”*
- *“We used the Ensembl database to access gene sequence data and only kept 1-to-1 orthologues for our analysis. Of the 94 amino acid sequences we used, only 4 were less than 200 amino acids long and were labeled as partial. These partial sequences likely resulted from genome sequence incompleteness but should not affect our results overall.”*
- This information is now included in the main text as well as part of the material and methods section in the revised manuscript.

On a related note, where there any issues in genome assembly qualities that impacted finding differences in genes between species? Many of these genomes are not complete. See Rhie et al 2021 Nature on the Vertebrate Genomes Project.

- Thank you for your comment. We want to get broad coverage of the evolutionary tree of mammals and birds that were requested to use species with not-so-high assembly qualities. However, we took only genomes with proteins annotation by NCBI that was supposed to be done in the same unbiased way. We have now added the following information into the material and methods of the revised manuscript:
- *“To compensate for non-ideal genomes, we have used many pairs for bird, reptile, and mammals genome. We used the Ensembl database to access gene sequence data and only kept 1-to-1 orthologues for our analysis . Of the 94 amino acid sequences we used, only 4 were less than 200 amino acids long and were labelled as partial. These partial sequences likely resulted from genome sequence incompleteness but should not affect our results overall.”*

Line 131. Give the URL for the Thera-base database.

- This is now included:
URL for the PanTHERIA <https://esapubs.org/archive/ecol/E090/184/metadata.htm>
 (“PanTHERIA_1-0_WR93_Aug2008.txt”)

Line 174. This is odd to reference a paper for an unpublished equation. In order to repeat this analysis, readers need to know the equation used.

- The manuscript we cite here was published in 2014. To accommodate this comment, we added the equation to the revised manuscript for the readers to know the equation used:

"We investigated if a haplotype this length could have survived since the time of the common ancestor as previously described (Huerta-Sanchez et al., 2014), i.e., using the equation $1-\text{GammaCDF}(m, \text{shape} = 2, \text{rate} = 1/L)$, where m is the measured haplotype length and L the expected length given by the equation $L=1/(r \times t)$. Here r is the recombination rate per generation per bp and t is the length of the human and Neanderthal branches since divergence"

Line 262-253. The authors state that there are no general morphology differences in the KO mice, but then later show data on weight and other features to indicate some general differences. It comes off as contradictory, without being more specific.

- Yes, we agree with the Reviewer, and we reformulated the corresponding sentences that now only refer to the observation in kidneys:

"Because adult homozygous knockout animals did not show any phenotype at the level of the kidney (Figure 4e-h), we hypothesized that FAME might convey..."

Line 350. I assume this is human tissue? Need to specific the species.

- Yes, we added this information about human tissue.

Line 385. I think the authors mean paralogs within the same genome, not homologs.

- This is corrected now. Thanks a lot!

Line 711. More details need to be stated about the RNA sequencing output. How many reads sequenced? What insert sizes?

- The following information has now been added to the revised manuscript (Methods section):

"Single-cell RNA sequencing of the five samples (KO female, KO male, KO mix, WT female, WT mix) resulted in 103,554,203 reads (86.30% and 67.90% of them were confidently mapped to the genome and to the transcriptome, respectively) for KO female; 58,088,304 reads (87.30% and 69.10%) for KO male; 73,512,818 (90.30% and 75.00%); 75,266,219 reads (86.40% and 65.40%) for WT female; and 58,636,386 reads (92.50% and 77.90%) for WT mix. The insert size for each sequencing was 350 bp. "

Sorry for my late review

- We thank the Reviewer for suggesting improvements and corrections!

Reviewer #3 (Remarks to the Author):

In this study, Petersen et al comprehensively characterized an unannotated gene 1700011H14RIK/C14ORF105/CCDC198 (named Kamp) in various species using multiple approaches including comparative genomics, molecular cloning, animal behavior, metabolic analysis, gene knockout, and single cell RNA-seq. Although the phenotypic change associated with Kamp is not strikingly drastic, the authors revealed some interesting observations such as the association of Kamp with body weight, energy expenditure, and iron metabolism. Since my field of study is in kidney single cell biology, my comments below will mostly be limited to the single cell analysis.

- We are happy the Reviewer appreciates the effort and work we have put into this manuscript. Below we are responding to all concerns in detail.

DISCLAIMER: We renamed the KAMP protein/gene in the resubmitted version to FAME (Factor Associated with Metabolism and Energy)

1. I can't find any data to support that Kamp has been successfully knockout. Can the authors include some genotyping data? It is also strongly advised the authors to check Kamp expression from their scRNA-seq data and compare the expression between wild type and KO. This will reassure that Kamp has been knockout'ed in multiple kidney cell types.

- This is a great comment, and we have performed antibody staining of FAME (former KAMP) in WT and KO kidney tissues to control for i) the antibody's validity and the successful knockout (new Supplementary Figure 4). We clearly see the absence of the FAME (former KAMP) protein in KO tissues, whereas we detect the protein in glomeruli and tubules of control animals as expected. Therefore, KO works.
Next, we compared the expression of FAME (former KAMP) using our scRNA-seq data. Our analysis revealed that there is no statistically significant difference between Fame mRNA expression in KO and WT samples. We present the sequencing results of the region of interest in the wild type and knockout samples to show that knockout was successful. The sequencing reads are visualized by the IGV desktop application (<https://software.broadinstitute.org/software/igv/>), and the data are shown in the **new Supplementary Figure 8**.

2. For the scRNA-seq data in figure 5, is the PT1 the only cell type that expressed Kamp? The authors should plot Kamp expression in the whole scRNA-seq data and see what cell types expressed Kamp.

- We agree with the Reviewer and have replotted the expression of Fame (former KAMP) in the scRNA-seq dataset. These data are now presented in new panels in Supplementary Figures 13B and C. *Fame* is expressed in each cluster, but most present in the proximal tubules, collecting duct, loop of Henle, and distal convoluted tubule-associated clusters. PT1 was picked for the manuscript Figure 5 as it has the most differentially expressed genes between wildtype and KO conditions.

3. The authors should generate a plot to highlight the number of DE genes identified from each cell type by comparing wild type versus KO. This Figure can support the conclusion that "This suggests that the knockout phenotype has a molecular nature without noticeable defects at tissue organization levels"

- We have replotted and highlighted the number of DE genes identified from each cell type comparing WT and KO. These data are now presented in the **new Supplementary Figure 8**.

4. In Figure 5a, the authors named a cluster GL endo. That might be not accurate because from supplementary figure 7, this cluster only expresses pan-EC markers such as Pecam1 and Kdr. The authors need to show that this cluster also expressed the glomerular EC specific marker Ehd3 in order to claim that this is a GL endo. Otherwise, just simply annotate it as EC.-

- We thank the Reviewer for this comment and are now showing the expression of Ehd3, Kdr, and Pecam1 in the GL endo cluster (**new Supplementary Figure 8e**).

5. From Figure 5c, it is very interesting to note that Tpt1 is down regulated after Kamp is KO. Tpt1 is a tumor gene that regulates cell growth and proliferation. Does this data link to one of the conclusions the authors made about the correlation of Kamp with cancer? When the authors overexpressed Kamp in HEK293 cells (Figure2d-g), any proliferation phenotype was observed?

- This is indeed an exciting point. To investigate the effect of FAME (former KAMP) on cell growth, we now performed (in addition to the overexpression experiment in Figure 7) a new CRISPR CAS9 Knockout in HEK293 and A549 cells (new panels in E-F in Figure 7). Since HEK293 cells do not express FAME (former KAMP) (confirmed by qPCR and in accord with the human protein atlas), we did not observe changes in cell growth and proliferation. Strikingly, A549 KO cells expressing FAME (former KAMP) endogenously had significantly higher cell proliferation than the control. Furthermore, overexpression of FAME (former KAMP) in these cells led to a drastic reduction in cell growth, similar to that shown in HEK293 cells. These data are now integrated into the updated **Figure 7**.

6. The data will be more convincing if the authors can choose some of the genes from scRNA-seq, and perform immunohistochemistry to validate. I would suggest the authors to choose some DE genes related to proliferation, metal metabolism, or fatty acid metabolism for validation since those genes, if validated, can add more evidence to support the role of Kamp in cancer, energy consumption, and metal homeostasis.

- Despite long and painful efforts, we were not able to perform reliable immunohistochemistry experiments using antibodies for FTH1 (2495-MSM1-P1_ThermoFisher), TPT1 (MA5-32830_ThermoFisher), SLC25a39 (NBP1-

59600_Novusbio), and CYP4A11 (ab3573_abcam) during the time of the revision. To compensate for the absence of reliable antibodies, we performed a sophisticated IP analysis combined with Bio-ID method (Figure 3), as well as generated CRISPR-KO in cell lines (Figure 7), and created the second alternative KO mouse model with broad phenotyping (**new Supplementary Figure 9, 10, 11, 12, 13, 15 and 16**). After these additional experiments, we generated a stronger support for the role of FAME (former KAMP) in cancer and energy processing.

7. The authors included both male and female mice in the single cell experiment, can the authors specify how many male and female mice are used in the wild type and KO group respectively? I ask this because a previous study has shown that there is strong sex dimorphism observed in the proximal tubule (PMID: 31689386). That may potentially affect the data interpretation in the PT data. For example, the differential genes shown in Figure 5c can also be sex specific genes.

- In the analysis, we used a total of five samples: (i) female wildtype, (ii) female KO, (iii) male KO, (iv) mixed (female and male) wildtype, and (v) mixed (female and male) KO. The first three were collected and sequenced simultaneously, and the last three were added later to increase the number of cells (and thus detected genes). All the samples were evenly distributed among the clusters in the final UMAP embedding (except for the female-specific PT_f cluster). To check whether DE genes between wildtype and knockout are not sex-specific, we compared them with the list of the corresponding DE genes between female and male proximal tubule (PT) samples from the paper kindly provided by the Reviewer. The genes whose adjusted p-values are less than 0.01 were excluded from the comparative analysis. Both lists have a little intersection; moreover, in the case of these few intersected genes, the most significant DE genes between wildtype and KO were not even in the top 40 sex-specific genes. The exceptions are *Inmt* (male-specific) and *Spp1* (female-specific) genes. The most intriguing findings are that the FAME-binding iron-transporting *Fth1* and potentially iron homeostasis associated *Slc25a39* gene seems to be gender neutral. These Data are now outlined in **Table 10** of the revised manuscript.

8. Supplemental Figure 8a-e, are the data about comparing the low iron diet in KO versus wildtype significant? The authors should show the p-value in the figure legends.

- We have performed the relevant statistic analysis on these samples and included all p-values in the figure legends. Furthermore, we have moved the data from the supplementary Figure to the main **Figure 4**.

We thank the Reviewer for all constructive comments and hope that the manuscript improves after revision.

Reviewer #4 (Remarks to the Author):

In the manuscript entitled "A previously uncharacterized Kidney-Associated Membrane Protein (KAMP) is associated with evolutionary adaptation, energy balance and kidney physiology", Petersen et al. have shown the importance of Kamp (Kidney-associated membrane protein) in evolutionary adaptation and kidney homeostasis. They have indicated that Kamp is involved in energy expenditure and scaling of organ with the experiments using Kamp knockout mice, especially in iron-deficit condition. Besides, they have suggested the possibility that Kamp plays an important role in cancer proliferation and invasion. Overall, the manuscript contains some novel information. The idea that Kamp is a crucial role in kidney physiology is attractive. However, there are several important concerns that should be addressed adequately.

- We are glad that the Reviewer is intrigued by our study and are grateful for the valuable suggestions. We performed many additional experiments during this revision to address the provided comments, and we hope they will give us a better picture of a protein function and evolution.

DISCLAIMER: We renamed the KAMP protein/gene in the resubmitted version to FAME (Factor Associated with Metabolism and Energy)

Major concerns:

1. Albuminuria is significantly augmented in KAMP KO mice, KAMP is highly expressed at tubules in the Kidney. Do the authors think that the main reason for albuminuria is an insufficient reabsorption of albumin in tubules? It should be very important to confirm the origin of albuminuria. In this regard, histological analysis (especially electronic microscope analysis) of the glomerulus is mandatory to exclude the possibility that glomerulus is the cause. Then, functional roles of the specific molecules that reabsorb proteinuria at proximal tubules (megalin, etc.) needs to be investigated.

- We fully agree with the Reviewer about the importance of investigating the origin of albuminuria in Kamp KO mice. Therefore, we performed histological and ultrastructural analyses of KO and WT kidneys. These showed normal kidney histomorphology and unaltered glomeruli in KO animals (see new **Supplementary Figure 6**), as confirmed by kidney experts Prof. Dr. Boor. The regular appearance of glomerular filtration barriers and podocyte foot processes indeed exclude protein leakage at the glomeruli and suggest protein reabsorption defects in the proximal tubules. We discussed it in the main text:
- *"Histological analyses of PAS sections showed normal histomorphology in both wt and FAME knockout mice, without any signs of pathological alterations of glomeruli, vessels or tubulointerstitium (**Supplementary Figure 6**). Since kidney function parameters, particularly proteinuria, can be affected by changes only visible at the ultrastructural level we have also analyzed the kidneys using transmission electron microscopy. The*

analysis showed an intact and normally developed filtration barrier of the glomeruli (Supplementary Figure 6), with regular shaped podocyte foot processes (arrowhead), regular glomerular basal membrane (star arrow) and thin fenestrated endothelium of glomerular capillaries. Also, the tubular cells showed a normal ultrastructural appearance with a prominent brush border in proximal tubuli and high amounts of mitochondria. No signs of metabolic stress were observed, including no signs of intracellular accumulations of lipids or glycogen, increased vesicles, or high lysosomal activity."

Unfortunately, our megalin and other experiments with specific reabsorption-related molecules were not fruitful during the short time of this revision. We will continue this research and hope to obtain these data during follow-up studies. This paper focused on KO and a very broad and general characterization of FAME function. For this, we created a second KO mouse in a different way and on a different genetic background, and we repeated the entire cycle of phenotyping. Due to new data and highly variable role of the protein in organismal Energy and metabolism, we shifted the main focus away from Kidney, which requires much more time and work, towards other possible roles of FAME. We also renamed KAMP as FAME (Factor Associated with Metabolism and Energy), highlighting this shift towards more generic functions. It seems the major roles of the protein are not restricted to the kidney domain. We are sorry that we could not advance the molecular kidney phenotype further during this revision. We will continue our attempts.

2. In related to the point above, main site of KAMP expression is tubules. However, I cannot understand why the authors evaluated the glomerulus size but not morphological or functional changes in tubules. In addition, I believe that the authors cannot conclude that "these results suggest insufficient kidney function and reabsorption defects" from the data in this manuscript.

- We agree with the Reviewer, and, as stated in the comment above, we have now carefully analyzed the kidney from WT and KO mice using ultrastructural analyses, both tubules and glomeruli. The regular appearance of glomerular filtration barriers and podocyte foot processes in KO kidneys suggests the exclusion of protein leakage at the glomeruli. It supports the hypothesis of protein reabsorption defects in the proximal tubules. Therefore, after obtaining this additional data, we have rephrased the text into a softer and more suggestive interpretation:

"Histological analyses of PAS sections showed normal histomorphology in both wt and FAME knockout mice, without any signs of pathological alterations of glomeruli, vessels or tubulointerstitium (Supplementary Figure 6). Since kidney function parameters, particularly proteinuria, can be affected by changes only visible at the ultrastructural level we have also analyzed the kidneys using transmission electron microscopy. The analysis showed an intact and normally developed filtration barrier of the glomeruli (Supplementary Figure 6), with regular shaped podocyte foot processes (arrowhead),

regular glomerular basal membrane (star arrow) and thin fenestrated endothelium of glomerular capillaries. Also the tubular cells showed normal ultrastructural appearance with a prominent brush boarder in proximal tubuli and high amounts of mitochondria. No signs of metabolic stress were observed, including no signs of intracellular accumulations of lipids or glycogen, increased vesicles or high lysosomal activity"

3. As authors noted, ferritin is the major intracellular iron storage protein, and ferritin heavy chain 1 and KAMP have intracellular interaction. Besides, the authors showed that genes associated with iron transport such as Fth1 and Slc25a39 are not much altered in KO mice. I cannot understand why this leads to lower serum ferritin observed in KAMP KO mice. Serum Fe, hemoglobin and hematocrit level should be evaluated.

- We would like to point out here that Fth1 and Slc25a39 are significantly altered in KO (please see **Figure 5 and Table 9**). However, we have performed various additional measurements of different blood parameters, including hemoglobin and hematocrit levels, which are not altered (**new Supplementary Figures 11 and 12**). It seems that the FAME (former KAMP) has a complex tuning role in these processes and only affects some parameters, being non-essential for basic functions.

4. It is not clear why the mice born from the KAMP KO female mice showed organomegaly. Since it is not clear whether the changes in iron metabolisms or anemia itself affect organ sizes, it should be investigated whether effect of anemia other than iron-deficit could cause similar changes.

- We apologize for not being clear in the first submitted version. The KO mice are not born with organomegaly. Instead, the organ size reduction is observed (see updated **Figure 4B**). Also, as the Reviewer suggested, we have now investigated if our mice are anemic. Therefore, during this revision, we have examined several different blood parameters. Here, only the platelet counts in females with an *FVB/Ant* background (**new Supplementary Figure 12**) and the eosinophil counts in females with a *C57Bl/6NCrl* background (**new Supplementary Figure 11**) are significantly altered. Therefore, we concluded that FAME KO (former KAMP KO) mice are not anemic, although the iron deficiency leads to the observed smaller size of investigated inner organs.

5. They showed that overexpression of Kamp in HEK293 cells decreased cell proliferation in vitro. It should be determined whether Kamp deletion actually induces cell proliferation. In addition, cell cycle analysis should be evaluated. It should also be important to determine whether KAMP is associated with ferroptosis.

- We are grateful for this valuable comment and have created two different CRISPR-Cas9 Knock-out cell lines (in HEK293 and A549 cells). The results from these experiments are now presented in updated **Figure 7**. The knockout of FAME (former KAMP) in A549 cells shows that cell proliferation is increasing. The proliferation of HEK293 cells was not affected. This is due to the fact that HEK293 cells do not express FAME from the

beginning (confirmed by qPCR and according to the human protein atlas). Furthermore, the Ferroptosis experiments in A549 cells showed that the knockout of FAME does not influence ferroptosis. These data are now part of **Supplementary Figure 19**.

6. It is not clear whether expression of Kamp is associated with survival of cancer patients. If they wish to show the association between Kamp and cancer prognosis, it is desirable to show whether Kamp deletion or overexpression affects cancer prognosis in rodents.

- In this manuscript, we use the human data of gene expression in tumors and survival curves of corresponding patients. The analysis of expression levels of FAME (former KAMP) correlates statistically with specific patient survival trends. Notably, we do not claim that FAME is causing better or worse survival and only mention correlative results. This might mean that FAME can be a part of a specific prognosis-related signature without being a driver or upstream of this signature. In line with the Reviewer's suggestion, we improved our text describing this in Results and made our conclusions more modest:

"The fine-tuning and pleiotropic functions of FAME discovered in animal models suggested that FAME might be associated with human tumor development. The analysis of expression levels of FAME in human kidney tumors differentially correlated with patient survival trends with a p-value close to significance in some cases. Although this analysis of survival curves of cancer patients did not result in predictions of strong effects, our experimental analysis of FAME localization in healthy versus tumor tissues suggested the potential importance of FAME in kidney tumors. Furthermore, our experiments with knockout and overexpression of FAME in human cell lines (HEK293 and A549) with and without endogenous expression of FAME showed the role of FAME in controlling cell proliferation. Further, they supported the potential role of FAME in cancer. However, to answer the direct question if FAME represents a cancer progression suppressor or driver (and not a passenger molecule, being only a part of the prognostic signature) in specific tumors or heterogeneous tumor cell populations, further experiments in rodents involving PDX (patient-derived xenograft) models are required."

Minor concerns:

1. They showed that KAMP is highly expressed in loop of Henle and collecting duct cell, but also in the proximal tubule. Do they express in lumen side or basolateral side? Please also discuss why KAMP KO does not result in much difference in electrolyte excretion.

- We observe main expression of FAME (former KAMP) at the basolateral side. However, some lower expression is also expressed at the lumen side as seen in **Figure 2**. As we do not know the exact molecular role (or a multitude of roles in different cell types) of FAME (former KAMP), it is hard to speculate why KO did not result in a strong

difference in electrolyte excretion. Nevertheless, we suggested that the protein roles are related instead to the fine modulation of molecular processes instead of being crucial for any basic function, including major characteristics of electrolyte excretion. Due to the amount of new data, including multiple blood parameters and characterization of the second knockout mouse line, we decided to remove the electrolyte excretion data from the manuscript as not clarifying anything, and focus in a separate manuscript on this particular aspect more in depth.

2. It seems that there is heterogeneity in KAMP staining in Kidney in that KAMP is not stained in some tubules. The authors should show better pictures and the reason for this should be discussed.

- We agree, so we repeated the validation experiments with staining and found the situation shown in the early version of the manuscript consistent (the images accurately correspond to the realistic situation in multiple staining rounds and independent experiments). Our new validation (**Supplementary Figure 21**) shows that only some tubules are stained as was seen before, and we provide a discussion of this in the main text:

"It is interesting to note that FAME is not stained in some tubules. We were unable to determine the reason for this, but it may be due to differences in FAME expression levels within the tubules depending on the region and section plane. It is possible that our antibody can only detect a certain threshold of FAME."

3. The meaning of Figure 4e is not clear. Please be clear about what you want to show with these data.

- We apologize for not being clear enough. Figure 4e shows the evaluation of the kidney glomerulus size. We improved the description of the Figure in the revised manuscript.

4. Please discuss the possibility that increases in body weight and lean body mass might be attributable to less fine movements or the changes in metabolic status.

- We revised the manuscript and discussed this in the Discussion section:
*" These two knockout models on different genetic backgrounds revealed that depending on the exact genetic background, the body weight and lean body mass show different significant alterations compared to the controls of the same background. The reason for these differences can be multifaceted and might include the complex and divergent context of differently expressed interacting molecules (for instance, the controls of different backgrounds showed differences in the excretion of Albumin and levels of ferritin in serum, please see **Figure 4 and Supplementary Figure 9**). Furthermore, the differences in fine movement and the changed metabolic status in both animal groups can cause differences in body weight and lean body mass"*

5. Please discuss why water content is higher in KO group?

- At this point, we cannot explain the higher water content without going into unfounded speculations. It seems that FAME (former KAMP) has a pleiotropic role possibly affecting multiple processes in different organs of the body. We had no resource to follow the water content line during this revision to make a better-founded conclusion, and we decided to showcase the other more meaningful data instead in the main text and figures. The water content data are now preserved in the reported phenotyping results (**Table 11 and 12**).

6. They indicated that food seeking activity of KO animals was less pronounced. I cannot understand why the metabolic changes caused by KAMP deletion can lead to reduced food seeking activity. Is there any possibility that appetite itself is altered due to the changes in brain or appetite-related hormones such as ghrelin or leptin? If you have plasma or serum samples, please measure these parameters.

- In response to this comment, we experimentally measured ghrelin and leptin in KO and WT animals in two different genetic backgrounds. We could not observe a difference in ghrelin and leptin levels (new **Supplementary Figure 10**). Therefore, we cannot unambiguously answer this question and hope that our or someone else's future research will solve this puzzle.

7. The authors have indicated that birds showed higher average of Kamp dN/dS than in reptiles. It seems that birds exhibit rather higher number of dN/dS than mammals. It should be discussed whether Kamp plays an important role also in birds, and the difference in significance of Kamp between mammals and birds. Indeed, the Kidney of mammals and birds is different in that birds have renal portal vein and do not have bladder.

- We agree with the Reviewer that anatomical and metabolic differences between birds and mammals might explain the higher evolutionary divergence of FAME (former KAMP) in birds. We introduced the following passage into the text of our Discussion sections:

“Overall, FAME appears to be a non-essential gene involved in tuning the metabolism, energy expenditure, and excretion processes. Being a fine-tuning and fast-evolving factor, it influences the evolutionary diversification of amniote animal groups and shows diverse effects depending on exact genetic background, metabolic needs and sexes of the animals. As the result, the high evolutionary speed of changes observed in FAME structure, and its correlation with bird speciation might be defined by specifics of bird's metabolism, excretion and anatomy. For instance, birds excrete ureic acid (unlike mammals excreting water-soluble urea), show the presence of a renal portal vein, and do not have a bladder. The last, but not the least, FAME might be a potential confounder of tumor progression and human metabolic disorders according to our experimental data and published GWAS analysis.”

8. The authors have indicated that pancreas or liver are also the organs highly express KAMP. It is rather understandable that the changes in these organs induce phenotype changes in

KAMP KO animals since pancreas and liver play rather dominant roles in metabolism. Please discuss these possibilities.

- We discussed the potentially important role of FAME (former KAMP) in pancreas and liver in relation to the discovered phenotype. We inserted several passages into Results and Discussion sections:

"Finally, since FAME is expressed in other inner organs, such as the pancreas and liver, which play rather dominant roles in metabolism (Wasserman, O'Doherty et al. 1995), it is critical to focus on these organs and corresponding interacting molecules in future studies in more detail"

References:

Wasserman, D. H., R. M. O'Doherty and B. A. Zinker (1995). "Role of the endocrine pancreas in control of fuel metabolism by the liver during exercise." Int J Obes Relat Metab Disord **19 Suppl 4**: S22-30.
Zhang, X., A. H. Smits, G. B. van Tilburg, H. Ovaa, W. Huber and M. Vermeulen (2018). "Proteome-wide identification of ubiquitin interactions using UbIA-MS." Nat Protoc **13**(3): 530-550.

REVIEWER COMMENTS

Reviewer #1 (Remarks to the Author):

1. In my original review, I suggested that the authors simply demonstrate that the FAME/KAMP/CCDC198 protein is actually expressed in mouse tissues. I considered this to be a very minor request. Unfortunately, they have yet to accomplish this very straightforward ask. In fact, as far as I can tell, the additional work included in the revised manuscript seems to provide evidence that FAME is actually NOT expressed as a protein in mouse tissues.

The authors seemed to have given this a good try - multiple antibodies were purchased from various commercial sources, but none of them were able to detect any endogenous protein by Western blot.

I note that a Western blot (containing lysates from multiple mouse tissues and human cell lines) probed with the antibody that the authors used in the manuscript appears to show a nice, single band in all lanes migrating at ~43kDa - this is on the Santa Cruz website, here: <https://www.scbt.com/p/ccdc198-antibody-b-1?requestFrom=search>

This signal was apparently not ever observed by the authors - in any of their blots. This could mean that the antibody simply does not work as advertised - a pretty common occurrence.

However, when the authors overexpressed an EGFP-tagged FAME protein in 293 cells, they demonstrate that they CAN detect this protein with the Santa Cruz antibody, arguing that the antibody can successfully detect the FAME/CCDC198 protein.

Since the authors detected no endogenous protein on any Western that they tried with mouse tissues, or in 293 cell lysates, this would suggest to me that the cells do not actually express any endogenous FAME - or that they express the protein at such low levels that it is not detectable by this antibody.

So, I honestly don't know how to interpret these results, other than to say that the current data is just not compelling or convincing that FAME is expressed endogenously in the cells and tissues studied here.

As a side note, the suggestion that this putative membrane protein is not solubilized by cell lysis buffers is also pretty unbelievable - if the Santa Cruz Western blots are to be believed, they managed to solubilize the protein just fine in many cells/tissues. And - at least to my knowledge - the inability to solubilize a protein in Laemmli sample buffer is quite rare. People use this method to do Westerns on membrane proteins all the time. Regardless, it seems that the authors disprove their own argument by successfully detecting the EGFP-tagged FAME protein by Western from lysed 293 cells. So none of this stuff makes any sense to me.

2. These same issues call all of the IP-MS data into question, too. If, as the authors argue, they are unable to liberate endogenous FAME with their lysis buffer, how can the same buffer (or even "gentler", low detergent buffers normally used in IP approaches) be used to study the exogenous FAME protein interactome, if it is actually localized properly, to the same (presumably insoluble) cellular location? This whole thing seems quite illogical to me.

3. Being undeterred, the authors also conducted IF with this same antibody. Supplementary Figures 21 and 23 are particularly confusing. I don't know what the authors were trying to show here, other than - as in Westerns - the antibody is able to detect exogenously expressed EGFP-FAME expressed in cells (the red and green signals somewhat overlap - in the few cells shown in the figure). They again do not show that it can detect endogenous protein. They do demonstrate that there might be some difference in the signal detected in WT vs KO animals, but it is not very convincing, showing just a couple of presumably "best case" micrographs.

4. Also of huge importance, since the authors have no way of monitoring FAME protein expression,

how do they know that it is not expressed in their KO mice? in my mind, this would be critical to demonstrate.

>Bottom line: At present the authors have simply not convincingly demonstrated that their gene is expressed at the protein level in mice. They have thus not ruled out the possibility that this is just a pseudogene, or perhaps a functional RNA, instead. The lack of a strong phenotype in two different KO mice could also be interpreted to mean that the authors are working with a pseudogene.

I strongly suggest that the authors work with a collaborator to analyze mouse tissues using mass spectrometry to prove that the FAME protein is actually present in these organs (and at the same time, they can report on relative expression levels in different tissues, etc).

5. The myristoylation of the EGFP-tagged protein in 293 cells is much more convincing now. However, I still see no evidence that the endogenous protein is expressed in kidney and similarly modified.

6. One final, important issue with the protein work: As claimed on lines 250-260 (regarding Figures 3f-h), the authors do not "validate" that FAME interacts with any of the proteins detected in Y2H, IP-MS or BioID. They show that FAME can co-localize with a few of the hits in these screens. This is very different, and the authors have hugely over-interpreted their data here. This entire section of the manuscript is very weak.

7. And what is "the catalytic complex"?

Reviewer #2 (Remarks to the Author):

Peterson made a major revision, performing many additional experiments requested by the reviewers. This was due in part to the original version of the paper having claims that either were not substantiated well enough or inadvertently not presented clearly. Now the paper is much stronger shape, and I congratulate the authors for that. It is good that the authors changed the name of the gene to remove the word kidney from it and include energy. I have just two minor comments:

One thing I did not consider before is that birds and mammals are convergent for warm-blooded body temperature regulation. Could FAME have something to do with that? I suggestion to consider bringing up in the discussion.

In the abstract and elsewhere the author say FAME is a previously uncharacterized gene. But then it was mentioned in GWAS studies. I think the authors mean a previously identified gene with a previously uncharacterized function?

Reviewer #3 (Remarks to the Author):

The manuscript has been improved a lot in the revised form. The authors have clear all my concerns.

Reviewer #4 (Remarks to the Author):

The authors performed several additional experiments to address the concerns raised by this reviewer. As a result, revised manuscript is much improved. I have no further concern. Congratulations!

Below we provide a point-by-point response to reviewer comments:

REVIEWER COMMENTS

- We are delighted that Reviewers #2, #3, and #4 are satisfied with our extensive revisions. However, we acknowledge that there are still concerns raised by Reviewer #1 regarding the expression of the gene as a protein in different tissues. In response, we have provided additional evidence to address these concerns and to support our assertion that the gene is indeed expressed as a protein in various tissues. We hope our efforts will satisfy Reviewer #1 and that our revised manuscript will be accepted for publication.

Reviewer #1 (Remarks to the Author):

1. In my original review, I suggested that the authors simply demonstrate that the FAME/KAMP/CCDC198 protein is actually expressed in mouse tissues. I considered this to be a very minor request. Unfortunately, they have yet to accomplish this very straightforward ask. In fact, as far as I can tell, the additional work included in the revised manuscript seems to provide evidence that FAME is actually NOT expressed as a protein in mouse tissues.

The authors seemed to have given this a good try - multiple antibodies were purchased from various commercial sources, but none of them were able to detect any endogenous protein by Western blot.

I note that a Western blot (containing lysates from multiple mouse tissues and human cell lines) probed with the antibody that the authors used in the manuscript appears to show a nice, single band in all lanes migrating at ~43kDa - this is on the Santa Cruz website, here: <https://www.scbt.com/p/ccdc198-antibody-b-1?requestFrom=search>

This signal was apparently not ever observed by the authors - in any of their blots. This could mean that the antibody simply does not work as advertised - a pretty common occurrence.

However, when the authors overexpressed an EGFP-tagged FAME protein in 293 cells, they demonstrate that they CAN detect this protein with the Santa Cruz antibody, arguing that the antibody can successfully detect the FAME/CCDC198 protein.

Since the authors detected no endogenous protein on any Western that they tried with mouse tissues, or in 293 cell lysates, this would suggest to me that the cells do not actually express any endogenous FAME - or that they express the protein at such low levels that it is not detectable by this antibody.

So, I honestly don't know how to interpret these results, other than to say that the current data is just not compelling or convincing that FAME is expressed endogenously in the cells and tissues studied here.

As a side note, the suggestion that this putative membrane protein is not solubilized by cell lysis buffers is also pretty unbelievable - if the Santa Cruz Western blots are to be believed, they managed to solubilize the protein just fine in many cells/tissues. And - at least to my knowledge - the inability to solubilize a protein in Laemmli sample buffer is quite rare. People use this method to do Westerns on membrane proteins all the time. Regardless, it seems that the authors disprove their own argument by successfully

detecting the EGFP-tagged FAME protein by Western from lysed 293 cells. So none of this stuff makes any sense to me.

- Thank you for reviewing our research thoroughly and expressing your concerns regarding the FAME protein expression in mouse tissues. We appreciate your attention to detail and the valuable feedback you have provided.

Though, we do not fully understand the skepticism. We acknowledge that our antibody may not be ideal for detecting the FAME protein. However, we have demonstrated that it can detect overexpressed FAME through our overexpression experiments in HEK293 cells and our IP validation data. Additionally, we could detect FAME on histological sections but failed to detect endogenous FAME by Western blot.

The reasons for this could be simple: the sensitivity of the antibody is reduced in denaturing conditions of the Western blot due to changed structure of the antigen. This is not uncommon. This might be combined with proportionally low protein levels in bulk tissues (although it might be high in rare cell types in the tissues at the same time) used for Western blot. As we cannot enrich the cells of collecting tubules, we do not see a way to improve this part because we have already tried.

2. These same issues call all of the IP-MS data into question, too. If, as the authors argue, they are unable to liberate endogenous FAME with their lysis buffer, how can the same buffer (or even "gentler", low detergent buffers normally used in IP approaches) be used to study the exogenous FAME protein interactome, if it is actually localized properly, to the same (presumably insoluble) cellular location? This whole thing seems quite illogical to me.

- Regarding the suggestion that cell lysis buffers do not solubilize the putative membrane protein, we would like to clarify that we did not argue that the sample buffer does not solubilize the protein. Instead, we suggested that the protein may be sensitive to the detergent used in the lysis buffer, which could affect its solubility, conformation, antigen masking, and subsequent detection by antibody in Western blot. This is a common issue with membrane-associated proteins and might require optimization of the lysis buffer conditions for successful detection. Since the overexpressed protein in our IP-MS experiment is detected by the antibody everywhere, we agree that this issue seems minor.

3. Being undeterred, the authors also conducted IF with this same antibody. Supplementary Figures 21 and 23 are particularly confusing. I don't know what the authors were trying to show here, other than - as in Westerns - the antibody is able to detect exogenously expressed EGFP-FAME expressed in cells (the red and green signals somewhat overlap - in the few cells shown in the figure). They again do not show that it can detect endogenous protein. They do demonstrate that there might be some difference in the signal detected in WT vs KO animals, but it is not very convincing, showing just a couple of presumably "best case" micrographs.

- We would like to present a detailed summary of our experiments to clarify the reviewer's concerns.

Since FAME cannot be detected in HEK293 cells as they do not express it endogenously according to RNA sequencing and our own qPCR experiments, we overexpressed the protein with a GFP-tag (FAME was cloned without its stop-codon into a pEGFP-N1 plasmid leading to the expression of a FAME-pEGFP fusion construct) and used an antibody to label the same protein using a different fluorophore (in this case, Alexa 555). Consequently, only cells expressing our GFP-tagged protein should exhibit detectable levels of FAME, provided that the antibody recognizes it. This was demonstrated in previous **Supplementary Figure 21**. However, we noticed, that the antibody didn't detect FAME ubiquitously in the cells expression FAME-pEGFP. Therefore, we conducted additional experiments, where we utilized the same setup but included the use of DAKO antigen retrieval solution before staining the cells. New **Supplementary Figure 6** displays the results of these experiments. These results offer additional proof that the antibody is functional and that protein conformation influences the specificity of the antibody, which is positively affected by DAKO antigen retrieval.

HEK293 cells transfected with FAME-pEGFP-N1 and stained with anti-FAME Antibody (without heat antigen retrieval)

HEK293 cells transfected with FAME-pEGFP-N1 and stained with anti-GFP and anti-FAME Antibody after heat antigen retrieval

We would like to emphasize that the reviewer seems to overlook the fact that we have provided additional staining of FAME in the liver (previous Supplementary Figure 21d)

and kidney (previous **Supplementary Figure 21c**). This particular staining is absent in our KO samples. This was done specifically to ensure that "we do not supply just a couple of best micrographs". We extended these experiments, and these results are stable and reproducible. Please, check, new **Supplementary Figure 5** where we now included staining's from different animals and different organs including FAME expression in the neural tube of an E9.5 mouse embryo. Using a single-cell transcriptomics datasets of the murine neural crest lineage and neural tube we found a highly specific expression of FAME mRNA in the pre-delaminating dorsal neural tube. This prediction showed that FAME must appear in a small subpopulation of the dorsal neural tube just underneath the epithelial layer. Therefore, we stained for the protein on cross-sections of an embryos at the E9-9.5 stage, which matches the time point of the mRNA single-cell sequencing. As expected, we observed a highly specific staining (with very low or no background) in the predicted position in a group of cells in the dorsal neural tube. Furthermore, the staining was mainly membranous and, to lower extents, cytoplasmic, corresponding to other experiments and validations in this manuscript.

Altogether, these experiment demonstrate, that the FAME antibody recognizes the protein in vivo in native, unperturbed mouse tissues.

NOTE: In our previous revision, we mistakenly labelled the KO as WT.

4. Also of huge importance, since the authors have no way of monitoring FAME protein expression, how do they know that it is not expressed in their KO mice? in my mind, this would be critical to demonstrate.

- We have performed antibody staining of FAME in WT and KO kidney tissues to control for the antibody's validity and the successful knockout (Figure 2c and new

Supplementary Figure 5). We clearly see the absence of the FAME protein in KO tissues, whereas we detect the protein in glomeruli and tubules of control animals as expected. Therefore, the KO has worked.

Furthermore, we present the sequencing results of the region of interest in the wild type and knockout samples to show that knockout was successful. The sequencing reads are visualized by the IGV desktop application (<https://software.broadinstitute.org/software/igv/>), and the data are shown in **Supplementary Figure 10.**

Bottom line: At present the authors have simply not convincingly demonstrated that their gene is expressed at the protein level in mice. They have thus not ruled out the possibility that this is just a pseudogene, or perhaps a functional RNA, instead. The lack of a strong phenotype in two different KO mice could also be interpreted to mean that the authors are working with a pseudogene. I strongly suggest that the authors work with a collaborator to analyze mouse tissues using mass spectrometry to prove that the FAME protein is actually present in these organs (and at the same time, they can report on relative expression levels in different tissues, etc).

Thank you for bringing this to our attention. However, based on publicly available mass spectrometry data for all major tissues and organs in mice and humans, there is an extensive and irrefutable evidence that FAME is indeed produced at the protein level in different tissues, indicating that it is not a pseudogene or a functional RNA. By utilizing publicly available mass spectrometry data, we further provided evidence for the expression of FAME at the protein level in different tissues and species. For example, FAME/14ORF105 has been identified using liquid chromatography-tandem mass spectrometry as a potential biomarker for opisthorchis viverrini infection and associated cholangiocarcinoma (Aksorn, Roytrakul et al. 2018). Furthermore, the analysis of human sperm proteins from normozoospermic men using 2-dimensional electrophoresis and mass spectrometry identified C14ORF105 as less characterized human sperm proteins (Nowicka-Bauer, Ozgo et al. 2018). Also, by digging into missing and low abundance proteins using an enrichment approach with ProteoMiner Li et al. identified FAME/CCDC198 (Li, He et al. 2017). In addition, FAME/1700011H14Rik has been detected in a recent tissue-specific atlas of mouse protein phosphorylation and expression using strong cation exchange chromatography and phosphopeptide enrichment via immobilized metal affinity chromatography (Huttlin, Jedrychowski et al. 2010). Additionally, quantitative phosphoproteomic analysis in renal collecting duct cells further detected FAME/1700011H14Rik (Rinschen, Yu et al. 2010).

We are now citing this important literature in our revised manuscript: *“By utilizing publicly available mass spectrometry data, we find the evidence for the presence of FAME at the protein level in different tissues and species. For instance, FAME protein is detected in ProteomicsDB (<https://www.proteomicsdb.org/>), Phosphomouse (<https://phosphomouse.hms.harvard.edu>) and PeptideAtlas (<https://db.systemsbiology.net/sbeams/cgi/PeptideAtlas/Search>) public mass spectrometry databases (Deutsch 2010, Huttlin, Jedrychowski et al. 2010, Schmidt, Samaras et al. 2018, Wang, Eraslan et al. 2019). Strong experimental evidence for FAME*

protein production exists in both the human (Li, He et al. 2017) and mouse kidney (Huttlin, Jedrychowski et al. 2010). Furthermore, FAME protein was detected in cultured murine collecting duct cells (Rinschen, Yu et al. 2010), validating the presence of FAME protein in a cell type shown to produce its mRNA in vivo (Figure 2b).

Therefore, we next focused on the kidney and validated the presence of FAME protein in the proximal tubules by immunohistochemistry. This is supported by the fact that we did not detect FAME in samples from knockout animals (Figure 2c and Supplementary Figure 5). Importantly, we made sure that our antibody is functional and specific via detecting FAME as a part of FAME-GFP fusion in cultured cells that do not produce FAME on their own (Supplementary Figure 5 and 6). However, although we validated the functionality of the antibody, we must also acknowledge its limitations connected to potential low sensitivity, which results in inability to detect FAME in western blot without overexpressing FAME, which we discussed in detail in the method section."

Furthermore, the evolutionary analysis of protein conservation and its domain structure strongly support that it cannot be a pseudogene or functional RNA. The conserved coiled-coil domain shows the pressure of stabilizing selection. Most importantly, the CDS contains no premature stop codons, which ensures CDS translation if the RNA is transcribed. As we demonstrated in this manuscript, the RNA is produced extensively, and the differential analysis for organs and tissues is available from independent GTEx studies and other RNA expression atlases, such as: <https://www.genecards.org/cgi-bin/carddisp.pl?gene=CCDC198> and <https://gtexportal.org/home/>. Finally, GWAS data presented in this manuscript support the importance of FAME in human diseases.

5. The myristoylation of the EGFP-tagged protein in 293 cells is much more convincing now. However, I still see no evidence that the endogenous protein is expressed in kidney and similarly modified.

We are pleased that the reviewer acknowledged our efforts in providing evidence of the myristoylation site in FAME. Here we would like to point out that our manuscript highlights several aspects and distinguishing features of a protein that has not been previously characterized. While we could not cover all aspects of the protein's complexity, our findings offer significant insights into its structure and function. We hope that future research will delve deeper into specific aspects of the protein to expand our understanding further.

6. One final, important issue with the protein work: As claimed on lines 250-260 (regarding Figures 3f-h), the authors do not "validate" that FAME interacts with any of the proteins detected in Y2H, IP-MS or BioID. They show that FAME can co-localize with a few of the hits in these screens. This is very different, and the authors have hugely over-interpreted their data here. This entire section of the manuscript is very weak.

We have now changed this in the revised manuscript and wrote, "*From these positive correlations, we could show the co-localization of genes specific for the microtubule, mitochondria, and PCP-pathway association of the FAME protein (Figure 3f-h).*"

7. And what is "the catalytic complex"?

In our manuscript, catalytic complex refers to the formal GO term used for computational analysis: GO:1902494 (<http://amigo.geneontology.org/amigo/term/GO:1902494>)

A catalytic complex is a molecular assembly consisting of two or more protein molecules that work together to catalyze a chemical reaction. These complexes are often involved in metabolic pathways or signaling cascades within cells and are essential for many biological processes. Examples of catalytic complexes include enzymes, ribosomes, and proteasomes, which are involved in protein synthesis, degradation, and regulation. The cell tightly regulates the formation and activity of catalytic complexes to ensure proper function and prevent any disruptions to cellular processes.

Reviewer #2 (Remarks to the Author):

Peterson made a major revision, performing many additional experiments requested by the reviewers. This was due in part to the original version of the paper having claims that either were not substantiated well enough or inadvertently not presented clearly. Now the paper is much stronger shape, and I congratulate the authors for that. It is good that the authors changed the name of the gene to remove the word kidney from it and include energy. I have just two minor comments:

Thank you for your positive feedback.

One thing I did not consider before is that birds and mammals are convergent for warm-blooded body temperature regulation. Could FAME have something to do with that? I suggestion to consider bringing up in the discussion.

We introduced a new phrase to point toward such an opportunity.

In the abstract and elsewhere the author say FAME is a previously uncharacterized gene. But then it was mentioned in GWAS studies. I think the authors mean a previously identified gene with a previously uncharacterized function?

The reviewer is correct, and we now refer to a "*previously identified gene with uncharacterized function.*"

Reviewer #3 (Remarks to the Author):

The manuscript has been improved a lot in the revised form. The authors have clear all my concerns.

Thank you for your positive feedback.

Reviewer #4 (Remarks to the Author):

The authors performed several additional experiments to address the concerns raised by this reviewer. As a result, revised manuscript is much improved. I have no further concern. Congratulations!

Thank you for your positive feedback.

Aksorn, N., S. Roytrakul, S. Kittisenachai, K. Leelawat, P. Chanvorachote, S. Topanurak, S. Hamano and U. Lek-Uthai (2018). "Novel Potential Biomarkers for Opisthorchis viverrini Infection and Associated Cholangiocarcinoma." *In Vivo* **32**(4): 871-878.

Deutsch, E. W. (2010). "The PeptideAtlas Project." *Methods Mol Biol* **604**: 285-296.

Huttlin, E. L., M. P. Jedrychowski, J. E. Elias, T. Goswami, R. Rad, S. A. Beausoleil, J. Villen, W. Haas, M. E. Sowa and S. P. Gygi (2010). "A tissue-specific atlas of mouse protein phosphorylation and expression." *Cell* **143**(7): 1174-1189.

Li, S., Y. He, Z. Lin, S. Xu, R. Zhou, F. Liang, J. Wang, H. Yang, S. Liu and Y. Ren (2017). "Digging More Missing Proteins Using an Enrichment Approach with ProteoMiner." *J Proteome Res* **16**(12): 4330-4339.

Nowicka-Bauer, K., M. Ozgo, A. Lepczynski, M. Kamieniczna, A. Malcher, W. Skrzypczak and M. Kurpisz (2018). "Human sperm proteins identified by 2-dimensional electrophoresis and mass spectrometry and their relevance to a transcriptomic analysis." *Reprod Biol* **18**(2): 151-160.

Rinschen, M. M., M. J. Yu, G. Wang, E. S. Boja, J. D. Hoffert, T. Pisitkun and M. A. Knepper (2010). "Quantitative phosphoproteomic analysis reveals vasopressin V2-receptor-dependent signaling pathways in renal collecting duct cells." *Proc Natl Acad Sci U S A* **107**(8): 3882-3887.

Schmidt, T., P. Samaras, M. Frejno, S. Gessulat, M. Barnert, H. Kienegger, H. Krcmar, J. Schlegl, H. C. Ehrlich, S. Aiche, B. Kuster and M. Wilhelm (2018). "ProteomicsDB." *Nucleic Acids Res* **46**(D1): D1271-D1281.

Wang, D., B. Eraslan, T. Wieland, B. Hallstrom, T. Hopf, D. P. Zolg, J. Zecha, A. Asplund, L. H. Li, C. Meng, M. Frejno, T. Schmidt, K. Schnatbaum, M. Wilhelm, F. Ponten, M. Uhlen, J. Gagneur, H. Hahne and B. Kuster (2019). "A deep proteome and transcriptome abundance atlas of 29 healthy human tissues." *Mol Syst Biol* **15**(2): e8503.